# On the Duality Gap of Constrained Cooperative Multi-Agent Reinforcement Learning

**Ziyi Chen, Heng Huang**
Department of Computer Science
University of Maryland
College Park, MD 20742, USA
{zc286,heng}@umd.edu

**Yi Zhou**
Department of Electrical and Computer Engineering
University of Utah
Salt Lake City, UT 84112, USA
yi.zhou@utah.edu

## Abstract

Constrained cooperative multi-agent reinforcement learning (MARL) is an emerging learning framework that has been widely applied to manage multi-agent systems, and many primal-dual type algorithms have been developed for it. However, the convergence of primal-dual algorithms crucially relies on strong duality – a condition that has not been formally proved in constrained cooperative MARL. In this work, we prove that strong duality fails to hold in constrained cooperative MARL, by revealing a nonconvex quadratic type constraint on the occupation measure induced by the product policy. Consequently, our reanalysis of the primal-dual algorithm shows that its convergence rate is hindered by the nonzero duality gap. Then, we propose a decentralized primal approach for constrained cooperative MARL to avoid the duality gap, and our analysis shows that its convergence is hindered by another gap induced by the advantage functions. Moreover, we compare these two types of algorithms via concrete examples, and show that neither of them always outperforms the other one. Our study reveals that constrained cooperative MARL is generally a challenging and highly nonconvex problem, and its fundamental structure is very different from that of single-agent constrained RL.

## 1 Introduction

Cooperative multi-agent reinforcement learning (MARL) (Zhang et al., 2018; Oroojlooy and Hajinezhad, 2022; Chen et al., 2022) is a popular learning framework where multiple agents interact with a dynamic environment independently and communicate with each other to collaboratively optimize their policies to gain more rewards. It has a wide range of applications including coordination of drones (Hammami et al., 2019; Jeon et al., 2022), autonomous vehicles (Garces et al., 2023), and directional sensors (Xu et al., 2020), etc.

Recently, cooperative MARL has been further generalized to constrained cooperative MARL – a more practical setting with safety constraints, in which the agents learn to gain more rewards while constraining their behavior to reduce certain safety-related costs (Diddigi et al., 2019; Oroojlooy and Hajinezhad, 2022). This is an important generalization of cooperative MARL that fits many applications. For example, in multi-agent autonomous driving (Shalev-Shwartz et al., 2016), the pursuit of fluent traffic flow should always obey speed limits and guarantee safety. In drone navigation (Hammami et al., 2019), the drones are subject to constraints on bandwidth and battery power.

In the existing literature, the mainstream approach for solving constrained cooperative MARL problems is primal-dual algorithm (Diddigi et al., 2019; Gu et al., 2021; Lu et al., 2021; Yang et al., 2023; Ying et al., 2023), which applies alternating updates to optimize the Lagrange function associated with the constrained cooperative MARL problem. This is a classic and popular algorithm for solving constrained optimization problems, and it is well-known that its convergence crucially relies on a strong duality condition of the underlying problem, which has been shown to hold for constrained convex optimization problems (Bertsekas, 2014) and constrained RL problems (i.e., constrained cooperative MARL with a single agent) (Altman, 2004; Paternain et al., 2019). However, strong duality has not been formally validated in constrained cooperative MARL, and therefore leaving convergence of the existing primal-dual type algorithms obscure. In fact, constrained cooperative

MARL can be more challenging than the special case of cooperative MARL (without any safety constraint), since intuitively the optimal product policy of cooperative MARL can be ruled out by the complex safety constraints. Hence, we are motivated to study the following fundamental problem.

***Q1:*** *Does strong duality hold for constrained cooperative MARL? Is constrained cooperative MARL more challenging than its special cases of cooperative MARL and constrained RL?*

The existing convergence analysis of primal-dual algorithms for constrained cooperative MARL developed in (Lu et al., 2021; Ying et al., 2023) does not validate the strong duality condition, and moreover, does not characterize the desired constraint violation and optimality of the output policy. Instead, they only establish a convergence result to a certain stationary point with vanishing gradient norm. In particular, Yang et al. (2023) decomposes the agents' policy into a base policy and a perturbation policy, and only the convergence of the perturbation policy update is established given a fixed base policy. This result does not characterize the convergence of the full algorithm. In contrast, the convergence of primal-dual algorithms is very well understood in the special case of constrained RL (with a single agent). There, strong duality has been shown to hold, and the convergence rates of constraint violation and optimality gap have been established (Li et al., 2021; Xu et al., 2021). Therefore, we are further motivated to explore the following problem.

***Q2:*** *If strong duality fails to hold in constrained cooperative MARL, how does the duality gap affect the convergence of the primal-dual algorithm? Moreover, can we develop an alternative algorithm with convergence rates that do not depend on the duality gap?*

## 1.1 OUR CONTRIBUTIONS

In this work, we provide comprehensive answers to the above questions, and show that constrained cooperative MARL is more challenging than its special cases of cooperative MARL and constrained RL. We summarize our contributions below.

We reformulate the constrained cooperative MARL problem as a constrained optimization problem on the occupation measure associated with the agents' product policy. It turns out that the reformulated optimization problem involves a linear objective function, some linear inequality constraints and certain highly nonconvex quadratic constraints, which are induced by the independence of the agents' product policy in the occupation measure space. To the best of our knowledge, such a nonconvex optimization problem has no known polynomial-time algorithm. In contrast, both constrained RL and cooperative MARL, as special cases of constrained cooperative MARL, have provably convergent polynomial-time algorithms. This indicates that the strong duality of constrained RL may no longer hold in constrained cooperative MARL, as elaborated in the next point.

We further construct an example to show that constrained cooperative MARL problems can have a strictly positive duality gap. Then, we reanalyze the convergence of the primal-dual algorithm in constrained cooperative MARL, and establish the first correct convergence rate result that characterizes the impact of duality gap on the constraint violation and optimality of the output policy.

We then propose a decentralized primal algorithm that utilizes decentralized natural policy gradient (NPG) updates to directly solve constrained cooperative MARL problems in their primal forms and thus avoids the duality gap. We develop new technical tools and tight bounds to analyze the convergence of this algorithm, and prove that both the constraint violation and the optimality gap converge at the sub-linear rate $\mathcal{O}\big(\sqrt{\frac{M}{T(1-\gamma)^5}} + \frac{\max_k \zeta_k}{(1-\gamma)^2}\big)$, where $M$ denotes the number of agents and $\zeta_k$ denotes an *advantage gap* induced by the global and local advantage functions. We will show that this advantage gap vanishes if and only if the $Q$ function satisfies a certain factorization structure (See Appendix H for more details). In particular, in the single-agent case, the convergence rates of our primal algorithm strictly improve those of the existing CRPO primal algorithm (Xu et al., 2021) by a factor of $\sqrt{|\mathcal{S}||\mathcal{A}|(1-\gamma)}$. We compare our convergence results with existing works on constrained cooperative MARL in Table 1 in Appendix J.

Lastly, we compare the primal-dual algorithm with the primal algorithm and show that neither of them always outperforms the other in constrained cooperative MARL, both theoretically and experimentally. Specifically, we construct an example where the primal-dual algorithm always generates infeasible policy whereas the primal algorithm converges to the optimal policy at a sublinear rate, vice versa. In particular, the examples we construct involve highly nonconcave constrained maximization problems,

making it challenging to study the convergence of the primal algorithm. Instead of using convex optimization analysis techniques, we prove the convergence of two highly nonconvex potential functions via multi-statement induction in various cases.

## 1.2 RELATED WORK

**Cooperative MARL:** Cooperative MARL has two tasks of interest, policy evaluation and policy optimization. Policy evaluation has been solved by temporal difference type algorithms, including (Wai et al., 2018; Doan et al., 2019; Wang et al., 2020; Sun et al., 2020; Liu and Olshevsky, 2023) for on-policy evaluation and (Macua et al., 2014; Stanković and Stanković, 2016; Cassano et al., 2020; Chen et al., 2021c) for off-policy evaluation. Multiple algorithms have been proposed to solve policy optimization problem, including actor-critic (Foerster et al., 2018; Lin et al., 2019; Suttle et al., 2019; Ma et al., 2021; Chen et al., 2022; Luo and Li, 2022), natural actor-critic (Chen et al., 2022; Luo and Li, 2022), fitted-Q (Zhang et al., 2020), value iteration (Chen et al., 2021a) etc.

**Constrained Markov Decision Processes:** Constrained RL proposed by (Altman, 2004) is a particular case of constrained cooperative MARL with safety constraints but only one agent. Primal-dual algorithms are also popular for constrained RL (Achiam et al., 2017; Tessler et al., 2018; Altman, 2004; Yang et al., 2019; Yu et al., 2019; Stooke et al., 2020; Ding et al., 2020; 2021; Li et al., 2021). There are also other kinds of algorithms for constrained RL, including Lyapunov function based algorithm (Chow et al., 2018; 2019), interior point methods (Liu et al., 2020), policy network that encodes safety constraints (Dalal et al., 2018), and CRPO algorithm (Xu et al., 2021). See (Gu et al., 2022) for a comprehensive review of constrained RL.

**Other constrained cooperative MARL frameworks:** We mainly focus on the main-stream constrained cooperative MARL framework (1) with lower bounds on the total discounted safety score. Some other constrained cooperative MARL frameworks have also been proposed. For example, the constrained cooperative MARL framework in (Liu et al., 2021) has partially observable states and bounds the total discounted safety score as well as the instantaneous safety score. Sheng et al. (2023) proposes a primal-dual algorithm for constrained cooperative MARL with an upper bound on the probability of safety violation. Mondal et al. (2022) uses a mean-field approximation to constrained cooperative MARL with a very large number of agents, which reduces multi-agent policy to a centralized policy, and this approximated problem is solved by a natural policy gradient-based primal-dual algorithm. Shang et al. (2023) proposes a constrained cooperative MARL framework for collaborative multi-phase tasks where each agent focuses on its own value and safety, and proposes a primal algorithm without theoretical analysis.

## 2 CHALLENGE OF CONSTRAINED COOPERATIVE MARL

We consider the standard setting of constrained cooperative MARL (Yang et al., 2023; Diddigi et al., 2019; Gu et al., 2021; Lu et al., 2021), in which $M$ agents explore and make decisions in a common environment. They communicate with each other via a decentralized network $\mathcal{G} = ([M], \mathcal{E})$ where $[M] := \{1, 2, \ldots, M\}$ denotes the set of agents and $\mathcal{E}$ denotes the set of communication links.

At time $t$, every agent $m$ observes the global environment state $s_t \in \mathcal{S}$ and accordingly takes an action $a_t^{(m)} \in \mathcal{A}^{(m)}$ based on its own policy $\pi^{(m)}(\cdot|s_t)$. These agents' policies are independent, and therefore their joint action $a_t = [a_t^{(1)}; \ldots; a_t^{(M)}] \in \mathcal{A}$ is generated by the product policy $\pi(a_t|s_t) := \prod_{m=1}^{M} \pi^{(m)}(a_t^{(m)}|s_t)$. Then, the state $s_t$ transfers to a new state $s_{t+1} \sim \mathcal{P}(\cdot|s_t, a_t)$ following the state transition kernel $\mathcal{P}$, and every agent $m$ receives a reward $r_{0,t}^{(m)} = r_0^{(m)}(s_t, a_t)$ and various safety scores $r_{k,t}^{(m)} = r_k^{(m)}(s_t, a_t)$ $(k = 1, \ldots, K)$, which are assumed to be in $[0, 1]$ throughout. The goal of constrained cooperative MARL is to find the optimal product policy that maximizes the cumulative average reward under various safety constraints, that is,

$$\text{(Constrained cooperative MARL):} \quad \max_{\text{product policy } \pi} V_0(\pi) := \mathbb{E}_\pi \Big[ \sum_{t=0}^{\infty} \gamma^t \overline{r}_{0,t} \Big| s_0 \sim \rho \Big], \tag{1}$$

$$\text{s.t. } V_k(\pi) := \mathbb{E}_\pi \Big[ \sum_{t=0}^{\infty} \gamma^t \overline{r}_{k,t} \Big| s_0 \sim \rho \Big] \geq \xi_k, \quad k = 1, \ldots, K,$$

where the value functions $V_k(\pi), k = 0, ..., K$ denote the expected accumulation of the agents' average reward/safety scores $\overline{r}_{k,t} = \frac{1}{M} \sum_{m=1}^{M} r_{k,t}^{(m)}$ with a discount factor $\gamma \in (0, 1)$, $\xi_k \in \mathbb{R}$ denotes the threshold for the $k$-th safety constraint, and $\rho$ is the initial state distribution.

When there is no safety constraint, problem (1) reduces to a standard cooperative MARL problem that can be solved by many decentralized policy optimization algorithms (Zhang et al., 2018; Chen et al., 2022). On the other hand, when there is only a single agent, problem (1) reduces to a standard constrained RL problem that can be solved by primal-dual algorithms (Altman, 2004; Achiam et al., 2017; Ding et al., 2021). However, as we show next, when imposing safety constraints on multiple cooperative agents, the problem becomes more challenging.

To illustrate the challenges to solve problem (1), we rewrite it using the following occupation measures associated with policy $\pi$, where $\mathbb{P}_\pi$ denotes the probability of visiting a certain $(s, a)$ under $\pi$.

$$\nu_\pi(s, a) := (1 - \gamma) \sum_{t=0}^{\infty} \gamma^t \mathbb{P}_\pi(s_t = s, a_t = a | s_0 \sim \rho), \quad \nu_\pi(s) := \sum_a \nu_\pi(s, a). \tag{2}$$

In particular, there is an almost one-to-one correspondence between a policy $\pi$ and its occupation measure $\nu_\pi(s, a)$, since $\pi(a|s) = \frac{\nu_\pi(s,a)}{\nu_\pi(s)}$ if $\nu_\pi(s) > 0$ (otherwise, $\pi(\cdot|s)$ can be any distribution on $\mathcal{A}$). Then, the value function $V_k(\pi)$ in (1) can be rewritten as a linear function $\widetilde{V}_k(\nu_\pi)$ as follows.

$$V_k(\pi) = \widetilde{V}_k(\nu_\pi) := \frac{1}{1 - \gamma} \sum_{s,a} \overline{r}_k(s, a)\nu_\pi(s, a), \tag{3}$$

where $\overline{r}_k(s, a) = \frac{1}{M} \sum_{m=1}^{M} r_k^{(m)}(s, a)$ denotes the average reward/safety score. However, in the multi-agent setting, $\nu_\pi$ associated with a product policy $\pi$ needs to satisfy the following additional complex constraints. Below, $a^{(\backslash m)}$ denotes the joint action of all the agents except agent $m$.

**Theorem 1.** *The constrained cooperative MARL problem* (1) *is equivalent to the following constrained optimization problem on function $\nu : \mathcal{S} \times \mathcal{A} \to \mathbb{R}$. That is, $\nu$ is the optimal solution to the following problem if and only if $\nu = \nu_\pi$ where $\pi$ is the optimal product policy of the problem* (1).

$$\max_\nu \frac{1}{1 - \gamma} \sum_{s,a} \overline{r}_0(s, a)\nu(s, a) \tag{4}$$

*s.t. (Occupation constraints):*

$$\nu \geq 0, \quad \sum_{s,a} \nu(s, a) = 1, \quad \sum_a \nu(s', a) = (1 - \gamma)\rho(s') + \gamma \sum_{s,a} \nu(s, a)\mathcal{P}(s'|s, a); \; \forall s'$$

*(Product policy constraints):*

$$\nu(s, a) \sum_{a'} \nu(s, a') = \sum_{a'^{(m)}} \nu\big(s, [a'^{(m)}, a^{(\backslash m)}]\big) \cdot \sum_{a'^{(\backslash m)}} \nu\big(s, [a^{(m)}, a'^{(\backslash m)}]\big); \; \forall s, a$$

*(Safety constraints):*

$$\frac{1}{1 - \gamma} \sum_{s,a} \overline{r}_k(s, a)\nu(s, a) \geq \xi_k; \; k = 1, 2, \ldots, K.$$

*Proof Sketch of Theorem 1.* Note that both the objective function and the safety constraints in (1) are rewritten using (3). The occupation constraints are standard for any occupation measure $\nu$. The challenge is to introduce the product policy constraints, which is equivalent to that the corresponding joint policy is a product policy. To do this, we observe that a joint policy $\pi$ is a product policy if and only if $\pi(a|s) = \pi^{(m)}(a^{(m)}|s)\pi^{(\backslash m)}(a^{(\backslash m)}|s)$ for all $m$, and also observe that the occupation measure satisfies $\nu_\pi(s, a) = \nu_\pi(s)\pi^{(m)}(a^{(m)}|s)\pi^{(\backslash m)}(a^{(\backslash m)}|s)$. Based on these two observations, we can show that any occupation measure $\nu_\pi$ is associated with a product policy $\pi$ if and only if $\nu_\pi(s, a) \sum_{a'} \nu_\pi(s, a') = \sum_{a'^{(m)}} \nu_\pi\big(s, [a'^{(m)}, a^{(\backslash m)}]\big) \cdot \sum_{a'^{(\backslash m)}} \nu_\pi\big(s, [a^{(m)}, a'^{(\backslash m)}]\big)$ for all $s, a$. $\quad\square$

Theorem 1 shows that the constrained cooperative MARL problem (1) is equivalent to an optimization problem with quadratic equality constraints, which are induced by the product structure of the joint policy. Unfortunately, optimization problems with both linear and quadratic equality constraints are

highly nonconvex and there is no known polynomial-time algorithm. Moreover, some studies argued that it is probably an NP-complete problem (Murty and Kabadi, 1987). Thus, constrained cooperative MARL is a challenging problem due to the presence of safety and product policy constraints, and we further illustrate this point in the perspective of duality gap in the next section.

As a comparison, both the constrained RL problem (with a single agent) and the cooperative MARL problem (without safety constraints), as special cases of the constrained cooperative MARL problem, can be solved in polynomial-time. To briefly explain, note that the constrained RL problem is equivalent to the problem (4) without the quadratic product policy constraints (not required in the single agent case), and the problem is simply a linear programming problem that can be solved in polynomial time (Altman, 2004). For the cooperative MARL problem, it is equivalent to the problem (4) without the safety constraints. In this case, it is well known that the problem always has an optimal product policy that is both deterministic and greedy, which can be obtained by standard value iteration or policy iteration approaches (Agarwal et al., 2022).

## 3 Duality Gap and Primal-Dual Algorithm

In the existing literature, the mainstream studies proposed to apply the popular primal-dual algorithm to solve constrained cooperative MARL problems (Diddigi et al., 2019; Gu et al., 2021; Lu et al., 2021; Yang et al., 2023; Ying et al., 2023). However, this algorithm converges only when the strong duality holds, which has not been formally justified in the constrained cooperative MARL setting. In this section, we prove that constrained cooperative MARL problems can have strictly positive duality gap, and consequently the primal-dual algorithm does not have exact convergence guarantee.

### 3.1 Constrained Cooperative MARL Has Nonzero Duality Gap

The constrained cooperative MARL problem (1) is equivalent to the following optimization problem.

$$\max_{\pi} \min_{\lambda \in \mathbb{R}_+^K} L(\pi, \lambda) := V_0(\pi) + \sum_{k=1}^{K} \lambda_k \big[ V_k(\pi) - \xi_k \big], \tag{5}$$

where $L(\pi, \lambda)$ denotes the Lagrange function with multiplier $\lambda = [\lambda_1, \ldots, \lambda_K]$. The primal-dual algorithm is based on a key assumption that the following duality gap equals zero.

$$\text{(Duality gap):} \quad \Delta := \min_{\lambda \in \mathbb{R}_+^K} \max_{\pi} L(\pi, \lambda) - \max_{\pi} \min_{\lambda \in \mathbb{R}_+^K} L(\pi, \lambda). \tag{6}$$

In the special case of a single agent, the problem reduces to a constrained RL problem that has been shown to have zero duality gap (Altman, 2004; Paternain et al., 2019). This can be easily seen by rewriting $L(\pi, \lambda) = \widetilde{V}_0(\nu_\pi) + \sum_{k=1}^{K} \lambda_k [\widetilde{V}_k(\nu_\pi) - \xi_k]$ using (3), which reduces to a bilinear function of $(\nu_\pi, \lambda) \in \mathcal{V} \times \mathbb{R}_+^K$. Since both of the sets $\mathcal{V} := \{\nu_\pi | \pi \text{ is a policy}\}$ and $\mathbb{R}_+^K$ are convex sets, zero duality gap follows from the standard minmax theorem (Lemma 9.2 of (Altman, 2004)). However, in constrained cooperative MARL, the set $\mathcal{V}$ changes to $\mathcal{V}_p := \{\nu_\pi | \pi \text{ is a product policy}\}$, which is nonconvex due to the product policy constraints in Theorem 1. Consequently, the duality gap $\Delta$ does not necessarily equal zero, which is formally proved in the following fact.

**Fact 1.** *Constrained cooperative MARL problems can have a strictly positive duality gap.*

**Remark:** Alatur et al. (2023) also obtains a similar result of positive duality gap for constrained Markov potential game with competitive agents. Their result applies to constrained cooperative MARL when all the agents use the same reward function $r_0$. Moreover, it can be easily seen that the duality gap has a constant upper bound $\Delta \leq \frac{1}{1-\gamma}$ as $\overline{r}_{k,t} \in [0, 1]$.

*Proof Sketch of Fact 1.* We construct Example 1 (see Appendix A) and show that it has a positive duality gap $\Delta = \frac{3}{4}$ (see Appendix C for the detailed proof). The reward $r_0^{(m)}$ and safety scores $r_1^{(m)}, r_2^{(m)}$ of this example are carefully selected based on the key observation that $\Delta > 0$ if and only if every optimal joint policy $\widetilde{\pi}^*$ of the constrained cooperative MARL problem (1) is a non-product policy. To elaborate, we show the following equivalent conditions on the Lagrange function.

$$\min_{\lambda \in \mathbb{R}_+^K} \max_{\text{product policy } \pi} L(\pi, \lambda) \overset{(i)}{=} \min_{\lambda \in \mathbb{R}_+^K} \max_{\text{joint policy } \pi} L(\pi, \lambda) \overset{(ii)}{=} \max_{\text{joint policy } \pi} \min_{\lambda \in \mathbb{R}_+^K} L(\pi, \lambda) = V_0(\widetilde{\pi}^*),$$

where (i) holds since $\max_{\text{product policy } \pi} L(\pi, \lambda)$ is essentially a cooperative MARL problem, which has an optimal deterministic policy that also solves $\max_{\text{joint policy } \pi} L(\pi, \lambda)$, and (ii) follows from the strong duality of constrained RL. Hence, $\Delta > 0$ if and only if $V_0(\widetilde{\pi}^*) > \max_{\text{product policy } \pi} \min_{\lambda \in \mathbb{R}_+^K} L(\pi, \lambda) = V_0(\pi^*)$ where $\pi^*$ is an optimal product policy of the constrained cooperative MARL problem (1), which implies that $\widetilde{\pi}^*$ cannot be a product policy. $\qquad\square$

## 3.2 REANALYSIS OF PRIMAL-DUAL ALGORITHM

Based on the positive duality gap result, we are further motivated to reanalyze the convergence guarantee of the primal-dual algorithm for constrained cooperative MARL. Throughout, we adopt the following standard Slater's condition (Paternain et al., 2019; Ding et al., 2020; 2021).

**Assumption 1** (Slater's condition). *There exists a policy $\widetilde{\pi}$ and constants $\delta_k > 0$ such that $V_k(\widetilde{\pi}) \geq \xi_k + \delta_k$ for all $k = 1, \ldots, K$.*

The primal-dual algorithm is a popular method for solving constrained RL type problems. We present the algorithm updates in Algorithm 1, whose main idea is to optimize the Largrange function $L(\pi, \lambda)$ alternatively between $\pi$ and $\lambda$. Specifically, in the primal update step (line 4), we fix $\lambda$ and update the policy $\pi$ by solving the subproblem $\max_{\pi} L(\pi, \lambda)$. In particular, define the surrogate reward $\overline{r}_{\lambda, t} := \overline{r}_{0,t} + \sum_{k=1}^K \lambda_k \overline{r}_{k,t}$ and then the subproblem reduces to a standard cooperative MARL problem with this surrogate reward. One can apply any of the existing MARL algorithms to solve this subproblem up to arbitrary precision $\epsilon_1 > 0$, e.g., decentralized policy gradient (Bai et al., 2021) and decentralized actor-critic (Zhang et al., 2018; Heredia and Mou, 2019; Chen et al., 2020; 2022). Moreover, in the dual update step (line 6), we fix $\pi$ and update $\lambda$ by solving the subproblem $\min_{\lambda} L(\pi, \lambda)$ via projected gradient descent. Note that for the policy evaluation step in line 5, one can apply the existing decentralized TD learning algorithms (Sun et al., 2020; Chen et al., 2021c).

We obtain the following new convergence result of Algorithm 1 in constrained cooperative MARL.

**Theorem 2.** *Consider a constrained cooperative MARL problem with duality gap $\Delta$, and let Assumption 1 hold. Apply the primal-dual Algorithm 1 to solve it with hyperparameters $\lambda_{k,\max} = \frac{2}{\delta_k(1-\gamma)} + \frac{2\Delta}{\delta_k}$, $\epsilon_1 = \frac{1}{1-\gamma}\sqrt{\frac{K}{2T}\sum_{k=1}^K \lambda_{k,\max}^2}$, $\epsilon_2 = \frac{1}{1-\gamma}\sqrt{\frac{\sum_{k=1}^K \lambda_{k,\max}^2}{2T(\sum_{k=1}^K \lambda_{k,\max})^2}}$, $\beta = (1-\gamma)\sqrt{\frac{1}{2KT}\sum_{k=1}^K \lambda_{k,\max}^2}$. We obtain the following results on optimality gap and constraint violation $((\cdot)_+ := \max(\cdot, 0))$.*

$$V_0(\pi^*) - \mathbb{E}_{\widetilde{T}}[V_0(\pi_{\widetilde{T}})] \leq \frac{7}{1-\gamma}\sqrt{\frac{K}{2T}\sum_{k=1}^K \lambda_{k,\max}^2}, \qquad (7)$$

$$\sum_{k=1}^K \lambda_{k,\max}\mathbb{E}_{\widetilde{T}}\big(\xi_k - V_k(\pi_{\widetilde{T}})\big)_+ \leq \frac{22}{1-\gamma}\sqrt{\frac{K}{2T}\sum_{k=1}^K \lambda_{k,\max}^2} + 2\Delta. \qquad (8)$$

*Furthermore, using the decentralized natural actor-critic algorithm (Chen et al., 2022) to obtain $\pi_t$ and model-based policy evaluation (Li et al., 2020) to obtain $\widehat{V}_k(\pi_t)$, the sample complexity is $\mathcal{O}(\epsilon^{-5}\ln \epsilon^{-1})$ to achieve $V_0(\pi^*) - \mathbb{E}_{\widetilde{T}}[V_0(\pi_{\widetilde{T}})] \leq \epsilon$ and $\sum_{k=1}^K \lambda_{k,\max}\mathbb{E}_{\widetilde{T}}\big(\xi_k - V_k(\pi_{\widetilde{T}})\big)_+ \leq \epsilon + 2\Delta$.*

Theorem 2 shows that in constrained cooperative MARL, the optimality gap $V_0(\pi^*) - \mathbb{E}[V_0(\pi_{\widetilde{T}})]$ of the primal-dual algorithm achieves a sub-linear convergence rate $\mathcal{O}(1/\sqrt{T})$. Moreover, the constraint violation $\sum_{k=1}^K \lambda_{k,\max}\mathbb{E}_{\widetilde{T}}\big(\xi_k - V_k(\pi_t)\big)_+$ converges at a similar rate, but up to a convergence error that depends on the duality gap $\Delta$ of the problem. Therefore, it is possible that the algorithm converges to a sub-optimal policy that strictly violates the safety constraints.

**Comparison with the existing art**. We note that the above sub-linear convergence rates match those of primal-dual algorithm in single-agent constrained RL ($\Delta = 0$) (Ding et al., 2020; 2021). Moreover, compared with the existing studies of the primal-dual algorithm for constrained cooperative MARL that only establish convergence to stationary points (Lu et al., 2021; Ying et al., 2023), our Theorem 2 directly characterizes the optimality and constraint violation of the output policy $\pi_{\widetilde{T}}$. To the best of our knowledge, this is the first convergence result of the primal-dual algorithm in constrained cooperative MARL that characterizes the impact of the nonzero duality gap $\Delta$.

**Proof logic.** The proof logic mainly follows that of primal-dual algorithm in constrained RL (Ding et al., 2020). However, since the duality gap $\Delta > 0$, we need to adopt a different bound for any product policy $\pi'$, i.e., $L(\pi', \lambda^*) \leq \max_\pi L(\pi, \lambda^*) = V_0(\pi^*) - \Delta$, where $\lambda^* \in \arg\min_{\lambda \in \mathbb{R}_+^K} \max_\pi L(\pi, \lambda)$, and $\pi^*$ is the optimal product policy of the constrained cooperative MARL problem (1). The above bound is used to bound the constraint violation of the policies $\pi' = \pi_t$ obtained by the primal-dual algorithm, and also bound $\lambda^*$ via $\pi' = \widetilde{\pi}$ in Assumption 1 (See Lemma 1 in Appendix I for

---

**Algorithm 1** Primal-Dual Algorithm
1: **Inputs:** $\epsilon_1, \epsilon_2, \beta > 0$, $\lambda_{k,\max} > 0$ for $k = 1, \ldots, K$,
2: **Initialize:** $\lambda_{k,0} = 0$ for $k = 1, \ldots, K$.
3: **for** iterations $t = 0, 1, 2, \ldots, T - 1$ **do**
4: $\quad$ Solve the cooperative MARL problem with surrogate reward $\overline{r}_{\lambda,t}$. Obtain an $\epsilon_1$-accurate solution $\pi_t$, i.e.,

$$\max_\pi L(\pi, \lambda_t) - L(\pi_t, \lambda_t) \leq \epsilon_1. \quad (9)$$

5: $\quad$ Perform TD learning to estimate $\widehat{V}_k(\pi_t)$ such that $|\widehat{V}_k(\pi_t) - V_k(\pi_t)| \leq \epsilon_2$.
6: $\quad$ Update the multipliers for $k = 1, 2, \ldots, K$ using projected gradient descent as follows.

$$\lambda_{t+1,k} = \text{Proj}_{[0,\lambda_{k,\max}]}\big[\lambda_{t,k} - \beta\big(\widehat{V}_k(\pi_t) - \xi_k\big)\big]. \quad (10)$$

7: **end for**
8: **Output:** $\pi_{\widetilde{T}}$ with $\widetilde{T} \overset{\text{uniform}}{\sim} \{0, 1, \ldots, T - 1\}$.

---

detail). The duality gap $\Delta$ in the above bound further affects the subsequent proof.

## 4 DECENTRALIZED PRIMAL ALGORITHM

In this section, we propose a primal-based algorithm for constrained cooperative MARL whose convergence does not involve the duality gap. Our algorithm extends the centralized CRPO algorithm (Xu et al., 2021) to the constrained cooperative setting, and involves new designs to enable decentralized implementation and new proof techniques that lead to improved convergence rates.

Our decentralized primal algorithm is presented in Algorithm 2. To explain, the main idea is to use (decentralized) TD learning to estimate the value functions $\{V_k(\pi_t)\}_{k=1}^K$ associated with the safety scores and select one that violates its constraint threshold by a pre-determined amount $\eta$ as the target value function. If no such violation exists, then we select $V_0$ as the target value function. After that, we update the current policy $\pi_t$ using a decentralized natural policy gradient algorithm based on the selected target value function. Compared to the existing CRPO algorithm for single-agent constrained RL (Xu et al., 2021), our algorithm design introduces several new elements. To elaborate, we update the agents' product policies via the following decentralized natural policy gradient (NPG) update

$$\pi_{t+1}^{(m)}(a^{(m)}|s) \propto \pi_t^{(m)}(a^{(m)}|s) \exp\big(\alpha \widehat{Q}_{k_t}^{(m)}(\pi_t; s, a^{(m)})\big); \quad \forall s, a^{(m)}, \quad (11)$$

where $\alpha > 0$ is the stepsize and $\widehat{Q}_k^{(m)}(\pi; s, a^{(m)})$ is an estimation of the local $Q$ function $Q_k^{(m)}(\pi; s, a^{(m)}) = \mathbb{E}_\pi\big[\sum_{t=0}^\infty \gamma^t \overline{r}_{k,t}\big|s_0 = s, a_0^{(m)} = a^{(m)}\big]$, which can be efficiently estimated by sample average estimation of $Q_k^{(m)}(\pi; s, a^{(m)}) = \mathbb{E}\big[\overline{r}_k(s, a) + \gamma V_k(\pi; s')|a^{(\backslash m)} \sim \pi^{(\backslash m)}(\cdot|s), s' \sim \mathcal{P}(\cdot|s, a)\big]$ (Wei et al., 2021; Chen et al., 2021b). In particular, such a decentralized update is crucial for performing optimization in the product policy space. Moreover, when we estimate the value functions $\{V_k(\pi_t)\}_{k=1}^K$ in line 5, we randomly permute their order and break the loop once a target value function is found. This helps avoid the undesirable situation where the same value function is frequently selected so that the policy stays at a stationary point (possibly infeasible) in the policy update (11), and also reduces computation. As a comparison, the CRPO algorithm requires to estimate $V_k(\pi_t)$ for all $k = 1, 2, \ldots, K$ in every iteration, and therefore is less efficient.

Next, define the advantage gap $\zeta_k := \sup_{s,a,\pi} \big|A_k(\pi; s, a) - \sum_{m=1}^M A_k^{(m)}(\pi; s, a^{(m)})\big|$, which corresponds to the gap between the local advantage function $A_k^{(m)}(\pi; s, a^{(m)}) := Q_k^{(m)}(\pi; s, a^{(m)}) - V_k(\pi; s)$ and the global advantage function $A_k(\pi; s, a) := Q_k(\pi; s, a) - V_k(\pi; s)$.

**Theorem 3.** *Apply Algorithm 2 with* $\alpha = \mathcal{O}\big(\sqrt{\frac{(1-\gamma)^3}{MT}}\big)$, $\epsilon_2 = \mathcal{O}\big(\sqrt{\frac{M}{T(1-\gamma)^5}}\big)$, $\epsilon_3 = \mathcal{O}\big(\sqrt{\frac{1-\gamma}{MT}}\big)$,
$\eta = \mathcal{O}\big(\sqrt{\frac{M}{T(1-\gamma)^5}} + \frac{\max_{1 \leq k \leq K} \zeta_k}{(1-\gamma)^2}\big)$ *(see Appendix E for details). Then the output policy* $\pi_{\widetilde{T}}$ *satisfies*

$$V_0(\pi^*) - \mathbb{E}_{\widetilde{T}}[V_0(\pi_{\widetilde{T}})] \leq \mathcal{O}\Big(\sqrt{\frac{M}{T(1-\gamma)^5}} + \frac{\zeta_0}{(1-\gamma)^2}\Big), \tag{12}$$

$$\xi_k - \mathbb{E}_{\widetilde{T}}[V_k(\pi_{\widetilde{T}})] \leq \mathcal{O}\Big(\sqrt{\frac{M}{T(1-\gamma)^5}} + \frac{\max_{1 \leq k \leq K} \zeta_k}{(1-\gamma)^2}\Big); \; k = 1, \ldots, K. \tag{13}$$

*Furthermore, using the model-based policy evaluation (Li et al., 2020) to obtain* $\widehat{V}_k(\pi_t)$ *and*
$\widehat{Q}_{k_t}^{(m)}(\pi_t; s, a^{(m)})$, *the sample complexity is* $\mathcal{O}(\epsilon^{-4})$ *to achieve* $V_0(\pi^*) - \mathbb{E}_{\widetilde{T}}[V_0(\pi_{\widetilde{T}})] \leq \mathcal{O}\big(\epsilon + \frac{\zeta_0}{(1-\gamma)^2}\big)$ *and* $\sum_{k=1}^K \lambda_{k,\max} \mathbb{E}_{\widetilde{T}}\big(\xi_k - V_k(\pi_{\widetilde{T}})\big)_+ \leq \mathcal{O}\big(\epsilon + \frac{\max_{1 \leq k \leq K} \zeta_k}{(1-\gamma)^2}\big)$.

Theorem 3 shows that both the optimality gap and the constraint violation converge at the sublinear rate $\mathcal{O}\big(\sqrt{M/[T(1-\gamma)^5]}\big)$, up to certain convergence errors that depend on the advantage gaps $\zeta_k$. Thus, the above convergence result has a very different nature from that of the primal-dual algorithm, which involves the problem's duality gap $\Delta$ instead. Moreover, $\zeta_k$ vanishes if and only if the $Q$ function satisfies a certain factorization structure (See Appendix H for more details) (Guestrin et al., 2001; Son et al., 2019; Rashid et al., 2020). Therefore, when the Q function can be approximated by a factorized form, $\zeta_k$ is small, so the primal algorithm is preferable to the primal-dual algorithm.

**Comparison with the existing art.** In the single-agent case $M = 1$, the advantage gap $\zeta_k$ vanishes, and the convergence rates in Theorem 3 reduce to $\mathcal{O}\big(\sqrt{M/[T(1-\gamma)^5]}\big)$, which strictly improves that of the CRPO algorithm (Xu et al., 2021) by a factor of $\sqrt{|\mathcal{S}||\mathcal{A}|(1-\gamma)}$ for large state and action spaces [1]. In particular, this improvement crucially relies on proving our new Lemma 3, which proves the bound $V_{k_t}(\pi_{t+1}; \rho') - V_{k_t}(\pi_t; \rho') \leq \frac{M\alpha}{(1-\gamma)^3} + \frac{2M\alpha\epsilon_3}{(1-\gamma)^2}$ that tightens the corresponding bound in (Xu et al., 2021) by a factor of $\mathcal{O}(1/[|\mathcal{S}||\mathcal{A}|(1-\gamma)])$ using two novel techniques as elaborated below.

**Technical novelty.** First, denote $p_i, p_i'$ as the distributions of state $s_i$ under $\pi_t$ and $\pi_{t+1}$, respectively. Then, by Markov decision process, we can show that $\|p_{i+1}' - p_{i+1}\|_1 \leq \max_s \|\pi_{t+1}(\cdot|s) - \pi_t(\cdot|s)\|_1 + \|p_i' - p_i\|_1$, which implies that $\|p_i' - p_i\|_1 \leq i \max_s \|\pi_{t+1}(\cdot|s) - \pi_t(\cdot|s)\|_1$. Hence, we have
$$V_{k_t}(\pi_{t+1}; \rho') - V_{k_t}(\pi_t; \rho') = \sum_{i=0}^{\infty} \gamma^i \sum_{s,a} \overline{r}_{k_t}(s, a)[p_i'(s)\pi_{t+1}(a|s) - p_i(s)\pi_t(a|s)]$$
$$\leq (1-\gamma)^{-2} \max_s \|\pi_{t+1}(\cdot|s) - \pi_t(\cdot|s)\|_1,$$
where the second inequality upper bounds $\sum_s$ by $\max_s$ without introducing the factor $|\mathcal{S}|$. Second, we further prove the following non-trivial tight bound
$$\|\pi_{t+1}(\cdot|s) - \pi_t(\cdot|s)\|_1 \leq \sum_{m=1}^M \sum_{a^{(m)}} |\pi_{t+1}^{(m)}(a^{(m)}|s) - \pi_t^{(m)}(a^{(m)}|s)|$$
$$\leq \sum_{m=1}^M \alpha\big(\max_{a^{(m)}} \widehat{Q}_{k_t}(\pi_t; s, a^{(m)}) - \min_{a^{(m)}} \widehat{Q}_{k_t}(\pi_t; s, a^{(m)})\big),$$
where the inequality is obtained by taking $\pi_{t+1}^{(m)}(a^{(m)}|s)$ in the update rule (11) as a function of $\alpha$ and bounding $\big|\frac{d}{d\alpha}\pi_{t+1}^{(m)}(a^{(m)}|s)\big|$ (see the proof of Lemma 2 for details). This bound upper bounds $\sum_{a^{(m)}}$ by $\max_{a^{(m)}} \widehat{Q}_{k_t}(\pi_t; s, a^{(m)}) - \min_{a^{(m)}} \widehat{Q}_{k_t}(\pi_t; s, a^{(m)})$. In contrast, (Xu et al., 2021) uses the Lipschitz property $V_{k_t}(\pi_{t+1}; \rho') - V_{k_t}(\pi_t; \rho') \leq \frac{2}{1-\gamma}\|w_{t+1} - w_t\|_2$ under the softmax policy parameterization $\pi_t(a|s) \propto \exp[w_t(s, a)]$. However, this further leads to the upper bound $\|w_{t+1} - w_t\|_2 = \alpha\|\widehat{Q}_{k_t}(\pi_t; \cdot, \cdot)\|_2 \leq \frac{\alpha}{1-\gamma}|\mathcal{S}||\mathcal{A}|$ that is much looser than our $\|\pi_{t+1} - \pi_t\|_1$.

## 5 PRIMAL-DUAL ALGORITHM V.S. PRIMAL ALGORITHM

We have shown that the primal-dual Algorithm 1 and the primal Algorithm 2 suffer from non-vanishing convergence errors that depend on the duality gap and the advantage gap, respectively. Next, we show that each of the two algorithms can offer advantages over the other in certain scenarios.

---

[1] The convergence rates of CRPO established in (Xu et al., 2021) should be $\mathcal{O}\big(\frac{1}{(1-\gamma)^2}\sqrt{\frac{|\mathcal{S}||\mathcal{A}|}{T}}\big)$. In the proof of their Lemma 7, (iii) should have used the update rule $w_{t+1} - w_t = \frac{\alpha}{1-\gamma}\overline{Q}_t^i$, but they used $w_{t+1} - w_t = \alpha\overline{Q}_t^i$.

First, we revisit Example 1 and show that Algorithm 2 outperforms Algorithm 1 in the following theorem. Here, the product policy $\pi_t$ is fully characterized by $p_t := \pi_t^{(1)}(0|s)$ and $q_t := \pi_t^{(2)}(0|s)$.

**Theorem 4.** *In Example 1, if we run Algorithm 1 with $\epsilon_1 = \epsilon_2 = 0$, then the generated policy $\pi_t$ is infeasible for all $t$. In contrast, if we run Algorithm 2 with $\epsilon_2 = \epsilon_3 = 0$, $\alpha \leq 10^{-3}$, $\eta = -6\alpha$ and an initial policy that satisfies $\frac{2}{3}q_0 \leq p_0 \leq \frac{3}{2}q_0$ and $0.06 \leq p_0q_0 \leq 0.135$, then the generated policy $\pi_t$ for all $t \geq \frac{13}{\alpha}\ln\left(\frac{1}{20\alpha}\right)$ is feasible and close to the optimal solution $\left(\frac{1}{4}, \frac{1}{4}\right)$ with $\max\left(\left|p_t - \frac{1}{4}\right|, \left|q_t - \frac{1}{4}\right|\right) \leq 14\alpha$.*

**Technical novelty:** The major challenge to prove Theorem 4 lies in the convergence analysis of Algorithm 2 in Example 1, which can be written as a nonconcave constrained maximization problem

---

**Algorithm 2** Decentralized Primal Algorithm

1: **Inputs:** $\alpha, \epsilon_2, \epsilon_3 > 0, \eta$
2: **Initialize:** Policy $\pi_0$.
3: **for** primal iterations $t = 0, 1, 2, \ldots, T-1$ **do**
4:     ▶ Let $k_t \leftarrow 0$.
5:     **for** $k = \sigma_t(1), \ldots, \sigma_t(K)$ where $\sigma_t$ is a random permutation on $\{1, 2, \ldots, K\}$ **do**
6:         ▶ Perform TD learning to estimate $\widehat{V}_k(\pi_t)$ such that $|\widehat{V}_k(\pi_t) - V_k(\pi_t)| \leq \epsilon_2$.
7:         ▶ If $\widehat{V}_k(\pi_t) < \xi_k - \eta$, let $k_t \leftarrow k$ and break.
8:     **end for**
9:     **for** agents $m = 1, 2, \ldots, M$ in parallel **do**
10:         ▶ Estimate $\widehat{Q}_{k_t}^{(m)}(\pi_t; s, a^{(m)})$ such that $|\widehat{Q}_{k_t}^{(m)}(\pi_t; s, a^{(m)}) - Q_{k_t}^{(m)}(\pi_t; s, a^{(m)})| \leq \epsilon_3$.
11:         ▶ Update local policy to $\pi_{t+1}^{(m)}$ following the decentralized NPG update rule (11).
12:     **end for**
13: **end for**
14: **Output:** $\pi_{\widetilde{T}}$ with $\widetilde{T} \overset{\text{uniform}}{\sim} \{0 \leq t \leq T-1 : k_t = 0\}$.

---

(37). Moreover, the primal update rule differs for $k_t = 0, 1, 2$. Hence, we cannot follow the standard convergence analysis for convex optimization. Instead, we utilize the multiplicative structure of the primal updates of $p_t$ and $q_t$ in Eqs. (42) and (43) to obtain the convergence of the potential functions $p_tq_t$ and $\frac{p_t}{q_t}$ to $\frac{1}{16}$ and 1, respectively. To elaborate, we prove the statement $(A_t)$: $\frac{2}{3}q_t \leq p_t \leq \frac{3}{2}q_t$ and $0.06 \leq p_tq_t \leq 0.135$, and the statement $(C_t)$: $\left|\frac{p_{t+1}}{q_{t+1}} - 1\right| \leq (1 - 0.079\alpha)\left|\frac{p_t}{q_t} - 1\right|$ whenever $\left|\frac{p_t}{q_t} - 1\right| > 5\alpha$, via inductions that $(A_t), (C_t) \Rightarrow (A_{t+1})$ and that $(A_t) \Rightarrow (C_t)$. In particular, $(A_t) \Rightarrow (C_t)$ is proved in 4 separate cases: either $p_t \geq q_t$ or $p_t < q_t$, and either $p_tq_t \geq \frac{1}{16} + 3\alpha$ or $p_tq_t < \frac{1}{16} + 3\alpha$. $(A_t), (C_t)$ imply that $\left|\frac{p_T}{q_T} - 1\right| \leq 10\alpha$ for a certain $T \leq \mathcal{O}\left(\alpha^{-1}\ln(\alpha^{-1})\right)$. To further show that $\left|\frac{p_t}{q_t} - 1\right| \leq 10\alpha; \forall t \geq T$, it suffices to prove $\left|\frac{p_{t+1}}{q_{t+1}} - \frac{p_t}{q_t}\right| \leq 4.66\alpha$, so that the ring area $5\alpha < \left|\frac{p_t}{q_t} - 1\right| \leq 10\alpha$ is sufficiently wide to drag $\frac{p_t}{q_t}$ back towards 1. The convergence rate of $p_tq_t$ is proved similarly via inductions in two separate cases where $p_tq_t \geq \frac{1}{16} + 3\alpha$ or $p_tq_t < \frac{1}{16} + 3\alpha$.

Next, we prove that Algorithm 1 outperforms Algorithm 2 in Example 2 (See Appendix A).

**Theorem 5.** *In Example 2, Algorithm 1 obtains the optimal policy in one iteration. In contrast, if we run Algorithm 2 with $\epsilon_2 = \epsilon_3 = \eta = 0$ and an initial policy that satisfies $p_0 + q_0 = 1$, then the generated policy $\pi_t$ is infeasible for all $t$.*

Since Example 2 is also a nonconcave maximization problem, proving the infeasibility of the function value $V_1(\pi_t)$ obtained by the primal algorithm also cannot follow the standard convex optimization convergence analysis. Instead, we prove that $p_t + q_t = 1$ via induction and show the constraint violation $V_1(\pi_t) = 4p_t(1 - p_t) \leq 1 < \xi_1$.

In Appendix A, we conduct simulations to verify the above theoretical comparison of both algorithms.

# 6   CONCLUSION

In this work, we have shown that constrained cooperative MARL is a highly nonconvex problem that is more challenging than cooperative MARL and single-agent constrained RL in the occupation measure space. Due to the challenges, the strong duality condition required by the mainstream primal-dual algorithms no longer holds in constrained cooperative MARL. Therefore, we reanalyze the convergence rates of the primal-dual algorithms with nonzero duality gap. Then, we propose a decentralized primal algorithm for constrained cooperative MARL to avoid the duality gap, and our analysis shows that its convergence is hindered by another gap induced by the advantage functions. We expect that our study will spark new research directions in multi-agent RL, and motivate to design better algorithms with rigorous convergence guarantee for constrained cooperative MARL.

ACKNOWLEDGMENTS

The work of Ziyi Chen at Utah and Yi Zhou was supported in part by U.S. National Science Foundation under the Grants CCF-2106216, DMS-2134223 and ECCS-2237830 (CAREER). Ziyi Chen at UMD and Heng Huang were partially supported by NSF IIS 2347592, 2347604, 2348159, 2348169, DBI 2405416, CCF 2348306, CNS 2347617.

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

# Appendix

## Table of Contents

## A  NUMERIC EXAMPLES AND EXPERIMENTS

In this section, we implement the primal-dual algorithm (Algorithm 1) and the primal algorithm (Algorithm 2) to the following two numeric examples to verify Theorems 4 and 5.

**Example 1.** *Consider a constrained cooperative MARL problem with two agents, a single state $\mathcal{S} = \{s\}$. Both agents share the same action space $\mathcal{A}^{(m)} = \{0, 1\}$ and the same reward and safety scores listed below. The discount factor is $\gamma = \frac{1}{2}$ and the safety thresholds are $\xi_1 = \xi_2 = \frac{1}{8}$.*

$$r_0^{(m)}(s, [0, 0]) = 1, \quad r_1^{(m)}(s, [0, 0]) = 1, \quad r_2^{(m)}(s, [0, 0]) = 0$$
$$r_0^{(m)}(s, [0, 1]) = 0, \quad r_1^{(m)}(s, [0, 1]) = 0, \quad r_2^{(m)}(s, [0, 1]) = 0$$
$$r_0^{(m)}(s, [1, 0]) = 0, \quad r_1^{(m)}(s, [1, 0]) = 0, \quad r_2^{(m)}(s, [1, 0]) = 0$$
$$r_0^{(m)}(s, [1, 1]) = 1, \quad r_1^{(m)}(s, [1, 1]) = 0, \quad r_2^{(m)}(s, [1, 1]) = 1$$

**Example 2.** *Consider modifying Example 1 so that both agents share the following reward and a single safety score. The safety threshold is $\xi_1 = 1.8$.*

$$r_0^{(m)}(s, [0, 0]) = 1, \quad r_1^{(m)}(s, [0, 0]) = 1$$
$$r_0^{(m)}(s, [0, 1]) = 0, \quad r_1^{(m)}(s, [0, 1]) = 0$$
$$r_0^{(m)}(s, [1, 0]) = 0, \quad r_1^{(m)}(s, [1, 0]) = 0$$
$$r_0^{(m)}(s, [1, 1]) = 0, \quad r_1^{(m)}(s, [1, 1]) = 1.$$

For Example 1, we implement the primal Algorithm 2 with $\alpha = 10^{-3}$, $\epsilon_2 = \epsilon_3 = 0$, $\eta = -6\alpha$, and try various initial policies $(p_0, q_0) \in \{(0.45, 0.3), (0.2, 0.3), (0.3, 0.3), (0.25, 0.25), (0.35, 0.35)\}$ which satisfy the conditions of Theorem 4. We obtain the results as shown in the first five figures of

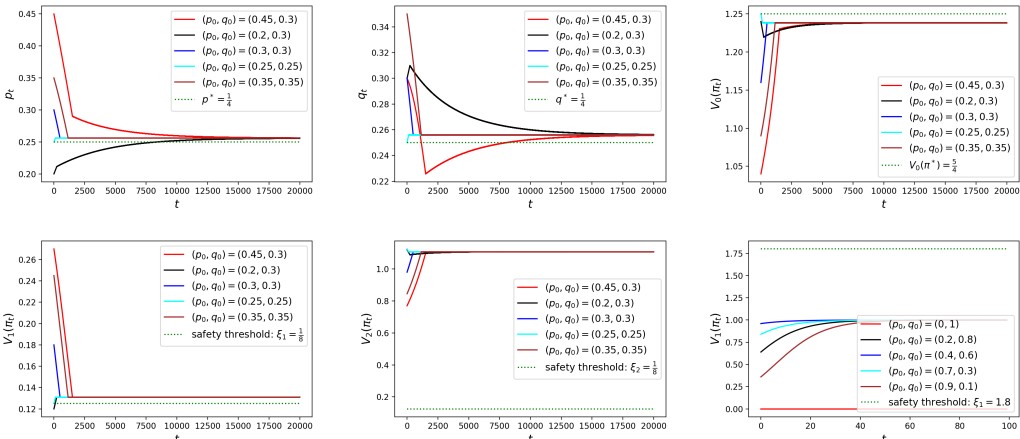

Figure 1: Results of the primal algorithm on Examples 1 (the first 5 figures) and 2 (the last figure).

Figure 1. The first two figures at the top of Figure 1 indicate that $(p_t, q_t)$ with various initializations converge to the same value which is close to the optimal solution $(\frac{1}{4}, \frac{1}{4})$. The top right figure of Figure 1 shows that $V_0(\pi_t)$ converges and is close to the optimal value $\frac{5}{4}$. The first two figures at the bottom of Figure 1 show that the policy $\pi_t$ is feasible (i.e., $V_1(\pi_t) \geq \xi_1, V_2(\pi_t) \geq \xi_2$) after $t \geq 2500$ iterations. In contrast, we implement the primal-dual Algorithm 1 with $\alpha = \beta = 0.1$, $\epsilon_1 = \epsilon_2 = 0$ and initial multiplier $\lambda = [0, 0]$. The policy parameter $(p_t, q_t)$ alternates between $(1, 1)$ and $(0, 0)$, both of which are infeasible since they satisfy $V_2(\pi_t) = 0 < \xi_2$ and $V_1(\pi_t) = 0 < \xi_1$ respectively. These results verify Theorem 4.

For Example 2, we implement the primal Algorithm 2 with $\alpha = 0.1$, $\epsilon_2 = \epsilon_3 = \eta = 0$, and try various initial policies $(p_0, q_0) \in \{(0, 1), (0.2, 0.8), (0.4, 0.6), (0.7, 0.3), (0.9, 0.1)\}$ which satisfy the conditions of Theorem 5. The learning curve of the value function $V_1(\pi_t)$ is shown in the last figure of Figure 1. It can be seen that $V_1(\pi_t)$ is always far below the safety threshold $\xi_1 = 1.8$. In contrast, implementing the primal-dual Algorithm 1 with $\alpha = \beta = 0.1$, $\epsilon_1 = \epsilon_2 = 0$ and initial multiplier $\lambda = 0$, we obtain $(p_t, q_t) \equiv (1, 1)$ which is the optimal solution to Example 2. These results verify Theorem 5.

## B  PROOF OF THEOREM 1

**Proof for the product policy constraints:** We will first prove that $\pi$ is a product policy if and only if $\nu_\pi$ satisfies the product policy constraints in Eq. (4).

Note that the following equality always holds for $\nu_\pi$ of any joint policy $\pi$.

$$
\nu_\pi(s, a) \sum_{a'} \nu_\pi(s, a') - \sum_{a'^{(m)}} \nu_\pi\big(s, [a'^{(m)}, a^{(\backslash m)}]\big) \cdot \sum_{a'^{(\backslash m)}} \nu_\pi\big(s, [a^{(m)}, a'^{(\backslash m)}]\big)
$$

$$
\overset{(i)}{=} \nu_\pi(s)\pi(a|s) \sum_{a'} \nu_\pi(s)\pi(a'|s) - \Big[\sum_{a'^{(m)}} \nu_\pi(s)\pi([a'^{(m)}, a^{(\backslash m)}]|s)\Big]
$$

$$
\Big[\sum_{a'^{(\backslash m)}} \nu_\pi(s)\pi([a^{(m)}, a'^{(\backslash m)}]|s)\Big]
$$

$$
\overset{(ii)}{=} \nu_\pi^2(s)\big[\pi(a|s) - \pi^{(\backslash m)}(a^{(\backslash m)}|s)\pi^{(m)}(a^{(m)}|s)\big], \tag{14}
$$

where (i) uses $\nu_\pi(s, a) = \nu_\pi(s)\pi(a|s)$, and (ii) uses $\pi^{(m)}(a^{(m)}|s) := \sum_{a'^{(m)}} \nu_\pi(s)\pi([a'^{(m)}, a^{(\backslash m)}]|s)$ and $\pi^{(\backslash m)}(a^{(\backslash m)}|s) := \sum_{a'^{(m)}} \pi([a'^{(m)}, a^{(\backslash m)}]|s)$.

If $\pi$ is a product policy, then $\pi(a|s) = \pi^{(\backslash m)}(a^{(\backslash m)}|s)\pi^{(m)}(a^{(m)}|s)$ where $\pi^{(\backslash m)}(a^{(\backslash m)}|s) = \prod_{m'=1, m'\neq m}^{M} \pi^{(m')}(a^{(m')}|s)$, which implies that Eq. (14) equals 0, i.e., $\nu_\pi$ satisfies the product policy constraints in Eq. (4).

Conversely, suppose that $\nu_\pi$ satisfies the product policy constraints in Eq. (4), i.e., Eq. (14) equals 0. Then for any state $s$, consider the following two cases.

If $\nu_\pi(s) \neq 0$, we have $\pi(a|s) = \pi^{(\backslash m)}(a^{(\backslash m)}|s)\pi^{(m)}(a^{(m)}|s)$, which means for any agent $m$, $a^{(\backslash m)}$ and $a^{(m)}$ are independent given $s$. Therefore, $a^{(1)}, a^{(2)}, \ldots, a^{(M)}$ are independent under the policy $\pi(\cdot|s)$, which means $\pi(a|s) = \prod_{m=1}^M \pi^{(m)}(a^{(m)}|s)$.

If $\nu_\pi(s) = 0$, $\pi(\cdot|s)$ can be arbitrarily defined, and thus we can define it such that the product policy condition $\pi(a|s) = \prod_{m=1}^M \pi^{(m)}(a^{(m)}|s)$ holds.

Therefore, $\pi$ can be a product policy if $\nu_\pi$ satisfies the product policy constraints.

**Proof of equivalence between the problems** (1) **and** (4)**:** Suppose $\pi^*$ is the optimal product policy for the constrained cooperative MARL problem (1). Then $\nu_{\pi^*}$ satisfies the occupation constraints in problem (4) based on Theorem 3.2 of (Altman, 2004), and further satisfies the product policy constraints as $\pi^*$ is a product policy. Therefore $\nu_{\pi^*}$ is a feasible point of the problem (4).

Then consider any function $\nu' : \mathcal{S} \times \mathcal{A} \to \mathbb{R}$ that satisfies all the constraints of the problem (4). Since $\nu'$ satisfies the occupation constraints, $\nu' = \nu_{\pi'}$ for some policy $\pi'$ based on Theorem 3.2 of (Altman, 2004). As proved above, since $\nu_{\pi'}$ satisfies the product policy constraints, $\pi'$ is a product policy. Also, the safety constraint $V_k(\pi') \stackrel{(i)}{=} \frac{1}{1-\gamma}\sum_{s,a}\bar{r}_k(s,a)\nu_{\pi'}(s,a) \geq \xi_k$ $(k = 1, \ldots, K)$ holds where (i) uses Eq. (2). Therefore, $\pi'$ is a feasible policy of the problem (1) and thus we have $V_0(\pi') \leq V_0(\pi^*)$, i.e., $\frac{1}{1-\gamma}\sum_{s,a}\bar{r}_0(s,a)\nu'(s,a) \leq \frac{1}{1-\gamma}\sum_{s,a}\bar{r}_0(s,a)\nu_{\pi^*}(s,a)$. Since $\nu'$ is an arbitrary feasible point of the problem (4), the feasible point $\nu_{\pi^*}$ is also the optimal solution to the problem (4).

Conversely, suppose $\nu^* : \mathcal{S} \times \mathcal{A} \to \mathbb{R}$ is the optimal solution to the problem (4). Then as $\nu^*$ satisfies the occupation constraints and product policy constraints of the problem (4), $\nu^* = \nu_{\pi^*}$ for some product policy $\pi^*$. Hence, the safety constraint $V_k(\pi^*) = \frac{1}{1-\gamma}\sum_{s,a}\bar{r}_k(s,a)\nu_{\pi^*}(s,a)$ $(k = 1, \ldots, K)$ means $\pi^*$ is a feasible product policy of the problem (1).

For any feasible product policy $\pi'$ of the problem (1), $\nu_{\pi'}$ satisfies the occupation constraints and product policy constraints, as well as the safety constraints that $\frac{1}{1-\gamma}\sum_{s,a}\bar{r}_k(s,a)\nu_{\pi'}(s,a) = V_k(\pi') \geq \xi_k$ $(k = 1, \ldots, K)$. Due to the optimality of $\nu^* = \nu_{\pi^*}$, we have $\frac{1}{1-\gamma}\sum_{s,a}\bar{r}_0(s,a)\nu_{\pi'}(s,a) \leq \frac{1}{1-\gamma}\sum_{s,a}\bar{r}_0(s,a)\nu_{\pi^*}(s,a)$, i.e. $V_0(\pi') \leq V_0(\pi^*)$. Hence, the feasible policy $\pi^*$ is also the optimal solution to the problem (1).

## C    PROOF OF FACT 1

We repeat Example 1 as follows.

**Example 1.** *Consider a constrained cooperative MARL problem with two agents, a single state $\mathcal{S} = \{s\}$. Both agents share the same action space $\mathcal{A}^{(m)} = \{0, 1\}$ and the same reward and safety scores listed below. The discount factor is $\gamma = \frac{1}{2}$ and the safety thresholds are $\xi_1 = \xi_2 = \frac{1}{8}$.*

$$r_0^{(m)}(s, [0,0]) = 1, \quad r_1^{(m)}(s, [0,0]) = 1, \quad r_2^{(m)}(s, [0,0]) = 0$$
$$r_0^{(m)}(s, [0,1]) = 0, \quad r_1^{(m)}(s, [0,1]) = 0, \quad r_2^{(m)}(s, [0,1]) = 0$$
$$r_0^{(m)}(s, [1,0]) = 0, \quad r_1^{(m)}(s, [1,0]) = 0, \quad r_2^{(m)}(s, [1,0]) = 0$$
$$r_0^{(m)}(s, [1,1]) = 1, \quad r_1^{(m)}(s, [1,1]) = 0, \quad r_2^{(m)}(s, [1,1]) = 1$$

In the above example, any product policy $\pi(a|s) = \pi^{(1)}(a^{(1)}|s)\pi^{(2)}(a^{(2)}|s)$ can be fully characterized by $p = \pi^{(1)}(0|s)$ and $q = \pi^{(2)}(0|s)$. Then the aim of the constrained cooperative MARL problem in Example 1 can be formulated as

$$
\begin{cases}
\max_{p,q \in [0,1]} V_0(\pi) := 2pq + 2(1-p)(1-q) \\
\text{s.t. } V_1(\pi) := 2pq \geq \frac{1}{8} \\
\phantom{\text{s.t. }} V_2(\pi) := 2(1-p)(1-q) \geq \frac{1}{8}
\end{cases}
$$

The above problem has two optimal solutions, $p = q = \frac{1}{4}$ and $p = q = \frac{3}{4}$. Both of them have $V_0(\pi) = \frac{5}{4}$. Therefore, $\max_{\text{product policy } \pi} \min_{\lambda \in \mathbb{R}_+^K} L(\pi, \lambda) = \frac{5}{4}$.

Now consider the following dual problem.

$$\min_{\lambda \in \mathbb{R}_+^2} \max_{p,q \in [0,1]} L(\pi, \lambda) := 2pq(1 + \lambda_1) + 2(1 - p)(1 - q)(1 + \lambda_2) - \frac{1}{8}(\lambda_1 + \lambda_2) \quad (15)$$

Fixing $\lambda \in \mathbb{R}_+^2$, $\max_{p,q \in [0,1]} L(\pi, \lambda)$ is equivalent to

$$\max_{p,q \in [0,1]} \left(p - \frac{1 + \lambda_2}{2 + \lambda_1 + \lambda_2}\right)\left(q - \frac{1 + \lambda_2}{2 + \lambda_1 + \lambda_2}\right)$$

If $\frac{1 + \lambda_2}{2 + \lambda_1 + \lambda_2} \leq \frac{1}{2}$ (i.e. $\lambda_1 \geq \lambda_2$), then the above problem has solution $p^* = q^* = 1$ which yields $L(p^*, q^*; \lambda) = 2(1 + \lambda_1) - \frac{1}{8}(\lambda_1 + \lambda_2)$; Otherwise if $\lambda_1 < \lambda_2$, $p^* = q^* = 0$ which yields $L(p^*, q^*; \lambda) = 2(1 + \lambda_2) - \frac{1}{8}(\lambda_1 + \lambda_2)$. Hence, $\max_{p,q \in [0,1]} L(\pi, \lambda) = 2 + 2\max(\lambda_1, \lambda_2) - \frac{1}{8}(\lambda_1 + \lambda_2)$, which has minimizer $\lambda^* = [0, 0]$ and the corresponding value $\min_{\lambda \in \mathbb{R}_+^2} \max_{p,q \in [0,1]} L(\pi, \lambda) = 2$. As a result, $\Delta = 2 - \frac{5}{4} = \frac{3}{4}$.

## D  PROOF OF THEOREM 2

Note that since $\bar{r}_{k,t} \in [0, 1]$, the value function $V_k(\pi)$ has the following bound for all policy $\pi$ and $k = 0, 1, \ldots, K$.

$$0 \leq V_k(\pi) = \mathbb{E}_\pi\Big[\sum_{t=0}^{\infty} \gamma^t \bar{r}_{k,t} \Big| s_0 \sim \rho\Big] \leq \frac{1}{1 - \gamma}. \quad (16)$$

Hence, the norm of $V(\pi) := [V_1(\pi); \ldots; V_K(\pi)] \in [0, 1]^K$ has the following bound

$$\|V(\pi)\| \leq \frac{\sqrt{K}}{1 - \gamma}. \quad (17)$$

Furthermore, Assumption 1 implies that there is a feasible product policy $\tilde{\pi}$ such that $0 \leq \xi_k \leq V_k(\tilde{\pi})$, so the norm of $\xi := [\xi_1; \ldots; \xi_K] \in \mathbb{R}^K$ has the following bound

$$\|\xi\| \leq \|V(\tilde{\pi})\| \leq \frac{\sqrt{K}}{1 - \gamma}. \quad (18)$$

Then,

$$0 \leq \|\lambda_T\|^2$$

$$\overset{(i)}{=} \sum_{t=0}^{T-1} \left(\|\lambda_{t+1}\|^2 - \|\lambda_t\|^2\right)$$

$$\overset{(ii)}{\leq} \sum_{t=0}^{T-1} \left(\big\|\lambda_t - \beta\big(\hat{V}(\pi_t) - \xi\big)\big\|^2 - \|\lambda_t\|^2\right)$$

$$\overset{(iii)}{\leq} 2\beta \sum_{t=0}^{T-1} \lambda_t^\top\big(\xi - \hat{V}(\pi_t)\big) + \beta^2 \sum_{t=0}^{T-1} \big(\big\|\hat{V}(\pi_t) - V(\pi_t)\big\| + \big\|V(\pi_t)\big\| + \|\xi\|\big)^2$$

$$\overset{(iv)}{\leq} 2\beta \sum_{t=0}^{T-1} \lambda_t^\top\big(V(\pi^*) - V(\pi_t)\big) + 2\beta \sum_{t=0}^{T-1} \lambda_t^\top\big(V(\pi_t) - \hat{V}(\pi_t)\big) + T\beta^2\Big(\frac{2\sqrt{K}}{1 - \gamma} + \epsilon_2\sqrt{K}\Big)^2$$

$$\overset{(v)}{\leq} 2\beta \sum_{t=0}^{T-1} \lambda_t^\top\big(V(\pi^*) - V(\pi_t)\big) + 2T\beta\epsilon_2 \sum_{k=1}^{K} \lambda_{k,\max} + \frac{8KT\beta^2}{(1 - \gamma)^2} + 2KT\beta^2\epsilon_2^2 \quad (19)$$

where (i) uses the initialization $\lambda_0 = 0$, (ii) uses the update rule (10), (iii) uses triangular inequality, and (iv) uses $|\hat{V}_k(\pi_t) - V_k(\pi_t)| \leq \epsilon_2$, Eqs. (17) and (18), $\lambda_{t,k} \geq 0$ as well as the constraint that

$V(\pi^*) \geq \xi$ satisfied by the optimal policy $\pi^*$ of the constrained cooperative MARL problem in Eq. (1), and (v) uses $\lambda_{t,k} \in [0, \lambda_{k,\max}]$ (based on the update rule (10)) as well as $|\widehat{V}_k(\pi_t) - V_k(\pi_t)| \leq \epsilon_2$. Rearranging the above inequality, we obtain that

$$\sum_{t=0}^{T-1} \lambda_t^\top \left(V(\pi_t) - V(\pi^*)\right) \leq T\epsilon_2 \sum_{k=1}^{K} \lambda_{k,\max} + \frac{4KT\beta}{(1-\gamma)^2} + KT\beta\epsilon_2^2. \tag{20}$$

Note that

$$\begin{aligned}
0 &\leq \sum_{t=0}^{T-1} \left(\max_\pi L(\pi, \lambda_t) - L(\pi^*, \lambda_t)\right) \\
&\overset{(i)}{\leq} \sum_{t=0}^{T-1} \left(\epsilon_1 + L(\pi_t, \lambda_t) - L(\pi^*, \lambda_t)\right) \\
&\overset{(ii)}{=} \sum_{t=0}^{T-1} \left(\epsilon_1 + V_0(\pi_t) - V_0(\pi^*) + \lambda_t^\top \left(V(\pi_t) - V(\pi^*)\right)\right) \\
&\overset{(iii)}{\leq} \sum_{t=0}^{T-1} \left(\epsilon_1 + V_0(\pi_t) - V_0(\pi^*)\right) + T\epsilon_2 \sum_{k=1}^{K} \lambda_{k,\max} + \frac{4KT\beta}{(1-\gamma)^2} + KT\beta\epsilon_2^2,
\end{aligned} \tag{21}$$

where (i) uses Eq. (9), (ii) uses the definition of the Lagrange function in Eq. (5), and (iii) uses Eq. (20). Rearranging the above inequality yields that

$$\begin{aligned}
V_0(\pi^*) - \mathbb{E}_{\widetilde{T}}\left[V_0(\pi_t)\right] &= \frac{1}{T} \sum_{t=0}^{T-1} \left[V_0(\pi^*) - V_0(\pi_t)\right] \\
&\leq \epsilon_2 \sum_{k=1}^{K} \lambda_{k,\max} + \frac{4K\beta}{(1-\gamma)^2} + \epsilon_1 + K\beta\epsilon_2^2 \\
&\overset{(i)}{\leq} \frac{7}{1-\gamma} \sqrt{\frac{K}{2T} \sum_{k=1}^{K} \lambda_{k,\max}^2},
\end{aligned}$$

where (i) uses the hyperparameter choices $\epsilon_1 = \frac{1}{1-\gamma}\sqrt{\frac{K}{2T}\sum_{k=1}^{K}\lambda_{k,\max}^2}$, $\epsilon_2 = \frac{1}{1-\gamma}\sqrt{\frac{\sum_{k=1}^{K}\lambda_{k,\max}^2}{2T(\sum_{k=1}^{K}\lambda_{k,\max})^2}} \leq \frac{1}{1-\gamma}$, $\beta = (1-\gamma)\sqrt{\frac{1}{2KT}\sum_{k=1}^{K}\lambda_{k,\max}^2}$. This proves the optimality gap in Eq. (7).

Next, we will prove the convergence rate (8) of the constraint violation.

For any $\widetilde{\lambda} := [\widetilde{\lambda}_1; \ldots; \widetilde{\lambda}_K] \in [0, \lambda_{k,\max}]^K$, it holds that

$$\begin{aligned}
&\|\lambda_{t+1} - \widetilde{\lambda}\|^2 \\
&\overset{(i)}{\leq} \left\|\lambda_t - \beta\left(\widehat{V}(\pi_t) - \xi\right) - \widetilde{\lambda}\right\|^2 \\
&\overset{(ii)}{\leq} \|\lambda_t - \widetilde{\lambda}\|^2 - 2\beta(\lambda_t - \widetilde{\lambda})^\top \left(V(\pi_t) - \xi\right) - 2\beta(\lambda_t - \widetilde{\lambda})^\top \left(\widehat{V}(\pi_t) - V(\pi_t)\right) \\
&\quad + \beta^2 \left(\|\widehat{V}(\pi_t) - V(\pi_t)\| + \|V(\pi_t)\| + \|\xi\|\right)^2 \\
&\overset{(iii)}{\leq} \|\lambda_t - \widetilde{\lambda}\|^2 - 2\beta(\lambda_t - \widetilde{\lambda})^\top \left(V(\pi_t) - \xi\right) + 2\beta\epsilon_2 \sum_{k=1}^{K} \lambda_{k,\max} + \beta^2 \left(\epsilon_2\sqrt{K} + \frac{2\sqrt{K}}{1-\gamma}\right)^2 \\
&\leq \|\lambda_t - \widetilde{\lambda}\|^2 - 2\beta(\lambda_t - \widetilde{\lambda})^\top \left(V(\pi_t) - \xi\right) + 2\beta\epsilon_2 \sum_{k=1}^{K} \lambda_{k,\max} + 2K\beta^2\epsilon_2^2 + \frac{8K\beta^2}{(1-\gamma)^2},
\end{aligned}$$

where (i) uses the update rule (10) and $\widetilde{\lambda}_k \in [0, \lambda_{k,\max}]$, (ii) uses triangular inequality, (iii) uses $\lambda_{t,k}, \widetilde{\lambda}_k \in [0, \lambda_{k,\max}]$, $|\widehat{V}_k(\pi_t) - V_k(\pi_t)| \leq \epsilon_2$, Eqs. (17) and (18). Telescoping the above inequality

over $t = 0, 1, \ldots, T - 1$ and using $\lambda_0 = 0$, we obtain that

$$\beta \sum_{t=0}^{T-1} (\lambda_t - \widetilde{\lambda})^\top (V(\pi_t) - \xi) \leq \frac{1}{2} \|\widetilde{\lambda}\|^2 + T\beta\epsilon_2 \sum_{k=1}^{K} \lambda_{k,\max} + KT\beta^2 \epsilon_2^2 + \frac{4TK\beta^2}{(1-\gamma)^2}. \qquad (22)$$

Since $V(\pi^*) \geq \xi$ and $\lambda_t \in \mathbb{R}_+^K$, Eq. (21) implies that

$$\beta \sum_{t=0}^{T-1} \lambda_t^\top (\xi - V(\pi_t)) \leq \beta \sum_{t=0}^{T-1} \left( \epsilon_1 + V_0(\pi_t) - V_0(\pi^*) \right) \qquad (23)$$

Summing up Eqs. (22) and (23) yields that

$$\beta \sum_{t=0}^{T-1} \widetilde{\lambda}^\top (\xi - V(\pi_t))$$

$$\leq \beta \sum_{t=0}^{T-1} \left( \epsilon_1 + V_0(\pi_t) - V_0(\pi^*) \right) + \frac{1}{2} \|\widetilde{\lambda}\|^2 + T\beta\epsilon_2 \sum_{k=1}^{K} \lambda_{k,\max} + KT\beta^2 \epsilon_2^2 + \frac{4KT\beta^2}{(1-\gamma)^2}. \qquad (24)$$

Note that

$$V_0(\pi^*) = \max_\pi \min_{\lambda \in \mathbb{R}_+^{d_m}} L(\pi, \lambda)$$

$$\overset{(i)}{=} \max_\pi L(\pi, \lambda^*) - \Delta$$

$$\geq L(\pi_t, \lambda^*) - \Delta$$

$$\overset{(ii)}{=} V_0(\pi_t) + (\lambda^*)^\top (V(\pi_t) - \xi) - \Delta$$

$$\overset{(iii)}{\geq} V_0(\pi_t) - (\lambda^*)^\top (\xi - V(\pi_t))_+ - \Delta \qquad (25)$$

where (i) uses the definition of the duality gap $\Delta$ in Eq. (6), (ii) uses the definition of the Lagrange function (5), and (iii) uses $\lambda^* \in \mathbb{R}_+^{d_m}$. Substituting the above inequality into Eq. (24) and rearranging it, we obtain that

$$\beta \sum_{t=0}^{T-1} \left( \widetilde{\lambda}^\top (\xi - V(\pi_t)) - (\lambda^*)^\top (\xi - V(\pi_t))_+ \right)$$

$$\leq \beta T(\Delta + \epsilon_1) + \frac{1}{2} \|\widetilde{\lambda}\|^2 + T\beta\epsilon_2 \sum_{k=1}^{K} \lambda_{k,\max} + KT\beta^2 \epsilon_2^2 + \frac{4KT\beta^2}{(1-\gamma)^2}. \qquad (26)$$

Using Eq. (64) and selecting $\widetilde{\lambda}_k = \lambda_{k,\max} I\{V_k(\pi_t) \leq \xi_k\}$ where $I\{\cdot\}$ is an indicator function, we obtain that

$$\widetilde{\lambda}^\top (\xi - V(\pi_t)) - (\lambda^*)^\top (\xi - V(\pi_t))_+ \geq \frac{1}{2} \sum_{k=1}^{K} \lambda_{k,\max} (\xi_k - V_k(\pi_t))_+,$$

Substituting the above inequality into Eq. (26) yields that

$$\frac{\beta}{2} \sum_{t=0}^{T-1} \sum_{k=1}^{K} \lambda_{k,\max} (\xi_k - V_k(\pi_t))_+$$

$$\leq \beta T(\Delta + \epsilon_1) + \frac{1}{2} \|\widetilde{\lambda}\|^2 + T\beta\epsilon_2 \sum_{k=1}^{K} \lambda_{k,\max} + KT\beta^2 \epsilon_2^2 + \frac{4KT\beta^2}{(1-\gamma)^2}$$

$$\overset{(i)}{\leq} \beta T(\Delta + \epsilon_1) + 2 \sum_{k=1}^{K} \lambda_{k,\max}^2 + T\beta\epsilon_2 \sum_{k=1}^{K} \lambda_{k,\max} + KT\beta^2 \epsilon_2^2 + \frac{4KT\beta^2}{(1-\gamma)^2},$$

where (i) uses $\|\widetilde{\lambda}\|^2 = \sum_{k=1}^{K} \widetilde{\lambda}_k^2 \leq 4 \sum_{k=1}^{K} \lambda_{k,\max}^2$. Finally, by dividing both sides of the above inequality by $T\beta$, we prove the convergence rate (8) of the constraint violation as follows.

$$\sum_{k=1}^{K} \lambda_{k,\max} \mathbb{E}_{\widetilde{T}} \big( \xi_k - V_k(\pi_{\widetilde{T}}) \big)_+$$

$$= \frac{1}{T} \sum_{t=0}^{T-1} \sum_{k=1}^{K} \lambda_{k,\max} \big( \xi_k - V_k(\pi_t) \big)_+$$

$$\leq 2\Delta + 2\epsilon_1 + \frac{4}{T\beta} \sum_{k=1}^{K} \lambda_{k,\max}^2 + 2\epsilon_2 \sum_{k=1}^{K} \lambda_{k,\max} + K\beta\epsilon_2^2 + \frac{8K\beta}{(1-\gamma)^2}$$

$$\leq 2\Delta + \frac{22}{1-\gamma} \sqrt{\frac{K}{2T} \sum_{k=1}^{K} \lambda_{k,\max}^2},$$

where (i) uses the hyperparameter choices $\epsilon_1 = \frac{1}{1-\gamma} \sqrt{\frac{K}{2T} \sum_{k=1}^{K} \lambda_{k,\max}^2}$, $\epsilon_2 = \frac{1}{1-\gamma} \sqrt{\frac{\sum_{k=1}^{K} \lambda_{k,\max}^2}{2T(\sum_{k=1}^{K} \lambda_{k,\max})^2}} \leq \frac{1}{1-\gamma}$, $\beta = (1-\gamma) \sqrt{\frac{1}{2KT} \sum_{k=1}^{K} \lambda_{k,\max}^2}$.

Furthermore, for any $\epsilon > 0$, implementing Algorithm 1 for $T = \frac{242}{K(1-\gamma)^2 \epsilon^2} \sum_{k=1}^{K} \lambda_{k,\max}^2 = \mathcal{O}(\epsilon^{-2})$ iterations, the output policy $\pi_{\widetilde{T}}$ satisfies the following convergence results based on the convergence rates (7) and (8).

$$V_0(\pi^*) - \mathbb{E}_{\widetilde{T}}[V_0(\pi_{\widetilde{T}})] \leq \frac{7}{1-\gamma} \sqrt{\frac{K}{2T} \sum_{k=1}^{K} \lambda_{k,\max}^2} \leq \frac{7\epsilon}{22},$$

$$\sum_{k=1}^{K} \lambda_{k,\max} \mathbb{E}_{\widetilde{T}} \big( \xi_k - V_k(\pi_{\widetilde{T}}) \big)_+ \leq \frac{22}{1-\gamma} \sqrt{\frac{K}{2T} \sum_{k=1}^{K} \lambda_{k,\max}^2} + 2\Delta \leq \epsilon + 2\Delta.$$

Each iteration of Algorithm 1 uses decentralized natural actor-critic algorithm (Chen et al., 2022) to obtain $\pi_t$ and model-based policy evaluation (Li et al., 2020) to obtain $\widehat{V}_k(\pi_t)$, which require $\mathcal{O}(\epsilon_1^{-3} \ln \epsilon_1^{-1})$ and $\mathcal{O}(\epsilon_2^{-2})$ samples to achieve precisions $\epsilon_1 = \frac{1}{1-\gamma} \sqrt{\frac{K}{2T} \sum_{k=1}^{K} \lambda_{k,\max}^2} = \mathcal{O}(\epsilon)$ and $\epsilon_2 = \frac{1}{1-\gamma} \sqrt{\frac{\sum_{k=1}^{K} \lambda_{k,\max}^2}{2T(\sum_{k=1}^{K} \lambda_{k,\max})^2}} = \mathcal{O}(\epsilon)$ respectively. Hence, the sample complexity of Algorithm 1 is

$$T\mathcal{O}(\epsilon_1^{-3} \ln \epsilon_1^{-1} + \epsilon_2^{-2}) = \mathcal{O}(\epsilon^{-2}) \mathcal{O}(\epsilon^{-3} \ln \epsilon^{-1} + \epsilon^{-2}) = \mathcal{O}(\epsilon^{-5} \ln \epsilon^{-1}).$$

## E    PROOF OF THEOREM 3

First, we list the hyperparameter choices of Algorithm 2 as follows.

$$\alpha = \sqrt{\frac{(1-\gamma)^3}{MT} \mathbb{E}_{s \sim \nu_{\pi^*}} \text{KL}[\pi^*(\cdot|s) \| \pi_0(\cdot|s)]}, \tag{27}$$

$$\eta = 8 \sqrt{\frac{M \mathbb{E}_{s \sim \nu_{\pi^*}} \text{KL}[\pi^*(\cdot|s) \| \pi_0(\cdot|s)]}{T(1-\gamma)^5} + \frac{2 \max_{1 \leq k \leq K} \zeta_k}{(1-\gamma)^2}}, \tag{28}$$

$$\epsilon_2 = \sqrt{\frac{M \mathbb{E}_{s \sim \nu_{\pi^*}} \text{KL}[\pi^*(\cdot|s) \| \pi_0(\cdot|s)]}{T(1-\gamma)^5}}, \tag{29}$$

$$\epsilon_3 = \sqrt{\frac{(1-\gamma) \mathbb{E}_{s \sim \nu_{\pi^*}} \text{KL}[\pi^*(\cdot|s) \| \pi_0(\cdot|s)]}{MT}}. \tag{30}$$

Specifically, $\alpha \leq 1$ if we choose the number of iterations $T \geq \frac{(1-\gamma)^3}{M} \mathbb{E}_{s \sim \nu_{\pi^*}} \text{KL}[\pi^*(\cdot|s) \| \pi_0(\cdot|s)]$. Furthermore, if we select uniform policy $\pi_0$ such that $\pi_0(a|s) = \frac{1}{|\mathcal{A}|}$, then $\text{KL}[\pi^*(\cdot|s) \| \pi_0(\cdot|s)] \leq \ln |\mathcal{A}|$ and thus we only require $T \geq \frac{(1-\gamma)^3}{M} \ln |\mathcal{A}|$ to let $\alpha \leq 1$.

Based on Eq. (75), we have

$$
\ln Z_t^{(m)}(s) - \alpha V_{k_t}(\pi_t; s)
$$

$$
= \ln \Big( \sum_{a'^{(m)}} \pi_t^{(m)}(a'^{(m)}|s) \exp \big(\alpha \widehat{Q}_{k_t}^{(m)}(\pi_t; s, a'^{(m)})\big) \Big) - \alpha V_{k_t}(\pi_t; s)
$$

$$
\geq \sum_{a'^{(m)}} \pi_t^{(m)}(a'^{(m)}|s) \ln \exp \big(\alpha \widehat{Q}_{k_t}^{(m)}(\pi_t; s, a'^{(m)})\big) - \alpha V_{k_t}(\pi_t; s)
$$

$$
= \alpha \sum_{a'^{(m)}} \pi_t^{(m)}(a'^{(m)}|s) \big(\widehat{Q}_{k_t}^{(m)}(\pi_t; s, a'^{(m)}) - Q_{k_t}^{(m)}(\pi_t; s, a'^{(m)})\big)
$$

$$
\geq -\alpha \max_{s,a^{(m)}} \big|\widehat{Q}_{k_t}^{(m)}(\pi_t; s, a^{(m)}) - Q_{k_t}^{(m)}(\pi_t; s, a^{(m)})\big| \geq -\alpha \epsilon_3,
$$

which means

$$
\frac{1}{\alpha} \ln Z_t^{(m)}(s) - V_{k_t}(\pi_t; s) + \epsilon_3 \geq 0. \tag{31}
$$

Therefore, we have

$$
(1-\gamma)\big(V_{k_t}(\pi_{t+1}; \rho') - V_{k_t}(\pi_t; \rho')\big)
$$

$$
\overset{(i)}{=} \mathbb{E}_{s,a\sim\nu_{t+1;\rho'}} A_{k_t}(\pi_t; s, a)
$$

$$
\overset{(ii)}{=} \mathbb{E}_{s\sim\nu_{t+1;\rho'}} \sum_{m=1}^{M} \sum_{a^{(m)}} \pi_{t+1}^{(m)}(a^{(m)}|s) \big(\widehat{Q}_{k_t}^{(m)}(\pi_t; s, a^{(m)}) - V_{k_t}(\pi_t; s)\big)
$$

$$
- \mathbb{E}_{s\sim\nu_{t+1;\rho'}} \sum_{m=1}^{M} \sum_{a^{(m)}} \pi_{t+1}^{(m)}(a^{(m)}|s) \big(\widehat{Q}_{k_t}^{(m)}(\pi_t; s, a^{(m)}) - Q_{k_t}^{(m)}(\pi_t; s, a^{(m)})\big)
$$

$$
+ \mathbb{E}_{s,a\sim\nu_{t+1;\rho'}} \left( A_{k_t}(\pi_t; s, a) - \sum_{m=1}^{M} A_{k_t}^{(m)}(\pi_t; s, a^{(m)}) \right)
$$

$$
\overset{(iii)}{\geq} \mathbb{E}_{s\sim\nu_{t+1;\rho'}} \sum_{m=1}^{M} \left( \frac{1}{\alpha} \ln Z_t^{(m)}(s) - V_{k_t}(\pi_t; s) + \frac{1}{\alpha} \sum_{a^{(m)}} \pi_{t+1}^{(m)}(a^{(m)}|s) \ln \frac{\pi_{t+1}^{(m)}(a^{(m)}|s)}{\pi_t^{(m)}(a^{(m)}|s)} \right)
$$

$$
- \sum_{m=1}^{M} \max_{s,a^{(m)}} \big|\widehat{Q}_{k_t}^{(m)}(\pi_t; s, a^{(m)}) - Q_{k_t}^{(m)}(\pi_t; s, a^{(m)})\big| - \zeta_k
$$

$$
\overset{(iv)}{\geq} \mathbb{E}_{s\sim\nu_{t+1;\rho'}} \sum_{m=1}^{M} \left( \frac{1}{\alpha} \ln Z_t^{(m)}(s) - V_{k_t}(\pi_t; s) + \epsilon_3 \right) - 2M\epsilon_3 - \zeta_{k_t}
$$

$$
\overset{(v)}{\geq} (1-\gamma)\mathbb{E}_{s\sim\rho'} \sum_{m=1}^{M} \left( \frac{1}{\alpha} \ln Z_t^{(m)}(s) - V_{k_t}(\pi_t; s) + \epsilon_3 \right) - 2M\epsilon_3 - \zeta_{k_t}
$$

where (i) denotes the occupation measure $\nu_{t+1;\rho'} := (1-\gamma)\sum_{t=0}^{\infty} \gamma^t \mathbb{P}_{\pi_{t+1}}(s_t = s|s_0 \sim \rho')$ and uses the performance difference lemma (Lemma 6.1 of Kakade and Langford (2002)), (ii) uses $A_{k_t}^{(m)}(\pi_t; s, a^{(m)}) = Q_{k_t}^{(m)}(\pi_t; s, a^{(m)}) - V_{k_t}(\pi_t; s)$, (iii) uses the policy update rule (76) and $\zeta_k := \sup_{s,a,\pi} \big|A_k(\pi; s, a) - \sum_{m=1}^{M} A_k^{(m)}(\pi; s, a^{(m)})\big|$, (iv) uses $\mathrm{KL}\big(\pi_{t+1}^{(m)}(\cdot|s)\|\pi_t^{(m)}(\cdot|s)\big) = \sum_{a^{(m)}} \pi_{t+1}^{(m)}(a^{(m)}|s) \ln \frac{\pi_{t+1}^{(m)}(a^{(m)}|s)}{\pi_t^{(m)}(a^{(m)}|s)} \geq 0$ and $\max_{s,a^{(m)}} \big|\widehat{Q}_{k_t}^{(m)}(\pi_t; s, a^{(m)}) - Q_{k_t}^{(m)}(\pi_t; s, a^{(m)})\big| \leq \epsilon_3$, and (v) uses Eq. (31) and $\nu_{t+1;\rho'}(s) \geq (1-\gamma)\rho'(s)$. The above inequality can be rearranged as follows.

$$
\mathbb{E}_{s\sim\rho'} \sum_{m=1}^{M} \Big( \ln Z_t^{(m)}(s) + \alpha\epsilon_3 - \alpha V_{k_t}(\pi_t; s) \Big)
$$

$$
\leq \frac{2\alpha M\epsilon_3}{1-\gamma} + \frac{\alpha\zeta_{k_t}}{1-\gamma} + \alpha\big(V_{k_t}(\pi_{t+1}; \rho') - V_{k_t}(\pi_t; \rho')\big)
$$

$$\stackrel{(i)}{\leq} \frac{2\alpha M\epsilon_3}{1-\gamma} + \frac{\alpha\zeta_{k_t}}{1-\gamma} + \alpha\Big(\frac{M\alpha}{(1-\gamma)^3} + \frac{2M\alpha\epsilon_3}{(1-\gamma)^2}\Big)$$

$$\stackrel{(ii)}{\leq} \frac{\alpha\zeta_{k_t}}{1-\gamma} + \frac{4M\alpha\epsilon_3}{(1-\gamma)^2} + \frac{M\alpha^2}{(1-\gamma)^3} \tag{32}$$

where (i) uses Eq. (69) and (ii) uses $\alpha \leq 1$. Then, we have

$$\mathbb{E}_{s\sim\nu_{\pi^*}}\big[\mathrm{KL}\big(\pi^*(\cdot|s)||\pi_{t+1}(\cdot|s)\big) - \mathrm{KL}\big(\pi^*(\cdot|s)||\pi_t(\cdot|s)\big)\big]$$

$$= \mathbb{E}_{s\sim\nu_{\pi^*}}\mathbb{E}_{a\sim\pi^*(\cdot|s)}\Big[\ln\frac{\pi^*(a|s)}{\pi_{t+1}(a|s)} - \ln\frac{\pi^*(a|s)}{\pi_t(a|s)}\Big]$$

$$= \mathbb{E}_{s,a\sim\nu_{\pi^*}}\sum_{m=1}^{M}\big[\ln\pi_t^{(m)}(a^{(m)}|s) - \ln\pi_{t+1}^{(m)}(a^{(m)}|s)\big]$$

$$\stackrel{(i)}{=} \mathbb{E}_{s,a\sim\nu_{\pi^*}}\sum_{m=1}^{M}\Big[\ln Z_t^{(m)}(s) + \alpha\epsilon_3 - \alpha V_{k_t}(\pi_t;s) + \alpha V_{k_t}(\pi_t;s) - \alpha\epsilon_3 - \alpha\widehat{Q}_{k_t}^{(m)}(\pi_t;s,a^{(m)})\Big]$$

$$\stackrel{(ii)}{\leq} \frac{\alpha\zeta_{k_t}}{1-\gamma} + \frac{4M\alpha\epsilon_3}{(1-\gamma)^2} + \frac{M\alpha^2}{(1-\gamma)^3} - \alpha\mathbb{E}_{s,a\sim\nu_{\pi^*}}\sum_{m=1}^{M}\big(Q_{k_t}^{(m)}(\pi_t;s,a^{(m)}) - V_{k_t}(\pi_t;s)\big)$$

$$\stackrel{(iii)}{=} \frac{\alpha\zeta_{k_t}}{1-\gamma} + \frac{4M\alpha\epsilon_3}{(1-\gamma)^2} + \frac{M\alpha^2}{(1-\gamma)^3} - \alpha\mathbb{E}_{s,a\sim\nu_{\pi^*}}\sum_{m=1}^{M}A_{k_t}^{(m)}(\pi_t;s,a^{(m)})$$

$$\stackrel{(iv)}{\leq} \frac{\alpha\zeta_{k_t}}{1-\gamma} + \frac{4M\alpha\epsilon_3}{(1-\gamma)^2} + \frac{M\alpha^2}{(1-\gamma)^3} - \alpha\mathbb{E}_{s,a\sim\nu_{\pi^*}}A_{k_t}(\pi_t;s,a) + \alpha\zeta_{k_t}$$

$$\stackrel{(v)}{\leq} \frac{\alpha\zeta_{k_t}}{1-\gamma} + \frac{4M\alpha\epsilon_3}{(1-\gamma)^2} + \frac{M\alpha^2}{(1-\gamma)^3} - \alpha(1-\gamma)\big(V_{k_t}(\pi^*) - V_{k_t}(\pi_t)\big), \tag{33}$$

where (i) uses the update rule (76), (ii) uses $\max_{s,a^{(m)}}\big|\widehat{Q}_{k_t}^{(m)}(\pi_t;s,a^{(m)}) - Q_{k_t}^{(m)}(\pi_t;s,a^{(m)})\big| \leq \epsilon_3$ and Eq. (32) for $\rho' = \nu_{\pi^*}$, (iii) uses the definition of the advantage function that $A_{k_t}^{(m)}(\pi_t;s,a^{(m)}) = Q_{k_t}^{(m)}(\pi_t;s,a^{(m)}) - V_{k_t}^{(m)}(\pi_t;s)$, (iv) denotes that $\zeta_k := \sup_{s,a,\pi}\big|A_k(\pi;s,a) - \sum_{m=1}^{M}A_k^{(m)}(\pi;s,a^{(m)})\big|$, and (v) uses $\alpha \leq 1$ as well as the performance difference lemma (Lemma 6.1 of Kakade and Langford (2002)) which implies that $\mathbb{E}_{s,a\sim\nu_{\pi^*}}A_{k_t}(\pi_t;s,a) = (1-\gamma)\big(V_{k_t}(\pi^*) - V_{k_t}(\pi_t)\big)$. Rearranging and averaging the above inequality (33) over $t = 0,1,\ldots,T-1$, we obtain that

$$\frac{1}{T}\sum_{t=0}^{T-1}\Big(V_{k_t}(\pi^*) - V_{k_t}(\pi_t) - \frac{\zeta_{k_t}}{(1-\gamma)^2} - \frac{4M\epsilon_3}{(1-\gamma)^3} - \frac{M\alpha}{(1-\gamma)^4}\Big)$$

$$\leq \frac{\mathbb{E}_{s\sim\nu_{\pi^*}}\mathrm{KL}\big(\pi^*(\cdot|s)||\pi_0(\cdot|s)\big)}{T\alpha(1-\gamma)}. \tag{34}$$

Denote $\mathcal{N}_k := \{0 \leq t \leq T-1 : k_t = k\}$. Then based on the design of Algorithm 2, for any $t \in \mathcal{N}_0$ (including $t = \widetilde{T}$) and $1 \leq k \leq K$, we have $\widehat{V}_k(\pi_t) \geq \xi_k - \eta$, so the convergence rate (13) of the constraint violation can be proved as follows.

$$V_k(\pi_{\widetilde{T}}) \geq \widehat{V}_k(\pi_{\widetilde{T}}) - |\widehat{V}_k(\pi_{\widetilde{T}}) - V_k(\pi_{\widetilde{T}})|$$

$$\stackrel{(i)}{\geq} \xi_k - \eta - \epsilon_2$$

$$\stackrel{(ii)}{=} \xi_k - 9\sqrt{\frac{M\mathbb{E}_{s\sim\nu_{\pi^*}}\mathrm{KL}[\pi^*(\cdot|s)||\pi_0(\cdot|s)]}{T(1-\gamma)^5}} - \frac{2\max_{1\leq k\leq K}\zeta_k}{(1-\gamma)^2}$$

where (i) uses $\widehat{V}_k(\pi_t) \geq \xi_k - \eta$ and $|\widehat{V}_k(\pi_t) - V_k(\pi_t)| \leq \epsilon_2$, and (ii) uses the hyperparameter choices (28) and (29). Conversely, for any $t \in \mathcal{N}_k$ ($1 \leq k \leq K$), we have $\widehat{V}_k(\pi_t) < \xi_k - \eta \leq V_k(\pi^*) - \eta$, so in a similar way we can prove that

$$V_k(\pi_t) \leq \widehat{V}_k(\pi_t) + |\widehat{V}_k(\pi_t) - V_k(\pi_t)| \leq V_k(\pi^*) - \eta + \epsilon_2. \tag{35}$$

Substituting Eq. (35) into Eq. (34), we obtain that

$$
\frac{\mathbb{E}_{s \sim \nu_{\pi^*}} \mathrm{KL}\big(\pi^*(\cdot|s) \| \pi_0(\cdot|s)\big)}{T\alpha(1-\gamma)}
$$

$$
\geq \frac{1}{T} \sum_{t \in \mathcal{N}_0} \Big( V_0(\pi^*) - V_0(\pi_t) - \frac{\zeta_0}{(1-\gamma)^2} - \frac{4M\epsilon_3}{(1-\gamma)^3} - \frac{M\alpha}{(1-\gamma)^4} \Big)
$$

$$
+ \frac{1}{T} \sum_{k=1}^{K} \sum_{t \in \mathcal{N}_k} \Big( V_k(\pi^*) - V_k(\pi_t) - \frac{\zeta_k}{(1-\gamma)^2} - \frac{4M\epsilon_3}{(1-\gamma)^3} - \frac{M\alpha}{(1-\gamma)^4} \Big)
$$

$$
\overset{(i)}{\geq} \frac{1}{T} \sum_{t \in \mathcal{N}_0} \Big( V_0(\pi^*) - V_0(\pi_t) - \frac{\zeta_0}{(1-\gamma)^2} - \frac{4M\epsilon_3}{(1-\gamma)^3} - \frac{M\alpha}{(1-\gamma)^4} \Big)
$$

$$
+ \frac{1}{T} \sum_{k=1}^{K} \sum_{t \in \mathcal{N}_k} \Big( \eta - \epsilon_2 - \frac{\max_{1 \leq k \leq K} \zeta_k}{(1-\gamma)^2} - \frac{4M\epsilon_3}{(1-\gamma)^3} - \frac{M\alpha}{(1-\gamma)^4} \Big)
$$

$$
\overset{(ii)}{=} \frac{1}{T} \sum_{t \in \mathcal{N}_0} \Big( V_0(\pi^*) - V_0(\pi_t) - \frac{\zeta_0}{(1-\gamma)^2} - \frac{4M\epsilon_3}{(1-\gamma)^3} - \frac{M\alpha}{(1-\gamma)^4} \Big)
$$

$$
+ \frac{T - |\mathcal{N}_0|}{T} \Big( \eta - \epsilon_2 - \frac{\max_{1 \leq k \leq K} \zeta_k}{(1-\gamma)^2} - \frac{4M\epsilon_3}{(1-\gamma)^3} - \frac{M\alpha}{(1-\gamma)^4} \Big),
$$

where (i) uses Eq. (35) and (ii) uses $\sum_{k=1}^{K} |\mathcal{N}_k| = T - |\mathcal{N}_0|$. Substituting the hyperparameters choices (27)-(30) into the above inequality, we obtain that

$$
\epsilon_2 \geq \frac{1}{T} \sum_{t \in \mathcal{N}_0} \Big( V_0(\pi^*) - V_0(\pi_t) - 5\epsilon_2 - \frac{2\zeta_0}{(1-\gamma)^2} \Big) + \frac{2\epsilon_2 (T - |\mathcal{N}_0|)}{T} \tag{36}
$$

If $\mathcal{N}_0 = \emptyset$, then Eq. (36) above implies the contradiction that $|\mathcal{N}_0| \geq \frac{T}{2} > 0$. Hence, $\mathcal{N}_0 \neq \emptyset$.

Then we prove the convergence rate (12) of the policy optimality in the following two cases.

(Case 1) If $\sum_{t \in \mathcal{N}_0} \big( V_0(\pi^*) - V_0(\pi_t) - 5\epsilon_2 - \frac{\zeta_0}{(1-\gamma)^3} \big) > 0$, then Eq. (36) implies that $|\mathcal{N}_0| \geq \frac{T}{2} > 0$ and that $\sum_{t \in \mathcal{N}_0} \big( V_0(\pi^*) - V_0(\pi_t) - 5\epsilon_2 - \frac{\zeta_0}{(1-\gamma)^3} \big) \leq T\epsilon_2$. Then the convergence rate (12) can be proved as follows

$$
\mathbb{E}\big( V_0(\pi^*) - V_0(\pi_{\widetilde{T}}) \big)
$$

$$
= \frac{1}{|\mathcal{N}_0|} \sum_{t \in \mathcal{N}_0} \big( V_0(\pi^*) - V_0(\pi_t) \big)
$$

$$
= \frac{1}{|\mathcal{N}_0|} \sum_{t \in \mathcal{N}_0} \Big( V_0(\pi^*) - V_0(\pi_t) - 5\epsilon_2 - \frac{2\zeta_0}{(1-\gamma)^2} \Big) + 5\epsilon_2 + \frac{2\zeta_0}{(1-\gamma)^2}
$$

$$
\leq \frac{T\epsilon_2}{T/2} + 5\epsilon_2 + \frac{2\zeta_0}{(1-\gamma)^2}
$$

$$
\leq 7\epsilon_2 + \frac{2\zeta_0}{(1-\gamma)^2} = 7\sqrt{\frac{M\mathbb{E}_{s \sim \nu_{\pi^*}} \mathrm{KL}[\pi^*(\cdot|s) \| \pi_0(\cdot|s)]}{T(1-\gamma)^5}} + \frac{2\zeta_0}{(1-\gamma)^2}.
$$

(Case 2) If $\sum_{t \in \mathcal{N}_0} \big( V_0(\pi^*) - V_0(\pi_t) - 5\epsilon_2 - \frac{\zeta_0}{(1-\gamma)^3} \big) \leq 0$, then the convergence rate (12) can be proved as follows.

$$
\mathbb{E}\big( V_0(\pi^*) - V_0(\pi_{\widetilde{T}}) \big)
$$

$$
= \frac{1}{|\mathcal{N}_0|} \sum_{t \in \mathcal{N}_0} \Big( V_0(\pi^*) - V_0(\pi_t) - 5\epsilon_2 - \frac{2\zeta_0}{(1-\gamma)^2} \Big) + 5\epsilon_2 + \frac{2\zeta_0}{(1-\gamma)^2}
$$

$$
\leq 5\epsilon_2 + \frac{2\zeta_0}{(1-\gamma)^3} = 5\sqrt{\frac{M\mathbb{E}_{s \sim \nu_{\pi^*}} \mathrm{KL}[\pi^*(\cdot|s) \| \pi_0(\cdot|s)]}{T(1-\gamma)^5}} + \frac{2\zeta_0}{(1-\gamma)^2}.
$$

Furthermore, for any $\epsilon > 0$, the output policy $\pi_{\widetilde{T}}$ of Algorithm 2 after $T = \mathcal{O}(\epsilon^{-2})$ iterations satisfies the following convergence results based on the convergence rates (12) and (13).

$$V_0(\pi^*) - \mathbb{E}_{\widetilde{T}}[V_0(\pi_{\widetilde{T}})] \leq \mathcal{O}\left(\sqrt{\frac{M}{T(1-\gamma)^5}} + \frac{\zeta_0}{(1-\gamma)^2}\right) \leq \mathcal{O}\left(\epsilon + \frac{\zeta_0}{(1-\gamma)^2}\right),$$

$$\xi_k - \mathbb{E}_{\widetilde{T}}[V_k(\pi_{\widetilde{T}})] \leq \mathcal{O}\left(\sqrt{\frac{M}{T(1-\gamma)^5}} + \frac{\max_{1 \leq k \leq K} \zeta_k}{(1-\gamma)^2}\right) \leq \mathcal{O}\left(\epsilon + \frac{\max_{1 \leq k \leq K} \zeta_k}{(1-\gamma)^2}\right); \ 1 \leq k \leq K.$$

Each iteration of Algorithm 2 uses model-based policy evaluation (Chen et al., 2022) to obtain $\widehat{V}_k(\pi_t)$ and $\widehat{Q}_{k_t}^{(m)}(\pi_t; s, a^{(m)})$, which require $\mathcal{O}(\epsilon_2^{-2})$ and $\mathcal{O}(\epsilon_3^{-2})$ samples to achieve precisions $\epsilon_2 = \mathcal{O}(\epsilon)$ (by substituting $T = \mathcal{O}(\epsilon^{-2})$ into Eq. (29)) and $\epsilon_3 = \mathcal{O}(\epsilon)$ (by substituting $T = \mathcal{O}(\epsilon^{-2})$ into Eq. (30)) respectively. Therefore, the sample complexity of Algorithm 2 is

$$T\mathcal{O}(\epsilon_2^{-2} + \epsilon_3^{-2}) = \mathcal{O}(\epsilon^{-2})\mathcal{O}(\epsilon^{-2} + \epsilon^{-2}) = \mathcal{O}(\epsilon^{-4}).$$

## F  PROOF OF THEOREM 4

We repeat Example 1 as follows.

**Example 1.** *Consider a constrained cooperative MARL problem with two agents, a single state $\mathcal{S} = \{s\}$. Both agents share the same action space $\mathcal{A}^{(m)} = \{0, 1\}$ and the same reward and safety scores listed below. The discount factor is $\gamma = \frac{1}{2}$ and the safety thresholds are $\xi_1 = \xi_2 = \frac{1}{8}$.*

$$r_0^{(m)}(s, [0, 0]) = 1, \quad r_1^{(m)}(s, [0, 0]) = 1, \quad r_2^{(m)}(s, [0, 0]) = 0$$
$$r_0^{(m)}(s, [0, 1]) = 0, \quad r_1^{(m)}(s, [0, 1]) = 0, \quad r_2^{(m)}(s, [0, 1]) = 0$$
$$r_0^{(m)}(s, [1, 0]) = 0, \quad r_1^{(m)}(s, [1, 0]) = 0, \quad r_2^{(m)}(s, [1, 0]) = 0$$
$$r_0^{(m)}(s, [1, 1]) = 1, \quad r_1^{(m)}(s, [1, 1]) = 0, \quad r_2^{(m)}(s, [1, 1]) = 1$$

In the above example, any product policy $\pi(a|s) = \pi^{(1)}(a^{(1)}|s)\pi^{(2)}(a^{(2)}|s)$ can be parameterized by $p = \pi^{(1)}(0|s) \in [0, 1]$ and $q = \pi^{(2)}(0|s) \in [0, 1]$. Then the aim of the constrained cooperative MARL problem in Example 1 can be formulated as

$$\begin{cases} \max_{p,q \in [0,1]} V_0(\pi) := 2pq + 2(1-p)(1-q) \\[2mm] \text{s.t. } V_1(\pi) := 2pq \geq \frac{1}{8} \\[2mm] \quad\quad V_2(\pi) := 2(1-p)(1-q) \geq \frac{1}{8} \end{cases} \tag{37}$$

**Proof for the primal-dual algorithm:** Since $\epsilon_1 = 0$, $(p_t, q_t)$ in the primal-dual algorithm (Algorithm 1) is obtained by solving $\arg\max_\pi L(\pi, \lambda_t)$. In Appendix B, we have obtained that $(p_t, q_t) = (1, 1)$ where $V_1(\pi_t) = 1 > \xi_1$ if $\lambda_1 \geq \lambda_2$ and $(p_t, q_t) = (0, 0)$ where $V_2(\pi_t) = 1 > \xi_2$ if $\lambda_1 < \lambda_2$. Hence, the policy $\pi_t$ is infeasible for all $t$.

**Update rules of the primal algorithm for Example 1:**

Next, we analyze the primal algorithm (Algorithm 2) on Example 1. Note that there is only one state $s$ in Example 1, so $V_k(\pi) \equiv V_k(\pi)(s)$, and thus the local Q function can be computed by Bellman equation as follows.

$$Q_{k_t}^{(m)}(\pi_t; s, a^{(m)}) = \sum_{a^{(\backslash m)}} \pi^{(\backslash m)}(a^{(\backslash m)}|s)\bar{r}_k(s, a) + \gamma V_k(\pi). \tag{38}$$

Hence, the NPG update rule (11) becomes

$$\pi_{t+1}^{(m)}(0|s) = \frac{\pi_t^{(m)}(0|s)\exp\left(\alpha\widehat{Q}_{k_t}^{(m)}(\pi_t; s, 0)\right)}{\pi_t^{(m)}(0|s)\exp\left(\alpha\widehat{Q}_{k_t}^{(m)}(\pi_t; s, 0)\right) + \pi_t^{(m)}(1|s)\exp\left(\alpha\widehat{Q}_{k_t}^{(m)}(\pi_t; s, 1)\right)}$$

$$\overset{(i)}{=} \frac{\pi_t^{(m)}(0|s)}{\pi_t^{(m)}(0|s) + \pi_t^{(m)}(1|s) \exp\left(\alpha Q_{k_t}^{(m)}(\pi_t; s, 1) - \alpha Q_{k_t}^{(m)}(\pi_t; s, 0)\right)} \tag{39}$$

where (i) uses $|\widehat{Q}_k^{(m)}(\pi; s, a^{(m)}) - Q_k^{(m)}(\pi; s, a^{(m)})| \le \epsilon_3 = 0$. Note that Eq. (38) implies that

$$
\begin{aligned}
&Q_{k_t}^{(1)}(\pi_t; s, 1) - Q_{k_t}^{(1)}(\pi_t; s, 0) \\
&= q_t\big(\overline{r}_{k_t}(s, [1, 0]) - \overline{r}_{k_t}(s, [0, 0])\big) + (1 - q_t)\big(\overline{r}_{k_t}(s, [1, 1]) - \overline{r}_{k_t}(s, [0, 1])\big)
\end{aligned} \tag{40}
$$

$$
\begin{aligned}
&Q_{k_t}^{(2)}(\pi_t; s, 1) - Q_{k_t}^{(2)}(\pi_t; s, 0) \\
&= p_t\big(\overline{r}_{k_t}(s, [0, 1]) - \overline{r}_{k_t}(s, [0, 0])\big) + (1 - p_t)\big(\overline{r}_{k_t}(s, [1, 1]) - \overline{r}_{k_t}(s, [1, 0])\big)
\end{aligned} \tag{41}
$$

Substituting Eqs. (40) and (41) as well as the expressions of $r_k^{(m)}(s, a)$ defined by Example 1 into the update rule (39), we further obtain the following update rules of $p_t := \pi_t^{(1)}(0|s)$ and $q_t := \pi_t^{(2)}(0|s)$.

$$
p_{t+1} = \begin{cases}
\dfrac{p_t}{p_t + (1 - p_t)\exp\big(\alpha(1 - 2q_t)\big)}; & \text{if } k_t = 0 \\[3ex]
\dfrac{p_t}{p_t + (1 - p_t)\exp\big(-\alpha q_t\big)}; & \text{if } k_t = 1 \\[3ex]
\dfrac{p_t}{p_t + (1 - p_t)\exp\big(\alpha(1 - q_t)\big)}; & \text{if } k_t = 2
\end{cases} \tag{42}
$$

$$
q_{t+1} = \begin{cases}
\dfrac{q_t}{q_t + (1 - q_t)\exp\big(\alpha(1 - 2p_t)\big)}; & \text{if } k_t = 0 \\[3ex]
\dfrac{q_t}{q_t + (1 - q_t)\exp\big(-\alpha p_t\big)}; & \text{if } k_t = 1 \\[3ex]
\dfrac{q_t}{q_t + (1 - q_t)\exp\big(\alpha(1 - p_t)\big)}; & \text{if } k_t = 2
\end{cases} \tag{43}
$$

Next, we prove the convergence of the above primal update rules (42) and (43) to the optimal solution $p, q = \frac{1}{4}$. Starting from an initial policy satisfying $\frac{2}{3}q_0 \le p_0 \le \frac{3}{2}q_0$ and $0.06 \le p_0 q_0 \le 0.135$, we will prove the following three useful statements for all $t \ge 0$:

$(A_t)$: $0.06 \le p_t q_t \le 0.135$ and $\frac{2}{3} \le \frac{p_t}{q_t} \le 1.5$, which implies that $p_t, q_t \in [0.2, 0.45]$.

$(B_t)$: If $p_t q_t \ge \frac{1}{16} - \frac{\eta}{2} = \frac{1}{16} + 3\alpha$, $p_{t+1}q_{t+1} \le \frac{p_t q_t}{1 + 0.11\alpha}$; Otherwise, $p_{t+1}q_{t+1} \ge \frac{p_t q_t}{1 - 0.19\alpha}$.

$(C_t)$: If $\left|\frac{p_t}{q_t} - 1\right| > 5\alpha$, then $\left|\frac{p_{t+1}}{q_{t+1}} - 1\right| \le (1 - 0.079\alpha)\left|\frac{p_t}{q_t} - 1\right|$.

Since $(A_0)$ holds, we will prove the above three statements by proving the induction arguments that $(A_t), (B_t), (C_t) \Rightarrow (A_{t+1})$ and that $(A_t) \Rightarrow (B_t), (C_t)$.

**Upper bound the change of $p_t q_t$ and $\frac{p_t}{q_t}$ under $(A_t)$:** Next, we will prove that under the statement $(A_t)$, the change of the potential functions $p_t q_t$ and $\frac{p_t}{q_t}$ will always be upper bounded by $\mathcal{O}(\alpha)$. Substituting $M = 2$ and $\epsilon_3 = 0$ into Eq. (68), we have

$$
\sum_{m=1}^{M} \sum_{a^{(m)}} |\pi_{t+1}^{(m)}(a^{(m)}|s) - \pi_t^{(m)}(a^{(m)}|s)|
$$

$$
= 2|p_{t+1} - p_t| + 2|q_{t+1} - q_t| \le M\alpha\left(\frac{1}{1 - \gamma} + 2\epsilon_3\right) = 4\alpha. \tag{44}
$$

Therefore, we have

$$
\left|\frac{p_{t+1}q_{t+1}}{p_t q_t} - 1\right| \le \left|\frac{p_{t+1}(q_{t+1} - q_t) + q_t(p_{t+1} - p_t)}{p_t q_t}\right|
$$

$$
\overset{(i)}{\le} 17|p_{t+1} - p_t| + 17|q_{t+1} - q_t| \overset{(ii)}{\le} 34\alpha, \tag{45}
$$

and

$$
\left|\frac{p_{t+1}}{q_{t+1}} - \frac{p_t}{q_t}\right|
$$

$$= \left| \frac{q_t(p_{t+1} - p_t) - p_t(q_{t+1} - q_t)}{q_{t+1}q_t} \right|$$

$$\overset{(iii)}{\leq} \frac{1}{0.43}(|p_{t+1} - p_t| + |q_{t+1} - q_t|)$$

$$\overset{(iv)}{\leq} 4.66\alpha, \tag{46}$$

where (i) uses $p_{t+1}, q_t \leq 1$ and $p_t q_t \geq 0.06$ (based on $(A_t)$), (ii) and (iv) use Eq. (44), and (iii) uses $p_t, q_t \leq 0.45$ and $q_{t+1}q_t \geq q_t^2 - q_t|q_{t+1} - q_t| \geq q_t(q_t - 2\alpha) \geq (0.45)(0.43)$ (based on Eq. (44)).

**Proof of $(A_t), (B_t), (C_t) \Rightarrow (A_{t+1})$:** Based on the statement $(A_t)$, we will prove that $0.06 \leq p_{t+1}q_{t+1} \leq 0.135$ in the following two cases of $p_t q_t$.

(Case I) If $0.06 \leq p_t q_t < \frac{1}{16} + 3\alpha$, then on one hand, based on the statement $(B_t)$, $p_{t+1}q_{t+1} \geq p_t q_t \geq 0.06$. On the other hand, based on Eq. (45), we have $p_{t+1}q_{t+1} \leq (1 + 34\alpha)p_t q_t \leq (1 + 34\alpha)\left(\frac{1}{16} + 3\alpha\right) \leq 0.135$.

(Case II) If $\frac{1}{16} + 3\alpha \leq p_t q_t \leq 0.135$, then on one hand, based on the statement $(B_t)$, $p_{t+1}q_{t+1} \leq p_t q_t \leq 0.135$. On the other hand, based on Eq. (45), we have $p_{t+1}q_{t+1} \geq (1 - 34\alpha)p_t q_t \geq (1 - 34 \times 10^{-3})\frac{1}{16} > 0.06$.

Then, we prove that $\frac{2}{3} \leq \frac{p_{t+1}}{q_{t+1}} \leq 1.5$ in the following two cases of $\frac{p_t}{q_t}$.

(Case I) If $\left| \frac{p_t}{q_t} - 1 \right| > 5\alpha$, then based on the statement $(C_t)$, we have $\left| \frac{p_{t+1}}{q_{t+1}} - 1 \right| \leq \left| \frac{p_t}{q_t} - 1 \right| \leq 1.5 - 1 = 0.5$ which implies that $\frac{p_{t+1}}{q_{t+1}} \leq 1.5$. Then suppose $\frac{p_{t+1}}{q_{t+1}} < \frac{2}{3} \leq \frac{p_t}{q_t}$, which along with Eq. (46) implies that $\frac{p_t}{q_t} \leq \frac{p_{t+1}}{q_{t+1}} + 4.66\alpha \leq \frac{2}{3} + 4.66 \times 10^{-3} < 1$. Hence, based on the statement $(C_t)$, $1 - \frac{p_{t+1}}{q_{t+1}} \leq 1 - \frac{p_t}{q_t}$, i.e., $\frac{p_{t+1}}{q_{t+1}} \geq \frac{p_t}{q_t}$, which contradicts with $\frac{p_{t+1}}{q_{t+1}} < \frac{2}{3} \leq \frac{p_t}{q_t}$. Therefore, $\frac{2}{3} \leq \frac{p_{t+1}}{q_{t+1}} \leq 1.5$ holds in Case I.

(Case II) If $\left| \frac{p_t}{q_t} - 1 \right| \leq 5\alpha$, then based on Eq. (46), we have

$$\left| \frac{p_{t+1}}{q_{t+1}} - 1 \right| \leq \left| \frac{p_t}{q_t} - 1 \right| + \left| \frac{p_{t+1}}{q_{t+1}} - \frac{p_t}{q_t} \right| \leq 9.66\alpha \leq 9.66 \times 10^{-3}, \tag{47}$$

which implies that $\frac{2}{3} \leq \frac{p_{t+1}}{q_{t+1}} \leq 1.5$.

**Proof of $(A_t) \Rightarrow (B_t), (C_t)$:** Since $p_t, q_t \in [0.2, 0.45]$ and $\eta = -6\alpha \geq -0.006$, the corresponding value function $V_2(\pi_t) = 2(1 - p_t)(1 - q_t) \geq 2(0.55)^2 > \frac{1}{8} - \eta$. Hence, we only need to consider the following two cases, $V_1(\pi_t) \geq \frac{1}{8} - \eta$ (i.e., $k_t = 0$) and $V_1(\pi_t) < \frac{1}{8} - \eta$ (i.e., $k_t = 1$).

(Case I) If $V_1(\pi_t) = 2p_t q_t \geq \frac{1}{8} - \eta$, then the case $k_t = 0$ of the update rules (42) and (43) is implemented. We will first bound the involved terms $\exp\big(\alpha(1 - 2q_t)\big)$ and $\exp\big(\alpha(1 - 2p_t)\big)$ as follows.

$$\exp\big(\alpha(1 - 2q_t)\big) \overset{(i)}{\leq} \exp(0.6\alpha) \overset{(ii)}{\leq} 1 + 0.6\alpha \exp(0.6\alpha) \overset{(iii)}{\leq} 1 + 0.7\alpha, \tag{48}$$

$$\exp\big(\alpha(1 - 2q_t)\big) \overset{(iv)}{\geq} \exp(0.1\alpha) \overset{(v)}{\geq} 1 + 0.1\alpha, \tag{49}$$

where (i) and (iv) use $q_t \in [0.2, 0.45]$, (ii) and (v) use $e^x = 1 + \int_0^x e^t dt \leq 1 + xe^x$ and $e^x \geq 1 + x$ respectively for any $x \geq 0$, and (iii) uses $\alpha \leq 10^{-3}$. In a similar way, we can obtain that

$$1 + 0.1\alpha \leq \exp\big(\alpha(1 - 2p_t)\big) \leq 1 + 0.7\alpha. \tag{50}$$

As the case $k_t = 0$ of the update rules (42) and (43) is implemented, we have

$$\frac{p_t q_t}{p_{t+1}q_{t+1}} = \big[p_t + (1 - p_t)\exp\big(\alpha(1 - 2q_t)\big)\big]\big[q_t + (1 - q_t)\exp\big(\alpha(1 - 2p_t)\big)\big]$$

$$\overset{(i)}{\geq} \big[p_t + (1 - p_t)(1 + 0.1\alpha)\big]\big[q_t + (1 - q_t)(1 + 0.1\alpha)\big]$$

$$= \big[1 + 0.1\alpha(1 - p_t)\big]\big[1 + 0.1\alpha(1 - q_t)\big]$$

$$\overset{(ii)}{\geq} (1 + 0.055\alpha)^2$$
$$\geq 1 + 0.11\alpha, \tag{51}$$

where (i) uses Eqs. (49) and (50), and (ii) uses $p_t, q_t \leq 0.45$.

When $p_t \geq q_t$, we have

$$\frac{p_{t+1}}{q_{t+1}} - 1$$

$$\overset{(i)}{=} \frac{p_t}{q_t} \frac{q_t + (1 - q_t) \exp\left(\alpha(1 - 2p_t)\right)}{p_t + (1 - p_t) \exp\left(\alpha(1 - 2q_t)\right)} - 1$$

$$= \frac{p_t}{q_t}\left(1 - \frac{p_t - q_t + (1 - p_t) \exp\left(\alpha(1 - 2q_t)\right) - (1 - q_t) \exp\left(\alpha(1 - 2p_t)\right)}{p_t + (1 - p_t) \exp\left(\alpha(1 - 2q_t)\right)}\right) - 1$$

$$= \frac{p_t}{q_t} - 1 - \frac{p_t}{q_t} \frac{(1 - q_t)\left[\exp\left(\alpha(1 - 2q_t)\right) - \exp\left(\alpha(1 - 2p_t)\right)\right] - (p_t - q_t)\left[\exp\left(\alpha(1 - 2q_t)\right) - 1\right]}{p_t + (1 - p_t) \exp\left(\alpha(1 - 2q_t)\right)}$$

$$\overset{(ii)}{\leq} \frac{p_t}{q_t} - 1 - \frac{p_t}{q_t} \frac{(0.55)2\alpha(p_t - q_t) - 0.7\alpha(p_t - q_t)}{1 + 0.7\alpha}$$

$$\overset{(iii)}{\leq} \frac{p_t}{q_t} - 1 - 0.079\alpha\left(\frac{p_t}{q_t} - 1\right)$$

$$\leq (1 - 0.079\alpha)\left(\frac{p_t}{q_t} - 1\right), \tag{52}$$

where (i) uses the case $k_t = 0$ of the update rules (42) and (43), (ii) uses $q_t \leq 0.45$, $p_t - q_t \geq 0$, Eq. (48) and $\exp\left(\alpha(1 - 2q_t)\right) - \exp\left(\alpha(1 - 2p_t)\right) \geq 2\alpha(p_t - q_t) \geq 0$, (iii) uses $\alpha \leq 10^{-3}$ and $p_t \geq 0.2$. Similarly, when $p_t < q_t$, we have

$$1 - \frac{p_{t+1}}{q_{t+1}}$$

$$= 1 - \frac{p_t}{q_t}\left(1 + \frac{(1 - q_t)\left[\exp\left(\alpha(1 - 2p_t)\right) - \exp\left(\alpha(1 - 2q_t)\right)\right] - (q_t - p_t)\left[\exp\left(\alpha(1 - 2q_t)\right) - 1\right]}{p_t + (1 - p_t) \exp\left(\alpha(1 - 2q_t)\right)}\right)$$

$$\leq 1 - \frac{p_t}{q_t} - \frac{p_t}{q_t} \frac{(0.55)2\alpha(q_t - p_t) - 0.7\alpha(q_t - p_t)}{1 + 0.7\alpha}$$

$$\leq 1 - \frac{p_t}{q_t} - 0.079\alpha p_t\left(1 - \frac{p_t}{q_t}\right)$$

$$\overset{(i)}{\leq} (1 - 0.079\alpha)\left(1 - \frac{p_t}{q_t}\right) \tag{53}$$

where (i) uses $q_t \leq 0.45$.

(Case II) If $V_1(\pi_t) = 2p_t q_t < \frac{1}{8} - \eta$, then the case $k_t = 1$ of the update rules (42) and (43) is implemented. Hence, we obtain that

$$\frac{p_t q_t}{p_{t+1} q_{t+1}} = \left[p_t + (1 - p_t) \exp(-\alpha q_t)\right]\left[q_t + (1 - q_t) \exp(-\alpha p_t)\right]$$

$$\overset{(i)}{\leq} \left[p_t + (1 - p_t)(1 - 0.19\alpha)\right]\left[q_t + (1 - q_t)(1 - 0.19\alpha)\right]$$

$$= [1 - 0.19\alpha(1 - p_t)][1 - 0.19\alpha(1 - q_t)]$$

$$\overset{(ii)}{\leq} (1 - 0.1\alpha)^2 \leq 1 - 0.2\alpha + 0.01\alpha^2 \overset{(iii)}{\leq} 1 - 0.19\alpha, \tag{54}$$

where (i) uses the following Eq. (55), (ii) uses $p_t, q_t \leq 0.45$, and (iii) uses $\alpha \leq 10^{-3}$.

$$\exp(-\alpha q_t) \leq 1 - \alpha q_t + \frac{1}{2}(\alpha q_t)^2$$

$$\leq 1 - \alpha q_t + \frac{(10^{-3})(0.45)}{2}\alpha q_t$$

$$\leq 1 - 0.99\alpha q_t \leq 1 - 0.99\alpha(0.2) \leq 1 - 0.19\alpha. \tag{55}$$

When $p_t \geq q_t$, we have

$$
\begin{aligned}
&\frac{p_{t+1}}{q_{t+1}} - 1 \\
&\overset{(i)}{=} \frac{p_t}{q_t} \frac{q_t + (1 - q_t)\exp(-\alpha p_t)}{p_t + (1 - p_t)\exp(-\alpha q_t)} - 1 \\
&= \frac{p_t}{q_t}\left(1 - \frac{p_t - q_t + (1 - p_t)\exp(-\alpha q_t) - (1 - q_t)\exp(-\alpha p_t)}{p_t + (1 - p_t)\exp(-\alpha q_t)}\right) - 1 \\
&= \frac{p_t}{q_t} - 1 - \frac{p_t}{q_t} \frac{(1 - q_t)\big[\exp(-\alpha q_t) - \exp(-\alpha p_t)\big] + (p_t - q_t)\big[1 - \exp(-\alpha q_t)\big]}{p_t + (1 - p_t)\exp(-\alpha q_t)} \\
&\overset{(ii)}{\leq} \frac{p_t}{q_t} - 1 - \frac{p_t}{q_t}\big[(0.55)(0.99\alpha)(p_t - q_t) + 0.19\alpha(p_t - q_t)\big] \\
&\overset{(iii)}{\leq} \frac{p_t}{q_t} - 1 - 0.14\alpha\left(\frac{p_t}{q_t} - 1\right) \\
&\leq (1 - 0.14\alpha)\left(\frac{p_t}{q_t} - 1\right), \tag{56}
\end{aligned}
$$

where (i) uses the case $k_t = 1$ of the update rules (42) and (43), (ii) uses $q_t \leq 0.45$, $p_t - q_t \geq 0$, Eq. (55) and the following Eq. (57), (iii) uses $\alpha \leq 10^{-3}$ and $p_t \in [0.2, 0.45]$.

$$\exp(-\alpha q_t) - \exp(-\alpha p_t) \geq \exp(-\alpha p_t)\alpha(p_t - q_t) \geq \alpha(1 - \alpha p_t)(p_t - q_t) \geq 0.99\alpha(p_t - q_t). \tag{57}$$

Similarly, when $p_t < q_t$, we have

$$
\begin{aligned}
&1 - \frac{p_{t+1}}{q_{t+1}} \\
&= 1 - \frac{p_t}{q_t}\left(1 + \frac{(1 - q_t)\big[\exp(-\alpha p_t) - \exp(-\alpha q_t)\big] + (q_t - p_t)\big[1 - \exp(-\alpha q_t)\big]}{p_t + (1 - p_t)\exp(-\alpha q_t)}\right) \\
&\leq (1 - 0.14\alpha)\left(1 - \frac{p_t}{q_t}\right). \tag{58}
\end{aligned}
$$

Now we will integrate the above two cases. Statement $(B_t)$ follows by combining Eqs. (51) and (54) in Cases I and II respectively. Combining Eqs. (52) & (53) in Case I and Eqs. (56) & (58) in Case II, we obtain that Eq. (52) always holds whenever $p_t \geq q_t$ and Eq. (53) always holds whenever $p_t < q_t$. Note that when $\left|\frac{p_t}{q_t} - 1\right| > 5\alpha$, Eq. (46) implies that $\frac{p_t}{q_t} - 1$ and $\frac{p_{t+1}}{q_{t+1}} - 1$ have the same sign. In this case, we can further combine Eqs. (52) and (53) and obtain the following inequality, which proves the statement $(C_t)$.

$$\left|\frac{p_{t+1}}{q_{t+1}} - 1\right| \leq (1 - 0.079\alpha)\left|\frac{p_t}{q_t} - 1\right|.$$

**Proof of the convergence rate for $p_t q_t \to \frac{1}{16}$:**

Next, we will prove that $T_1 := \left\{t : 0 \leq p_t q_t - \frac{1}{16} \leq 6\alpha\right\} \leq \frac{8}{\alpha}$ in the following three cases.

(Case I) If $0 \leq p_0 q_0 - \frac{1}{16} \leq 6\alpha$, then $T_1 = 0$.

(Case II) If $\frac{1}{16} + 6\alpha < p_0 q_0 \leq 0.135$, then we have $\frac{1}{16} + 6\alpha < p_t q_t \leq 0.135$ for all $0 \leq t \leq T_1 - 1$. Otherwise, there must exists $0 \leq t \leq T_1 - 2$ such that $\frac{1}{16} + 6\alpha < p_t q_t \leq 0.135$ and $p_{t+1} q_{t+1} < \frac{1}{16}$, so $\frac{p_{t+1} q_{t+1}}{p_t q_t} < \frac{1/16}{1/16 + 6\alpha} < 1 - 34\alpha$ (since $\alpha \leq 10^{-3}$) which contradicts with Eq. (45). Therefore, $\frac{1}{16} - \frac{\eta}{2} \leq \frac{1}{16} + 6\alpha < p_t q_t \leq 0.135$ for all $0 \leq t \leq T_1 - 1$, so based on the statement $(B_t)$, we have

$$\frac{1}{16} < p_{T_1 - 1} q_{T_1 - 1} \leq \frac{p_0 q_0}{(1 + 0.11\alpha)^{T_1 - 1}} \leq \frac{0.135}{(1 + 0.11\alpha)^{T_1 - 1}},$$

which implies that

$$T_1 \leq 1 + \frac{\ln 2.16}{\ln(1 + 0.11\alpha)} \leq 1 + \frac{7.1}{\alpha} \leq \frac{8}{\alpha},$$

where we use $\alpha \leq 10^{-3}$.

(Case III) If $0.06 \leq p_0 q_0 < \frac{1}{16}$, then similarly we can prove that $0.06 \leq p_t q_t < \frac{1}{16}$ for all $0 \leq t \leq T_1 - 1$. Hence, based on the statement $(B_t)$, we have

$$\frac{1}{16} > p_{T_1-1} q_{T_1-1} \geq \frac{p_0 q_0}{(1 - 0.19\alpha)^{T_1-1}} \geq \frac{0.06}{(1 - 0.19\alpha)^{T_1-1}}, \tag{59}$$

which implies that

$$T_1 \leq 1 + \frac{\ln 0.96}{\ln(1 - 0.19\alpha)} \leq \frac{8}{\alpha},$$

where we use $\alpha \leq 10^{-3}$.

Next, we will prove that $0 \leq p_t q_t - \frac{1}{16} \leq 6\alpha$ for all $t \geq T_1$ via induction. It holds at $t = T_1$ based on the definition of $T_1$. Then suppose $0 \leq p_t q_t - \frac{1}{16} \leq 6\alpha$ holds for a certain $t \geq T_1$ and we will prove that $0 \leq p_{t+1} q_{t+1} - \frac{1}{16} \leq 6\alpha$ in the following two cases.

(Case I) If $3\alpha \leq p_t q_t - \frac{1}{16} \leq 6\alpha$, then on one hand, based on the statement $(B_t)$, we have $p_{t+1} q_{t+1} \leq p_t q_t \leq \frac{1}{16} + 6\alpha$. On the other hand, based on Eq. (45), $p_{t+1} q_{t+1} \geq (1 - 34\alpha) p_t q_t \geq \frac{1}{16}(1 - 0.034) > 0.06$.

(Case II) If $0 \leq p_t q_t - \frac{1}{16} < 3\alpha$, then on one hand, based on the statement $(B_t)$, we have $p_{t+1} q_{t+1} \geq p_t q_t \geq \frac{1}{16}$. On the other hand, based on Eq. (45), $p_{t+1} q_{t+1} \leq (1 + 34\alpha) p_t q_t \leq (1 + 34\alpha)\left(\frac{1}{16} + 3\alpha\right) \leq \frac{1}{16} + 6\alpha$.

As a result, $0 \leq p_t q_t - \frac{1}{16} \leq 6\alpha$ for all $t \geq \frac{8}{\alpha} \geq T_1$.

**Proof of the convergence rate for $\frac{p_t}{q_t} \to 1$:**

Next, we will prove that $T_2 := \left\{t : \left|\frac{p_t}{q_t} - 1\right| \leq 10\alpha\right\} \leq \frac{13}{\alpha} \ln\left(\frac{1}{20\alpha}\right)$. Then based on the statement $(C_t)$, we have

$$10\alpha \leq \left|\frac{p_{T_2-1}}{q_{T_2-1}} - 1\right| \leq (1 - 0.079\alpha)^{T_2-1}\left|\frac{p_0}{q_0} - 1\right| \overset{(i)}{\leq} \frac{1}{2}(1 - 0.079\alpha)^{T_2-1},$$

where (i) uses $\frac{2}{3} \leq \frac{p_0}{q_0} \leq 1.5$. The above inequality along with $\alpha \leq 10^{-3}$ implies that

$$T_2 \leq 1 + \frac{\ln(20\alpha)}{\ln(1 - 0.079\alpha)} \leq \frac{13}{\alpha} \ln\left(\frac{1}{20\alpha}\right).$$

Next, we will prove that $\left|\frac{p_t}{q_t} - 1\right| \leq 10\alpha$ for all $t \geq T_2$ by induction. This holds for $t = T_2$ and suppose that it holds for a certain $t \geq T_2$. Then if $\left|\frac{p_t}{q_t} - 1\right| \leq 5\alpha$, Eq. (46) implies that $\left|\frac{p_{t+1}}{q_{t+1}} - 1\right| \leq \left|\frac{p_t}{q_t} - 1\right| + 4.66\alpha \leq 10\alpha$; Otherwise, if $5\alpha < \left|\frac{p_t}{q_t} - 1\right| \leq 10\alpha$, then the statement $(C_t)$ implies that $\left|\frac{p_{t+1}}{q_{t+1}} - 1\right| \leq \left|\frac{p_t}{q_t} - 1\right| \leq 10\alpha$. Hence, $\left|\frac{p_{t+1}}{q_{t+1}} - 1\right| \leq \left|\frac{p_t}{q_t} - 1\right| \leq 10\alpha$ always holds and thus we have proved that $\left|\frac{p_t}{q_t} - 1\right| \leq 10\alpha$ for all $t \geq \frac{13}{\alpha} \ln\left(\frac{1}{20\alpha}\right) \geq T_2$.

**Obtain the final convergence rates:** Combining the convergence rates for $p_t q_t \to \frac{1}{16}$ and $\frac{p_t}{q_t} \to 1$, we obtain that $0 \leq p_t q_t - \frac{1}{16} \leq 6\alpha$ and $\left|\frac{p_t}{q_t} - 1\right| \leq 10\alpha$ for all $t \geq \frac{13}{\alpha} \ln\left(\frac{1}{20\alpha}\right)$. Therefore, we conclude the proof by providing the ranges of $p_t, q_t$ and the lower bounds of $V_1(\pi_t)$ and $V_2(\pi_t)$ for $t \geq \frac{13}{\alpha} \ln\left(\frac{1}{20\alpha}\right)$ as follows.

$$p_t = \sqrt{p_t q_t \cdot \frac{p_t}{q_t}} \in \left[\sqrt{\frac{1}{16}(1 - 10\alpha)}, \sqrt{\left(\frac{1}{16} + 6\alpha\right)(1 + 10\alpha)}\right] \subseteq \left[\frac{1}{4} - 2\alpha, \frac{1}{4} + 14\alpha\right],$$

$$q_t = \sqrt{p_t q_t \left(\frac{p_t}{q_t}\right)^{-1}} \in \left[\sqrt{\frac{1/16}{1+10\alpha}}, \sqrt{\frac{1/16+6\alpha}{1-10\alpha}}\right] \subseteq \left[\frac{1}{4} - 2\alpha, \frac{1}{4} + 14\alpha\right],$$

where the two $\subseteq$ use $\alpha \leq 10^{-3}$. Therefore, we can prove that $\pi_t$ is feasible as follows.

$$V_1(\pi_t) = 2p_t q_t \geq 2\left(\frac{1}{16}\right) = \xi_1,$$

$$V_2(\pi_t) = 2(1-p_t)(1-q_t) = 2 - 2(p_t + q_t) + 2p_t q_t \overset{(i)}{\geq} 2 - 2\left(\frac{1}{2} + 28\alpha\right) + \frac{1}{8} \overset{(ii)}{>} \frac{1}{8} = \xi_2,$$

where (i) uses $p_t, q_t \geq \frac{1}{4} + 14\alpha$ and $p_t q_t \geq \frac{1}{16}$, and (ii) uses $\alpha \leq 10^{-3}$.

## G    PROOF OF THEOREM 5

Example 2 is equivalent to the following constrained optimization problem

$$\begin{cases} \max\limits_{p,q \in [0,1]} V_0(\pi) := 2pq \\ \text{s.t. } V_1(\pi) := 2pq + 2(1-p)(1-q) \geq 1.8 \end{cases}, \tag{60}$$

which has the unique optimal solution $p = q = 1$.

**Proof for the primal-dual algorithm:** For the problem (60), the Lagrange function (5) can be computed as follows.

$$\begin{aligned} L(\pi, \lambda) &= V_0(\pi) + \lambda_1[V_1(\pi) - \xi_1] \\ &= 2pq + \lambda_1(2pq + 2(1-p)(1-q) - 1.8) \\ &= 2(1+2\lambda_1)pq - 2\lambda_1(p+q) + 0.2\lambda_1 \\ &= 2(1+2\lambda_1)\left(p - \frac{\lambda_1}{1+2\lambda_1}\right)\left(q - \frac{\lambda_1}{1+2\lambda_1}\right) + 0.2\lambda_1 - \frac{2\lambda_1^2}{1+2\lambda_1}. \end{aligned}$$

For all $\lambda_1 > 0$, $\frac{\lambda_1}{1+2\lambda_1} < \frac{1}{2}$, so $\arg\max_{p,q} L(\pi, \lambda) = \{(1,1)\}$. Therefore, the primal-dual algorithm always achieves the optimal solution $p = q = 1$ in the first iteration.

**Proof for the primal algorithm:** In the same way as the proof of item 1 for Example 1, we obtain the update rules of the primal algorithm as follows.

$$p_{t+1} = \begin{cases} \dfrac{p_t}{p_t + (1-p_t)\exp\left(-\alpha q_t\right)}; & \text{if } k_t = 0 \\ \dfrac{p_t}{p_t + (1-p_t)\exp\left(\alpha(1-2q_t)\right)}; & \text{if } k_t = 1 \end{cases} \tag{61}$$

$$q_{t+1} = \begin{cases} \dfrac{q_t}{q_t + (1-q_t)\exp\left(-\alpha p_t\right)}; & \text{if } k_t = 0 \\ \dfrac{q_t}{q_t + (1-q_t)\exp\left(\alpha(1-2p_t)\right)}; & \text{if } k_t = 1 \end{cases}. \tag{62}$$

With initialization $p_0 + q_0 = 1$ and $p_0 \in [0.1, 0.9]$, we will first prove that $p_t + q_t \equiv 1$ by induction. Suppose $p_t + q_t = 1$ holds for a certain $t$. Then $V_1(\pi_t) = 2p_t q_t + 2(1-p_t)(1-q_t) = 4p_t(1-p_t) \leq 1 < \xi_1 = 1.8$. Hence, the case $k_t = 1$ of the update rules (42) and (43) is implemented which implies that

$$\begin{aligned} p_{t+1} + q_{t+1} &= \frac{p_t}{p_t + (1-p_t)\exp\left(\alpha(1-2q_t)\right)} + \frac{q_t}{q_t + (1-q_t)\exp\left(\alpha(1-2p_t)\right)} \\ &= \frac{p_t}{p_t + (1-p_t)\exp\left(\alpha(1-2q_t)\right)} + \frac{1-p_t}{1-p_t + p_t\exp\left(\alpha(2q_t-1)\right)} \\ &= \frac{p_t}{p_t + (1-p_t)\exp\left(\alpha(1-2q_t)\right)} + \frac{(1-p_t)\exp\left(\alpha(1-2q_t)\right)}{(1-p_t)\exp\left(\alpha(1-2q_t)\right) + p_t} = 1. \end{aligned}$$

Hence, $p_t + q_t \equiv 1$, which proves that $V_1(\pi_t) = 2p_t q_t + 2(1-p_t)(1-q_t) = 4p_t(1-p_t) \leq 1 < \xi_1 = 1.8$ for all $t$.

# H EQUIVALENT CONDITION OF $\zeta_k = 0$

**Theorem 6.** $\zeta_k = 0$ *if and only if the Q function has the commonly used factorization structure below (Guestrin et al., 2001; Son et al., 2019; Rashid et al., 2020)*

$$Q_k(\pi; s, a) = \sum_{m=1}^{M} \widetilde{Q}_k^{(m)}(\pi; s, a^{(m)}). \tag{63}$$

*Proof.* **Proof of "if":** Suppose Eq. (63) holds. Then for any $s, a$ and product policy $\pi$, we have

$$\sum_{m=1}^{M} A_k^{(m)}(\pi; s, a^{(m)})$$

$$\overset{(i)}{=} \sum_{m=1}^{M} [Q_k^{(m)}(\pi; s, a^{(m)}) - V_k(\pi; s)]$$

$$\overset{(ii)}{=} \sum_{m=1}^{M} \Big[ \sum_{a^{(\backslash m)}} [\pi^{(\backslash m)}(a^{(\backslash m)}|s) Q_k(\pi; s, a^{(m)})] - V_k(\pi; s) \Big]$$

$$\overset{(iii)}{=} \Big( \sum_{m=1}^{M} \sum_{a^{(\backslash m)}} \pi^{(\backslash m)}(a^{(\backslash m)}|s) \sum_{m'=1}^{M} \widetilde{Q}_k^{(m')}(\pi; s, a^{(m')}) \Big) - MV_k(\pi; s)$$

$$= \sum_{m=1}^{M} \sum_{a^{(\backslash m)}} \pi^{(\backslash m)}(a^{(\backslash m)}|s) \Big( \widetilde{Q}_k^{(m)}(\pi; s, a^{(m)}) + \sum_{m'=1,m'\neq m}^{M} \widetilde{Q}_k^{(m')}(\pi; s, a^{(m')}) \Big) - MV_k(\pi; s)$$

$$= \Big( \sum_{m=1}^{M} \sum_{a^{(\backslash m)}} \pi^{(\backslash m)}(a^{(\backslash m)}|s) \widetilde{Q}_k^{(m)}(\pi; s, a^{(m)}) \Big)$$

$$+ \Big( \sum_{m=1}^{M} \sum_{m'=1,m'\neq m}^{M} \sum_{a^{(m')}} \pi^{(m')}(a^{(m')}|s) \widetilde{Q}_k^{(m')}(\pi; s, a^{(m')}) \Big) - MV_k(\pi; s)$$

$$\overset{(iv)}{=} \Big( \sum_{m=1}^{M} \widetilde{Q}_k^{(m)}(\pi; s, a^{(m)}) \Big) + \Big( \sum_{m'=1}^{M} \sum_{m=1,m\neq m'}^{M} \sum_{a^{(m')}} \pi^{(m')}(a^{(m')}|s) \widetilde{Q}_k^{(m')}(\pi; s, a^{(m')}) \Big)$$

$$- MV_k(\pi; s)$$

$$\overset{(v)}{=} Q_k(\pi; s, a) - V_k(\pi; s) + (M-1)\Big( \sum_{m'=1}^{M} \sum_{a^{(m')}} \pi^{(m')}(a^{(m')}|s) \widetilde{Q}_k^{(m')}(\pi; s, a^{(m')}) \Big)$$

$$- (M-1)V_k(\pi; s)$$

$$\overset{(vi)}{=} A_k(\pi; s, a) + (M-1)\Big( \sum_{m'=1}^{M} \sum_{a^{(m')}} \sum_{a^{(\backslash m')}} \pi(a^{(m')}|s) \pi^{(\backslash m')}(a^{(\backslash m')}|s) \widetilde{Q}_k^{(m')}(\pi; s, a^{(m')}) \Big)$$

$$- (M-1)V_k(\pi; s)$$

$$\overset{(vii)}{=} A_k(\pi; s, a) + (M-1)\Big( \sum_{m'=1}^{M} \sum_{a} \pi(a|s) \widetilde{Q}_k^{(m')}(\pi; s, a^{(m')}) \Big) - (M-1)V_k(\pi; s)$$

$$= A_k(\pi; s, a) + (M-1)\Big( \sum_{a} \pi(a|s) \sum_{m'=1}^{M} \widetilde{Q}_k^{(m')}(\pi; s, a^{(m')}) \Big) - (M-1)V_k(\pi; s)$$

$$\overset{(viii)}{=} A_k(\pi; s, a) + (M-1)\Big( \sum_{a} \pi(a|s) Q_k(\pi; s, a) \Big) - (M-1)V_k(\pi; s)$$

$$\overset{(ix)}{=} A_k(\pi; s, a),$$

where (i) uses the definition of the local advantage function $A_k^{(m)}(\pi; s, a^{(m)})$, (ii) uses the relationship that $Q_k^{(m)}(\pi; s, a^{(m)}) = \sum_{a^{(\backslash m)}} [\pi^{(\backslash m)}(a^{(\backslash m)}|s) Q_k(\pi; s, a^{(m)})]$ where $\pi^{(\backslash m)}(a^{(\backslash m)}|s) := \prod_{m'=1, m' \neq m}^{M} \pi^{(m')}(a^{(m')}|s)$ denotes the policy of all the agents except the agent $m$, which can be seen from the definition of the local Q function $Q_k^{(m)}(\pi; s, a^{(m)}) = \mathbb{E}_\pi \left[ \sum_{t=0}^{\infty} \gamma^t \bar{r}_{k,t} | s_0 = s, a_0^{(m)} = a^{(m)} \right]$ and the global Q function $Q_k^{(m)}(\pi; s, a) = \mathbb{E}_\pi \left[ \sum_{t=0}^{\infty} \gamma^t \bar{r}_{k,t} | s_0 = s, a_0 = a \right]$, (iii), (v) and (viii) use Eq. (63), (iv) uses $\sum_{a^{(\backslash m)}} \pi^{(\backslash m)}(a^{(\backslash m)}|s) = 1$, (vi) uses the definition of the advantage function $A_k(\pi; s, a) := Q_k(\pi; s, a) - V_k(\pi; s)$ and uses $\sum_{a^{(\backslash m')}} \pi^{(\backslash m')}(a^{(\backslash m')}|s) = 1$, (vii) uses $\pi(a^{(m')}|s)\pi^{(\backslash m')}(a^{(\backslash m')}|s) = \pi(a|s)$ for the joint action $a = [a^{(m')}, a^{(\backslash m')}]$, and (ix) uses $V_k(\pi; s) = \sum_a \pi(a|s) Q_k(\pi; s, a)$. This indicates that $\zeta_k = 0$.

**Proof of "only if":** If $\zeta_k = 0$, then $A_k(\pi; s, a) = \sum_{m=1}^{M} A_k^{(m)}(\pi; s, a^{(m)})$. Hence, we can prove Eq. (63) as follows.

$$Q_k(\pi; s, a) = V_k(\pi; s) + A_k(\pi; s, a) = V_k(\pi; s) + \sum_{m=1}^{M} A_k^{(m)}(\pi; s, a^{(m)}) = \sum_{m=1}^{M} \widetilde{Q}_k^{(m)}(\pi; s, a^{(m)}),$$

where $\widetilde{Q}_k^{(m)}(\pi; s, a^{(m)}) := A_k^{(m)}(\pi; s, a^{(m)}) + \frac{1}{M} V_k(\pi; s)$. $\qquad \square$

## I    SUPPORTING LEMMAS

**Lemma 1.** *Any optimal Lagrange multiplier* $\lambda^* \in \arg\min_{\lambda \in \mathbb{R}_+^K} \max_\pi L(\pi, \lambda)$ *satisfies the following range.*

$$\lambda_k^* \leq \frac{1}{2}\lambda_{k,\max} := \frac{1}{\delta_k(1-\gamma)} + \frac{\Delta}{\delta_k}, k = 1, \ldots, K. \tag{64}$$

*Proof.* Use the policy $\widetilde{\pi}$ in Assumption 1, (i.e., $V_k(\widetilde{\pi}) \geq \xi_k + \delta_k$) and denote $\pi^*$ as the optimal solution to the constrained cooperative MARL problem (1). Then we have

$$\frac{1}{1-\gamma} \overset{(i)}{\geq} V_0(\pi^*)$$
$$= \max_\pi \min_{\lambda \in \mathbb{R}_+^K} L(\pi, \lambda)$$
$$\overset{(ii)}{=} \max_\pi L(\pi, \lambda^*) - \Delta$$
$$\geq L(\widetilde{\pi}, \lambda^*) - \Delta$$
$$= V_0(\widetilde{\pi}) + \sum_{k=1}^{K} \lambda_k^* (V_k(\widetilde{\pi}) - \xi_k) - \Delta$$
$$\overset{(iii)}{\geq} \sum_{k=1}^{K} \lambda_k^* \delta_k - \Delta,$$

where (i) and (iii) use $V_k(\pi) \in [0, 1/(1-\gamma)]$ since $\bar{r}_k(s, a) \in [0, 1]$, (ii) uses the definition of the duality gap $\Delta$ in Eq. (6), and (iii) also uses $\lambda_k^* \geq 0$ and $V_k(\widetilde{\pi}) \geq \xi_k + \delta_k$. Since $\lambda_k^*, \delta_k > 0$, the above inequality implies Eq. (64). $\qquad \square$

**Lemma 2.** *For any probability vector* $p \in \mathbb{R}^d$ *(every entry* $p_k \geq 0$ *and* $\sum_{k=1}^{d} p_k = 1$*) and any* $b \in \mathbb{R}^d$*, denote the probability vector* $q \in \mathbb{R}^d$ *with entries* $q_k = \frac{p_k e^{b_k}}{\sum_{j=1}^{d} p_j e^{b_j}}$*. Then the distance between* $p$ *and* $q$ *has the following upper bound.*

$$\|q - p\|_1 := \sum_{k=1}^{d} |q_k - p_k| \leq b_{\max} - b_{\min} \tag{65}$$

*where* $b_{\max} = \max_{1 \leq k \leq d} b_k$ *and* $b_{\min} = \min_{1 \leq k \leq d} b_k$.

*Proof.* For $t \in [0, 1]$ and $k = 1, 2, \ldots, d$, define the following function

$$v_k(t) = \frac{p_k e^{tb_k}}{\sum_{j=1}^d p_j e^{tb_j}}, \tag{66}$$

which has the following derivative bound.

$$
\begin{aligned}
|v_k'(t)| &= \left| \frac{p_k b_k e^{tb_k} \sum_{j=1}^d p_j e^{tb_j} - p_k e^{tb_k} \sum_{j=1}^d p_j b_j e^{tb_j}}{(\sum_{j=1}^d p_j e^{tb_j})^2} \right| \\
&= \frac{p_k e^{tb_k} |\sum_{j=1}^d p_j (b_k - b_j) e^{tb_j}|}{(\sum_{j=1}^d p_j e^{tb_j})^2} \\
&\leq \frac{p_k e^{tb_k} \sum_{j=1}^d p_j |b_k - b_j| e^{tb_j}}{(\sum_{j=1}^d p_j e^{tb_j})^2} \\
&\leq \frac{p_k e^{tb_k} (b_{\max} - b_{\min}) \sum_{j=1}^d p_j e^{tb_j}}{(\sum_{j=1}^d p_j e^{tb_j})^2} = \frac{p_k e^{tb_k} (b_{\max} - b_{\min})}{\sum_{j=1}^d p_j e^{tb_j}}
\end{aligned} \tag{67}
$$

As a result,

$$\sum_{k=1}^d |q_k - p_k| = \sum_{k=1}^d |v_k(1) - v_k(0)| = \sum_{k=1}^d \left| \int_0^1 v_k'(t) dt \right| \leq \int_0^1 \sum_{k=1}^d |v_k'(t)| dt \leq b_{\max} - b_{\min}.$$

$\square$

Next, we change initial state distribution $\rho$ to be any state distribution $\rho'$, and replace the value function $V_k(\pi)$ (defined in Eq. (1)) and occupation measure $\nu_{t+1} := \nu_{\pi_{t+1}}$ (defined in Eq. (2)) with $V_{k;\rho'}(\pi)$ and $\nu_{t+1;\rho'}$ respectively to emphasis their dependence on $\rho'$.

**Lemma 3.** *The policy $\pi_t$ and index $k_t$ generated from Algorithm 2 satisfy the following bounds for any state $s \in \mathcal{S}$.*

$$\sum_{m=1}^M \sum_{a^{(m)}} |\pi_{t+1}^{(m)}(a^{(m)}|s) - \pi_t^{(m)}(a^{(m)}|s)| \leq M\alpha \left( \frac{1}{1-\gamma} + 2\epsilon_3 \right) \tag{68}$$

$$V_{k_t}(\pi_{t+1}; \rho') - V_{k_t}(\pi_t; \rho') \leq \frac{M\alpha}{(1-\gamma)^2} + \frac{2M\alpha\epsilon_3}{1-\gamma} \tag{69}$$

*Proof.* First, consider two MDPs $\{S_i, A_i\}_i$, $\{S_i', A_i'\}_i$ following the same transition kernel $\mathcal{P}$ and policies $\pi_t$ and $\pi_{t+1}$ respectively. Then the state transition distribution of the two MDPs are respectively $p(s'|s) = P(S_{i+1} = s'|S_i = s) = \sum_a \mathcal{P}(s'|s, a)\pi_t(a|s)$ and $p'(s'|s) = P(S_{i+1}' = s'|S_i' = s) = \sum_a \mathcal{P}(s'|s, a)\pi_{t+1}(a|s)$ respectively. Denote $p_i$ and $p_i'$ as the distribution of $S_i$ and $S_i'$ respectively under the same initial distribution $p_0 = p_0' = \rho'$. Then we have

$$
\begin{aligned}
\|p_{i+1}' - p_{i+1}\|_1 &= \sum_{s'} |p_{i+1}'(s') - p_{i+1}(s')| \\
&= \sum_{s'} \left| \sum_s \left( p'(s'|s)p_i'(s) - p(s'|s)p_i(s) \right) \right| \\
&\leq \sum_{s'} \left| \sum_s p_i'(s)\left( p'(s'|s) - p(s'|s) \right) \right| + \sum_{s'} \left| \sum_s p(s'|s)\left( p_i'(s) - p_i(s) \right) \right| \\
&\leq \sum_{s'} \sum_s p_i'(s)|p'(s'|s) - p(s'|s)| + \sum_{s'} \sum_s p(s'|s)|p_i'(s) - p_i(s)| \\
&= \sum_s p_i'(s) \sum_a \sum_{s'} \mathcal{P}(s'|s, a)|\pi_{t+1}(a|s) - \pi_t(a|s)| + \|p_i' - p_i\|_1 \\
&\leq \max_s \|\pi_{t+1}(\cdot|s) - \pi_t(\cdot|s)\|_1 + \|p_i' - p_i\|_1. \tag{70}
\end{aligned}
$$

Since $p_0' = p_0$, iterating the above inequality yields that

$$\|p_i' - p_i\|_1 \leq i \max_s \|\pi_{t+1}(\cdot|s) - \pi_t(\cdot|s)\|_1. \tag{71}$$

Hence, the state occupation measure difference can be upper bounded as follows.

$$\|\nu_{t+1;\rho'}(\cdot) - \nu_{t;\rho'}(\cdot)\|_1 \leq (1-\gamma) \sum_{i=0}^{\infty} \gamma^i \|p_i' - p_i\|_1$$

$$\overset{(i)}{\leq} (1-\gamma) \max_s \|\pi_{t+1}(\cdot|s) - \pi_t(\cdot|s)\|_1 \sum_{i=0}^{\infty} i\gamma^i$$

$$\overset{(ii)}{=} \frac{\gamma}{1-\gamma} \max_s \|\pi_{t+1}(\cdot|s) - \pi_t(\cdot|s)\|_1, \tag{72}$$

where (i) uses Eq. (71) and (ii) uses the fact that the function $f(\gamma) = \sum_{i=0}^{\infty} \gamma^i = (1-\gamma)^{-1}$ has the following derivative

$$f'(\gamma) = \sum_{i=0}^{\infty} i\gamma^{i-1} = (1-\gamma)^{-2}. \tag{73}$$

Therefore, the state action occupation measure difference can be bounded as follows.

$$\|\nu_{t+1;\rho'}(\cdot,\cdot) - \nu_{t;\rho'}(\cdot,\cdot)\|_1$$

$$= \sum_{s,a} |\nu_{t+1;\rho'}(s)\pi_{t+1}(a|s) - \nu_{t;\rho'}(s)\pi_t(a|s)|$$

$$\leq \sum_{s,a} \nu_{t+1;\rho'}(s)|\pi_{t+1}(a|s) - \pi_t(a|s)| + \sum_{s,a} \pi_t(a|s)|\nu_{t+1;\rho'}(s) - \nu_{t;\rho'}(s)|$$

$$\leq \sum_s \nu_{t+1;\rho'}(s)\|\pi_{t+1}(\cdot|s) - \pi_t(\cdot|s)\|_1 + \|\nu_{t+1;\rho'}(\cdot) - \nu_{t;\rho'}(\cdot)\|_1$$

$$\overset{(i)}{\leq} \max_s \|\pi_{t+1}(\cdot|s) - \pi_t(\cdot|s)\|_1 + \frac{\gamma}{1-\gamma} \max_s \|\pi_{t+1}(\cdot|s) - \pi_t(\cdot|s)\|_1$$

$$= \frac{1}{1-\gamma} \max_s \|\pi_{t+1}(\cdot|s) - \pi_t(\cdot|s)\|_1, \tag{74}$$

where (i) uses Eq. (72).

To bound the policy difference $\|\pi_{t+1}(\cdot|s) - \pi_t(\cdot|s)\|_1$, we rewrite the NPG rule (11) as follows

$$Z_t^{(m)}(s) = \sum_{a'^{(m)}} \pi_t^{(m)}(a'^{(m)}|s) \exp\left(\alpha \widehat{Q}_{k_t}^{(m)}(\pi_t; s, a'^{(m)})\right), \tag{75}$$

$$\pi_{t+1}^{(m)}(a^{(m)}|s) = \frac{\pi_t^{(m)}(a^{(m)}|s)}{Z_t^{(m)}(s)} \exp\left(\alpha \widehat{Q}_{k_t}^{(m)}(\pi_t; s, a^{(m)})\right). \tag{76}$$

Therefore,

$$\|\pi_{t+1}(\cdot|s) - \pi_t(\cdot|s)\|_1$$

$$= \sum_a \left| \prod_{m=1}^M \pi_{t+1}^{(m)}(a^{(m)}|s) - \prod_{m=1}^M \pi_t^{(m)}(a^{(m)}|s) \right|$$

$$\overset{(i)}{\leq} \sum_a \sum_{m'=1}^M \left| \prod_{m=1}^{m'} \pi_{t+1}^{(m)}(a^{(m)}|s) \prod_{m=m'+1}^M \pi_t^{(m)}(a^{(m)}|s) - \prod_{m=1}^{m'-1} \pi_{t+1}^{(m)}(a^{(m)}|s) \prod_{m=m'}^M \pi_t^{(m)}(a^{(m)}|s) \right|$$

$$= \sum_{m'=1}^M \sum_a \left( \prod_{m=1}^{m'-1} \pi_{t+1}^{(m)}(a^{(m)}|s) \prod_{m=m'+1}^M \pi_t^{(m)}(a^{(m)}|s) \right) |\pi_{t+1}^{(m')}(a^{(m')}|s) - \pi_t^{(m')}(a^{(m')}|s)|$$

$$\overset{(ii)}{=} \sum_{m=1}^M \sum_{a^{(m)}} |\pi_{t+1}^{(m)}(a^{(m)}|s) - \pi_t^{(m)}(a^{(m)}|s)|$$

$$\overset{(iii)}{\leq} \sum_{m=1}^{M} \alpha \Big( \max_{a^{(m)}} \widehat{Q}_{k_t}^{(m)}(\pi_t; s, a^{(m)}) - \min_{a^{(m)}} \widehat{Q}_{k_t}^{(m)}(\pi_t; s, a^{(m)}) \Big)$$

$$\leq \alpha \sum_{m=1}^{M} \Big( \max_{a^{(m)}} Q_{k_t}^{(m)}(\pi_t; s, a^{(m)}) - \min_{a^{(m)}} Q_{k_t}^{(m)}(\pi_t; s, a^{(m)})$$

$$+ 2 \max_{a^{(m)}} |\widehat{Q}_{k_t}^{(m)}(\pi_t; s, a^{(m)}) - Q_{k_t}^{(m)}(\pi_t; s, a^{(m)})| \Big)$$

$$\overset{(iv)}{\leq} M\alpha \Big( \frac{1}{1-\gamma} + 2\epsilon_3 \Big), \tag{77}$$

where (i) uses the following relation for any joint action $a$ where $C_{m'}(a) := \prod_{m=1}^{m'} \pi_{t+1}^{(m)}(a^{(m)}|s) \prod_{m=m'+1}^{M} \pi_t^{(m)}(a^{(m)}|s)$, (ii) and (iv) prove Eq. (68), (iii) applies Lemma 2 where the $a^{(m)}$-th entries of vectors $p, b, q \in \mathbb{R}^{|\mathcal{A}^{(m)}|}$ are $\pi_t^{(m)}(a^{(m)}|s)$, $\alpha\widehat{Q}_{k_t}^{(m)}(\pi_t; s, a^{(m)})$ and $\pi_{t+1}^{(m)}(a^{(m)}|s)$ respectively, and (iii) uses $Q_k^{(m)}(\pi; s, a^{(m)}) \in [0, 1/(1-\gamma)]$ since $\overline{r}_{k,t} \in [0, 1]$.

$$|C_M(a) - C_0(a)| = \Big| \sum_{m'=1}^{M} [C_{m'}(a) - C_{m'-1}(a)] \Big| \leq \sum_{m'=1}^{M} |C_{m'}(a) - C_{m'-1}(a)|.$$

As a result, Eq. (69) can be proved as follows.

$$|V_{k_t}(\pi_{t+1}; \rho') - V_{k_t}(\pi_t; \rho')|$$

$$= \Big| \sum_{s,a} \overline{r}_{k_t}(s, a) \big[ \nu_{t+1;\rho'}(s, a) - \nu_{t;\rho'}(s, a) \big] \Big|$$

$$\overset{(i)}{\leq} \frac{1}{1-\gamma} \| \nu_{t+1;\rho'}(\cdot, \cdot) - \nu_{t;\rho'}(\cdot, \cdot) \|_1$$

$$\overset{(ii)}{\leq} \frac{1}{(1-\gamma)^2} \max_{s} \| \pi_{t+1}(\cdot|s) - \pi_t(\cdot|s) \|_1$$

$$\overset{(iii)}{\leq} \frac{M\alpha}{(1-\gamma)^3} + \frac{2M\alpha\epsilon_3}{(1-\gamma)^2}$$

where (i) uses $\overline{r}_{k_t}(s, a) \in [0, 1]$, (ii) uses Eq. (74) and (iii) uses Eq. (77). $\qquad \square$

## J COMPARISON OF CONVERGENCE RESULTS ON CONSTRAINED COOPERATIVE MARL

Table 1: Comparison of Convergence Results on Constrained Cooperative MARL

| Works | Algorithm | Assumptions | Convergence measure |
|---|---|---|---|
| Lu et al. (2021) | primal-dual | bounded reward, Lipschitz continuity, Slater's condition | gradient |
| Ying et al. (2023) | primal-dual | bounded reward, Lipschitz continuity, bounded optimal Lagrange multiplier[2] | gradient |
| Yang et al. (2023) | primal-dual | Fixing base policy, perturbation policy in compact convex space Lipschitz continuity | convergence of perturbation policy |
| Algorithm 1 (Ours) | primal-dual | bounded reward Slater's condition | constraint violation optimality gap |
| Algorithm 2 (Ours) | primal | bounded reward | constraint violation optimality gap |

## K EXPERIMENT ON CONSTRAINED GRID-WORLD

We slightly adapt the constrained grid-world task (Diddigi et al., 2019) where two agents explore the $4 \times 4$ grid-world in Figure 2. The agents start from position 3 and aim at the target 11. Both agents can observe their positions and accordingly select to move up, down, left or right. If an agent $m$ has reached the destination (target 11), then it will always stay there and obtains reward $r_{0,t}^{(m)} = 0$ regardless of the selected action. If an agent is at a non-target marginal grid and the action points outside the grid, then the agent stays there and obtains reward -5 (For example, an agent will stay at position 7 if it selects to move right.). In all the other cases, the agent moves one step and obtains reward -1. The safety score $r_{1,t}^{(m)} = -1$ for both agents $m = 1, 2$ if they collide at a non-target position (including initial position 3). Otherwise, $r_{1,t}^{(m)} = 0$. The discount factor is $\gamma = 0.9$ and the safety threshold is $\xi_1 = -1$, which allows no collision between the agents except at the initial time. Therefore, the optimal solution is to let the agents deterministically select the two

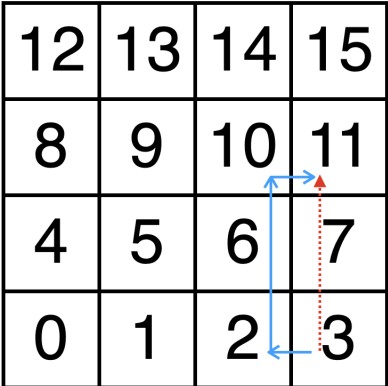

Figure 2: Constrained grid-world.

paths shown in Figure 2 respectively with $V_0(\pi) = -2.6695$ and $V_1(\pi) = -1$, which indicates that this problem has zero duality gap.

We compare the non-stochastic versions of the primal-dual algorithm (Algorithm 1), the primal algorithm (Algorithm 2) and the centralized nested actor-critic (CNAC) algorithm (Diddigi et al., 2019) on this constrained grid-world task where transition kernel and reward/safety score functions are available. Specifically, in Algorithm 1, we use 50 value iterations to obtain the greedy policy $\pi_t$, exactly evaluate $\widehat{V}_k(\pi_t) = V_k(\pi_t) = \frac{1}{1-\gamma} \sum_{s,a} \overline{r}_0(s,a) \nu_{\pi_t}(s,a)$ where the occupation measure $\nu_{\pi_t}(s,a)$ is known to be the stationary distribution of the mixed transition kernel $\mathcal{P}_\rho(\cdot|s,a) := \gamma \mathcal{P}(\cdot|s,a) + (1-\gamma)\rho(\cdot)$, and update the multipliers with stepsize $\beta = 1$ and threshold $\lambda_{1,\max} = 10$. In Algorithm 2, we also exactly evaluate $\widehat{V}_k(\pi_t) = V_k(\pi_t)$ and $\widehat{Q}_{k_t}^{(m)}(\pi_t; s, a^{(m)}) = Q_{k_t}^{(m)}(\pi_t; s, a^{(m)})$, and select stepsize $\alpha = 1$ and tolerance $\eta = 10^{-3}$. The CNAC algorithm essentially follows the primal-dual framework (Algorithm 1) except that the policy $\pi_t$ is updated with one projected

stochastic policy gradient ascent step as follows.

$$\pi \leftarrow \text{Proj}_{\mathcal{V}_p}\left[\pi + \alpha\widehat{\nabla}_\pi L(\pi, \lambda_t)\right].$$

Here, $\mathcal{V}_p$ is the product policy space, and we use the exact policy gradient $\widehat{\nabla}_\pi L(\pi, \lambda_t) = \nabla_\pi L(\pi, \lambda_t)$ and select stepsize $\alpha = 0.2$. The update rule of the multipliers for the CNAC algorithm is the same as that for our primal-dual algorithm.

We implement these algorithms for 100 iterations. The initial policy of each agent at each state is randomly generated from Dirichlet distribution $\text{Dir}(1, 1, 1, 1)$. We plot the learning curves of $V_0(\pi_t)$ and $V_1(\pi_t)$ in Figure 3. It can be seen from Figure 3 that all these algorithms converge fast to the feasible region $V_1(\pi_t) \geq -1$ within 10 iterations. As to optimality, our primal-dual algorithm and primal algorithm converge to the optimal value $V_0(\pi_t) = -2.6695$ within 3 iterations and 70 iterations respectively. The CNAC algorithm converges to a sub-optimal value $V_0(\pi_t) \approx -4.5$ within 10 iterations, since it uses policy gradient ascent update which may stuck at a stationary point.

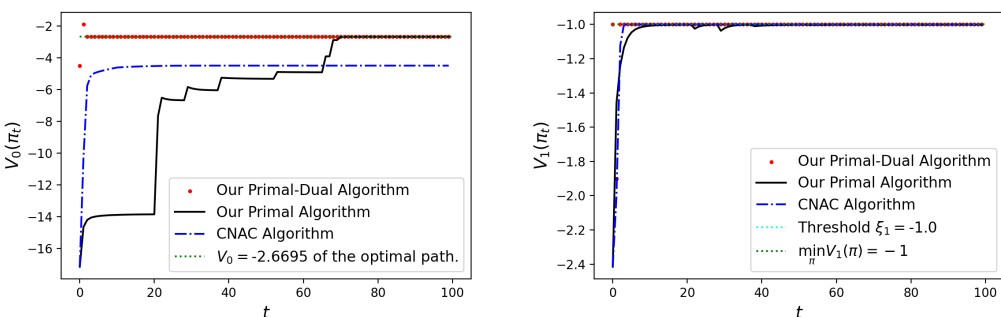

Figure 3: Results on the constrained grid task with constraint $V_1(\pi_t) \geq -1$.

Furthermore, we decrease the threshold $\xi_1$ to $-1.1$, where the deterministic paths in Figure 2 become near-optimal and **the duality gap becomes nonzero**. We implement these algorithms for 100 iterations using the same initial policy as that for the threshold $\xi_1 = -1$. Our primal-dual algorithm uses stepsize $\beta = 1$ and 50 value iterations. Our primal algorithm uses stepsize $\alpha = 0.4$ and tolerance $\eta = 10^{-3}$. The CNAC algorithm uses stepsizes $\alpha = 0.8$ and $\beta = 1$. From the result in Figure 4, we can see that all the algorithms become less stable in the constrained-related value $V_1(\pi_t)$ and occasionally falls below the threshold $-1.1$ due to the nonzero duality gap. Regarding the objective $V_0(\pi_t)$, our primal-dual algorithm and primal algorithm converge to the near-optimal value, and primal-dual converges faster, but CNAC converges to a lower sub-optimal value.

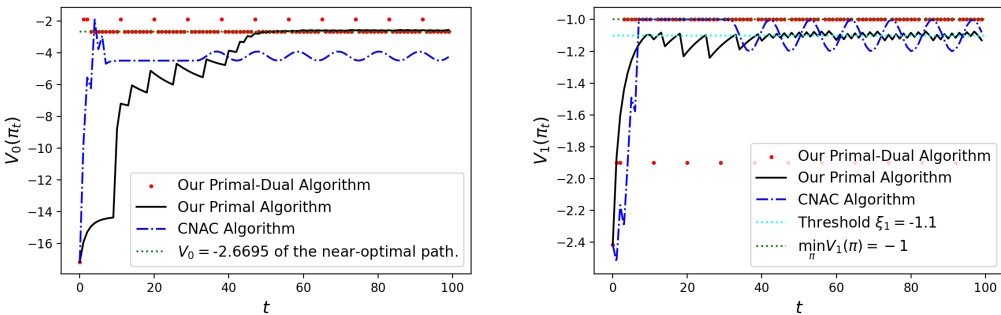

Figure 4: Results on the constrained grid task with constraint $V_1(\pi_t) \geq -1.1$.

