# OpenReview forum: "On the Hardness of Constrained Cooperative Multi-Agent Reinforcement Learning"
_ICLR.cc/2024/Conference — ICLR 2024 poster_

### Official Review · Reviewer_fuRV · 2023-10-29

**Soundness:** 3 good
**Presentation:** 3 good
**Contribution:** 2 fair
**Rating:** 5
**Confidence:** 3

**Summary:**

The paper studies a constrained MARL problem, where a group of agents operate independently in an MDP and jointly optimize a reward function. There are safty constraints that must be satisfied by their joint policy as well. The first focus of the paper is on the hardness of the problem, where the authors showed that strong duality does not hold. The authors then propose a decentralized primal approach to solve the problem. Via examples, it is illustrated either the decentralized primal algorithm or the primal-dual algorithm outperform the other in all cases.

**Strengths:**

The problem studied is interesting and relevant to the topics of ICLR. The authors made efforts to derive rigorous analyses of the problem.

**Weaknesses:**

From a computational perspective, the statements regarding the hardness of the studied problem are rather loose. Theorem 1 shows that the problem can be reduced to an optimization problem with quadratic constraints, which is in general hard. But nevertheless this doesn't imply the constrained MARL problem is also hard (for that requires a reduction in the other direction, i.e., quadratic optimization can be reduced to the studied MARL problem). Statements such as "some studies argued that it is **probably** an NP-complete problem" and  "Thus, constrained cooperative MARL is a hard problem due to the presence of safety and product policy constraints" are very loose. It also unclear whether the hardness comes from the safety constraints or the product constraints. Results in the subsequent sections do not seem to provide any clear message regarding the computational complexity of the problem, nor the time complexity of the algorithms discussed. Overall, the insights provided are a bit limited.

**Questions:**

Is the problem known to be hard without the safety constraints (with only the product constraints)? How much is known about this in the literature?

---

> ### Author Response · Authors · 2023-11-18
> **Authors' reply to Reviewer fuRV**
>
> Thank you very much for reviewing our manuscript and providing valuable feedback. Below is a response to the review comments. We have submitted a revised version with all revisions marked in **"red"**. Please let us know if further clarifications are needed.
>
> **Q1:** The hardness results are loose.
>
> **A:** Thanks for pointing this out.
> We realize that our statement of hardness is not consistent with the actual contributions of the paper, as also pointed out by Reviewer WbB4. Therefore, in the revised paper, we follow the reviewers' suggestions and have changed the emphasis of our claim. Specifically, we changed "hardness" to "duality gap" in the title, and changed it to "challenging" or "challenges" throughout the paper. We also changed the emphasis of our main contribution, focusing on the nonzero duality gap and development of algorithms and their convergence results.
> We agree that it is challenging to figure out the specific complexity class of such a problem, and we think this is an interesting and fundamental problem deserving future research.
>
> **Q2:** Does the hardness come from the safety constraints or the product constraints? Is the problem known to be hard without the safety constraints (with only the product constraints)? How much is known about this in the literature?
>
> **A:** Great question. The hardness comes from both safety constraints and the product constraints. This is because if we remove either the safety constraints or the product constraints, the constrained cooperative MARL reduces to either cooperative MARL or single-agent constrained RL, both having polynomial time algorithms. We have clarified this in the second paragraph of Section 1.1 and at the end of Section 2.
>
> **Q3:** Computational complexity and time complexity of the algorithms are not discussed.
>
> **A:** Thanks for the great suggestion. We have added the complexity results to Theorems 3 and 4 in the revised paper. In these results we derive sample complexity (the total number of required samples), as it is a standard metric used in finite-time analysis of RL algorithms and is also proportional to computation and time complexities. We sketch the derivation as follows.
>
> The sample complexity of Algorithm 1 (Primal-Dual algorithm) is $\mathcal{O}(\epsilon^{-5}\ln\epsilon^{-1})$. To elaborate, based on Theoren 3, Algorithm 1 takes $T=\mathcal{O}(\epsilon^{-2})$ iterations, and each iteration requires $\mathcal{O}(\epsilon_1^{-3}\ln\epsilon_1^{-1})$ and $\mathcal{O}(\epsilon_2^{-2})$ samples to obtain $\pi_t$ and $\widehat{V}_k(\pi_t)$ with precisions $\epsilon_1=\mathcal{O}(T^{-1/2})=\mathcal{O}(\epsilon)$ and $\epsilon_2=\mathcal{O}(T^{-1/2})=\mathcal{O}(\epsilon)$ respectively [1,2]. Therefore, the sample complexity of Algorithm 1 is
> $$
> T\mathcal{O}(\epsilon_1^{-3}\ln\epsilon_1^{-1}+\epsilon_2^{-2})=\mathcal{O}(\epsilon^{-2})\mathcal{O}(\epsilon^{-3}\ln\epsilon^{-1}+\epsilon^{-2})=\mathcal{O}(\epsilon^{-5}\ln\epsilon^{-1}).
> $$
>
> The sample complexity of Algorithm 2 (Primal algorithm) is $\mathcal{O}[T(\epsilon_2^{-2}+\epsilon_3^{-2})]=\mathcal{O}(\epsilon^{-4})$. To elaborate, based on Theorem 4, Algorithm 2 takes $T=\mathcal{O}(\epsilon^{-2})$ iterations, and each iteration requires $\mathcal{O}(\epsilon_2^{-2})$ and $\mathcal{O}(\epsilon_3^{-2})$ samples to obtain $\widehat{V} _ k(\pi_t)$ and $\widehat{Q } _{k_t}^{(m)}(\pi_t;s,a^{(m)})$ with precisions $\epsilon_2=\mathcal{O}(T^{-1/2})=\mathcal{O}(\epsilon)$ and $\epsilon_3=\mathcal{O}(T^{-1/2})=\mathcal{O}(\epsilon)$ respectively [2]. Therefore,  the sample complexity of Algorithm 2 is
> $$
> T\mathcal{O}(\epsilon_2^{-2}+\epsilon_3^{-2})=\mathcal{O}(\epsilon^{-2})\mathcal{O}(\epsilon^{-2}+\epsilon^{-2})=\mathcal{O}(\epsilon^{-4}).
> $$
>
> [1] Chen, Z., Zhou, Y., Chen, R. R., \& Zou, S. (2022). Sample and communication-efficient decentralized actor-critic algorithms with finite-time analysis. In International Conference on Machine Learning (pp. 3794-3834). PMLR.
>
> [2] Li, G., Wei, Y., Chi, Y., Gu, Y., \& Chen, Y. (2020). Breaking the sample size barrier in model-based reinforcement learning with a generative model. Advances in neural information processing systems, 33, 12861-12872.

---

> > ### Comment · Reviewer_fuRV · 2023-11-22
> > **Thank you for your feedback**
> >
> > Thank you for your clarification. I agree that it's better to change "hardness" to "duality gap" in the title.

---

> > > ### Author Response · Authors · 2023-11-22
> > > **Authors' 2nd reply to Reviewer FuRV**
> > >
> > > Thank you for your response. We are glad that our revision addresses your concern. Since your major concern is the wording about the ``hardness'' and we have revised our statement throughout the paper, we kindly ask the reviewer to consider raising the rating  based on the revision.

---

### Official Review · Reviewer_2Wr4 · 2023-10-30

**Soundness:** 3 good
**Presentation:** 3 good
**Contribution:** 3 good
**Rating:** 6
**Confidence:** 2

**Summary:**

This paper studies the hardness of constrained cooperative multi-agent reinforcement learning. In particular, it argues that there is a strictly positive duality gap. In addition, neither primal-dual or primal algorithms are strictly better than the other one.

**Strengths:**

+ The paper addresses an important question.
+ The paper is well-written.
+ The results seem correct and the explanations and theorems make intuitive sense.

**Weaknesses:**

- The bound $\Delta$ seems to be trivial? It just comes from the geometric sum and assuming that the risk at each step is less than 1?
- I'm not sure I see the only if part of Theorem 7.
- It would be good to offer some advice on when to use which type of algorithm.
- Some simulation would be helpful to illustrate the results.

**Questions:**

The paper seems to want to say that the problem is NP-hard but stops short. It seems that the quadratic equality is more of an analogy rather than equivalence to standard optimization problems. It would be good to make this more precise.

---

> ### Author Response · Authors · 2023-11-18
> **Authors' reply to Reviewer 2Wr4**
>
> Thank you very much for reviewing our manuscript and providing valuable feedback. Below is a response to the review comments. We have submitted a revised version with all revisions marked in **"red"**. Please let us know if further clarifications are needed.
>
> **Q1:** The bound seems to be trivial? It just comes from the geometric sum and assuming that the risk at each step is less than 1?
>
> **A:** Yes, the simple bound $\Delta\le \frac{1}{1-\gamma}$ directly comes from the geometric sum and the assumption that the risk at each step is less than 1. In the revision, we have moved this result to the remark after Theorem 2. The main result of Theorem 2 is to prove the non-zero duality gap.
>
> **Q2:** I'm not sure I see the only if part of Theorem 7.
>
> **A:** Thanks for pointing this out. In the revised paper, we have added multiple bold subtitles to highlight the parts of "if" and "only if" in the proof of Theorem 7.
>
> **Q3:** It would be good to offer some advice on when to use which type of algorithm.
>
> **A:** Thanks for the suggestion. When the Q function can be approximated by a factorized form, the primal algorithm is preferred since the advantage gaps $\zeta_k$ in the convergence rate are small. Otherwise, there is no significant difference between these two algorithms based on our current understanding.
>
> **Q4:** Some simulation would be helpful to illustrate the results.
>
> **A:** Thanks for the suggestion. We are currently conducting MARL experiments in a real world environment and will keep the reviewer posted once the results are available.
>
> **Q5:** The paper seems to want to say that the problem is NP-hard but stops short. It seems that the quadratic equality is more of an analogy rather than equivalence to standard optimization problems. It would be good to make this more precise.
>
> **A:** Thanks for the great suggestion. We agree that it is challenging to figure out the specific complexity class of such a problem, and we think this is an interesting and fundamental problem deserving future research. We also realize that our statement of hardness does not reflect the actual contribution of the paper, as also pointed out by Reviewer WbB4. Therefore, in the revised paper, we have changed the emphasis
> of our claim. Specifically, we changed "hardness" to "duality gap" in the title, and changed it to "challenging" or "challenges" throughout the paper. We also changed the emphasis of our main contribution, focusing on the nonzero duality gap and development of algorithms and their convergence results.
>
> Regarding the quadratic equality, our Theorem 1 strictly proved that the original RL problem (1) on the policy $\pi$ is equivalent to the optimization problem (4) on the occupation measure $\nu$. So this is not an analogy.

---

### Official Review · Reviewer_MmRm · 2023-11-02

**Soundness:** 3 good
**Presentation:** 3 good
**Contribution:** 2 fair
**Rating:** 6
**Confidence:** 4

**Summary:**

The paper studies cooperative multi-agent RL problems in which all agents aim to maximize the average reward value function subject to a constraint on the average safety value functions. The authors provide a counter-example to show that the strong duality fails. Nevertheless, the authors extend two existing constrained policy search methods in the single-agent setting to cooperative multi-agent RL under constraints and prove their non-asymptotic optimality gap and constrained violation error bounds. The authors also investigate the pros and cons of the two methods in numerical examples.

**Strengths:**

- The paper is well written, and statements are supported by justifications.

- The authors show the existence of a strict duality gap in a simple example of cooperative constrained MDPs. This structural property reveals the limitations of methods in the single-agent case, which is useful for developing new algorithms.

- The authors present a primal-dual algorithm and investigate its limitation by revealing the dependence of error bounds on the duality gap.

- The authors also present a primal algorithm that works in a decentralized way, and provide finite-time error bounds on the optimality gap and constraint violation.

- Numerical examples are provided to show that either one can perform better than the other.

**Weaknesses:**

- For the duality, the authors didn't discuss the connection to the duality of constrained Markov potential games: Provably Learning Nash Policies in Constrained Markov Potential Games. Since constrained cooperative Markov games are a particular case, a non-zero duality gap directly follows.

- Due to the non-zero duality gap, the proposed two algorithms suffer some gaps caused by the multi-agent and constraint coupling. It is more expected that the primal-dual algorithm suffers a duality gap. The primal algorithm has a dependence on an advantage gap that is less expected. It is not clear if these gaps are necessary and which one is better.

- It is interesting to check the policy iterate convergence of two algorithms in simple examples. If this can be proved for the algorithms under certain conditions, it would be more beneficial to guide the practice.

- Experiments are done with artificial examples. It is favorable to check the performance of the two actor-critic algorithms for solving real constrained tasks, and compare it with existing methods as mentioned.

**Questions:**

- Is the non-product form of optimal policy the only reason for the duality gap? Does there exist a more fundamental metric that can characterize the cause of the duality gap?

- As shown in convergence analysis, the effect of the duality gap in different algorithms can be different. Does this suggest a better way to design algorithms? Is it possible to remove such gap dependence?

- The advantages of the two algorithms are discussed in terms of policy iterates, which are stronger than the output policy in algorithms. Is it possible to show them in convergence theory?

- As mentioned, prior works also studied constrained cooperative Markov games and the authors have improved the analysis. Can the authors illustrate the analysis differences due to the lack of zero duality gap? It is useful if the authors could compare assumptions and results with them in a table.

---

> ### Author Response · Authors · 2023-11-18
> **Authors' reply to Reviewer MmRm (Q1-Q4)**
>
> Thank you very much for reviewing our manuscript and providing valuable feedback. Below is a response to the review comments. We have submitted a revised version with all revisions marked in **"red"**. Please let us know if further clarifications are needed.
>
> **Q1:** For the duality, the authors didn't discuss the connection to the duality of constrained Markov potential games: Provably Learning Nash Policies in Constrained Markov Potential Games. Since constrained cooperative Markov games are a particular case, a non-zero duality gap directly follows.
>
> **A:** Thank you very much for pointing out this related work [1], which studies constrained Markov potential game with competing agents.
> In the case where all the agents share the same reward function $r_0$, the potential function $\Phi$ of the constrained Markov potential game reduces to the value function $V_0$ associated with $r_0$. Thus, we agree that the non-zero duality gap result in their Proposition 3 also applies to our case. In the revision, we have cited this work and added a remark after Theorem 2 to mention the related results.
>
> We also want to highlight the differences between our work and this related work [1]. To elaborate, their work focuses on Nash equilibrium of constrained Markov potential game, which is very different from the optimal product policy of constrained cooperative MARL studied in our work. More specifically, in the special case where all the agents share the same reward (i.e., cooperative case), their Nash equilibrium
> corresponds to a feasible product policy $\pi$ where each agent's policy $\pi^{(m)}$ is optimal fixing all the other agents' policies $\pi^{(\backslash m)}$. Such a notion of agent-wise optimal product policy is weaker than the global optimal product policy studied in our work.
>
> [1] Alatur, P., Ramponi, G., He, N., \& Krause, A. (2023). Provably Learning Nash Policies in Constrained Markov Potential Games. ArXiv:2306.07749.
>
> **Q2:** Experiments are done with artificial examples. It is favorable to check the performance of the two actor-critic algorithms for solving real constrained tasks, and compare it with existing methods as mentioned.
>
> **A:** Thanks for the suggestion. We are currently conducting MARL experiments in a real constrained task
> and will keep the reviewer posted once the results are available.
>
> **Q3:** Is the product form of optimal policy the only reason for the nonzero duality gap? Does there exist a more fundamental metric that can characterize the cause of the nonzero duality gap?
>
> **A:** Yes. In fact, even partially product form of optimal policy (e.g., only one agent is independent from all the other agents) can lead to nonzero duality gap. For example, the policy $\pi(a|s)=\pi^{(1,2)}(a^{(1,2)}|s)\pi^{(3)}(a^{(3)}|s)$ is a partially product policy for three agents, where agents 1 and 2 take joint action $a^{(1,2)}$ that is independent from the action $a^{(3)}$ taken by agent 3. Such a (partial) product form of the policy is the only reason that causes the nonzero duality gap.
>
> **Q4:** Is it possible to remove such gap dependence? Which gap is better, duality gap and advantage gap? As shown in convergence analysis, the effect of the duality gap in different algorithms can be different. Does this suggest a better way to design algorithms?
>
> **A:** Great questions. Our current analysis cannot avoid the duality gap for the primal-dual algorithm and the advantage gap for the primal algorithm. We think the dependence of the primal-dual algorithm on the duality gap is unavoidable due to the nature of primal-dual updates. In general, we believe this is a challenging problem that deserves future research.
>
> On the other hand, we find that it is possible to remove the advantage gap for the primal algorithm by carefully selecting the initialization point. Specifically, in Example 1, all the advantage gaps $\zeta_0, \zeta_1, \zeta_2>0$, but the primal algorithm can still converge to the optimal solution under certain choices of initialization. Hence, the primal algorithm can be very sensitive to the initialization, indicating that the dependence of the convergence on the advantage gap $\zeta_k$ may not necessarily be tight. This indicates the possibility to address this issue by carefully selecting the initialization, which we leave for future study.
>
> It is unclear whether the duality gap or the advantage gap is better, as neither of the primal-dual and primal algorithms strictly outperform the other one.

---

> > ### Author Response · Authors · 2023-11-18
> > **Authors' reply to Reviewer MmRm (Q5-Q6)**
> >
> > **Q5:** The advantages of the two algorithms are discussed in terms of policy iterates, which are stronger than the output policy in algorithms. Is it possible to show them in convergence theory?
> >
> > **A:** We think the reviewer was referring to the last-iterate convergence of the policy $\pi_T$. If so, we think it is possible to prove this result by leveraging some existing techniques. Specifically, for the primal-dual algorithm, the existing work [2] has established last-iterate convergence in single-agent constrained RL by introducing a **regularized** Lagrangian function. This idea can be applied to our multi-agent setting to prove the last-iterate convergence. For the primal algorithm, we may consider introducing entropy-regularization to the natural policy gradient (NPG) updates,
> > since entropy-regularized NPG has  been shown to have last-iterate convergence in single-agent RL [3].
> >
> > [2] Ding, D., Wei, C. Y., Zhang, K., \& Ribeiro, A. (2023). Last-Iterate Convergent Policy Gradient Primal-Dual Methods for Constrained MDPs. ArXiv:2306.11700.
> >
> > [3] Cen, S., Cheng, C., Chen, Y., Wei, Y., \& Chi, Y. (2022). Fast global convergence of natural policy gradient methods with entropy regularization. Operations Research, 70(4), 2563-2578.
> >
> > **Q6:** As mentioned, prior works also studied constrained cooperative Markov games and the authors have improved the analysis. Can the authors illustrate the analysis differences due to the lack of zero duality gap? It is useful if the authors could compare assumptions and results with them in a table.
> >
> > **A:** Thanks for the great suggestion. To the best of our knowledge, no existing approaches for constrained cooperative MARL has convergence guarantee on the optimality gap. Specifically, as we discussed after Q1 in the introduction, the existing works only obtained  either convergence of gradient norm (weaker than the optimality gap), or convergence of only part of the policy instead of the full algorithm. In the revision, we have added Table 1 to summarize this comparison.

---

> ### Comment · Reviewer_MmRm · 2023-11-22
>
> Thank you for the response. I have a few more questions.
>
> For Q1, the difference with the work [1] is a bit vague to me. As I see, [1] also considers the product policy, and illustrates non-zero duality in a constrained QP problem.
>
> For Q3, do partially product policies also introduce certain quadratic constraints in the occupancy measure?

---

> > ### Author Response · Authors · 2023-11-22
> > **Authors' 2nd reply to Reviewer MmRm on Q1 and Q3**
> >
> > Dear Reviewer MmRm:
> >
> > Thank you for providing additional feedback,.
> >
> > Regarding your Q1, we double checked [1] and agree that it also considers the product policy, and proves the non-zero duality gap result by constructing a constrained QP problem (eq. (9) in [1]) similar to ours. Given this, we have changed our Theorem 2 on nonzero duality gap into Fact 1.
> >
> > Regarding your Q3, yes, partially product policies also introduce similar quadratic constraints in the occupancy measure. For example, $\pi(a|s)=\pi^{(1,2)}(a^{(1,2)}|s)\pi^{(3)}(a^{(3)}|s)\pi^{(4,5,6)}(a^{(4,5,6)}|s)$ is a partially product policy for 6 agents, which can be seen as a product policy among 3 teams, namely, agents 1-2, agent 3 and agents 4-6. Hence, following the same proof logic of Theorem 1, policy $\pi$ can be factorized in the above form if and only if $\nu=\nu_{\pi}$ satisfies the following quadratic constraints
> > $$\nu(s,a)\sum_{a'}\nu(s,a')=\sum_{a'^{(1,2)}}\nu(s,[a'^{(1,2)},a^{(3,4,5,6)}])\sum_{a'^{(3,4,5,6)}}\nu(s,[a^{(1,2)},a'^{(3,4,5,6)}])$$
> > $$\nu(s,a)\sum_{a'}\nu(s,a')=\sum_{a'^{(3)}}\nu(s,[a'^{(3)},a^{(1,2,4,5,6)}])\sum_{a'^{(1,2,4,5,6)}}\nu(s,[a^{(3)},a'^{(1,2,4,5,6)}])$$
> > $$\nu(s,a)\sum_{a'}\nu(s,a')=\sum_{a'^{(1,2,3)}}\nu(s,[a'^{(1,2,3)},a^{(4,5,6)}])\sum_{a'^{(4,5,6)}}\nu(s,[a^{(1,2,3)},a'^{(4,5,6)}])$$
> > The same logic applies to any partially product policy by taking each team as a super-agent.

---

### Official Review · Reviewer_Ycbe · 2023-11-06

**Soundness:** 3 good
**Presentation:** 3 good
**Contribution:** 3 good
**Rating:** 6
**Confidence:** 4

**Summary:**

The paper provides a comprehensive analysis of the strong duality condition in constrained cooperative Multi-Agent Reinforcement Learning (MARL), (for the first time) revealing its failure to hold and its impact on convergence rates of primal-dual algorithms. Then the authors presents/proposes a new decentralized primal algorithm to avoid the duality gap in constrained cooperative MARL. But their analysis shows that the convergence of this new algorithm is hindered by another gap induced by the advantage functions.The authors contribute to the understanding of the complexity of the constrained cooperative MARL problem by comparing it to cooperative MARL and single-agent constrained RL, and It is rigorously showed in this paper that constrained cooperative MARL is fundamentally harder than its special cases of cooperative MARL and constrained RL. Note that, before this work, strong duality has not been formally validated in constrained cooperative MARL, and therefore leaving convergence of the existing primal-dual type algorithms obscure.

**Strengths:**

Strengths:

The problem studied in this paper is important and interesting, and the valuable findings here could be beneficial to MARL research community.

The paper provides a comprehensive analysis of the strong duality condition in constrained cooperative Multi-Agent Reinforcement Learning (MARL), (for the first time) revealing its failure to hold and its impact on convergence rates of primal-dual algorithms.And then the authors present/propose a new decentralized primal algorithm to avoid the duality gap in constrained cooperative MARL.

The authors compare the primal-dual algorithm with the primal algorithm and show that neither of them always outperforms the other in constrained cooperative MARL, both theoretically and experimentally. Such theoretical and empirical  analysis are valuable to better understand hardness of constrained cooperative MARL and the performances of different algorithms.

The authors contribute to better understanding of the complexity of the constrained cooperative MARL problem by comparing it to cooperative MARL and single-agent constrained RL.

**Weaknesses:**

The authors identify and reveal the issue about the previous primal-dual algorithms for constrained cooperative MARL, which is valuable, but the contribution would be much more significant if the authors could also propose a solution to successfully solve this identified problem .

The authors did attempt to propose a new decentralized primal algorithm to resolve the detected issue/challenge, but it seems no much success of the proposed solution. The proposed decentralized primal algorithm's convergence is hindered by a gap induced by the advantage functions, which can be seen as a major limitation. The comparison of the proposed decentralized primal-dual algorithm with the existing primal-dual algorithm doesn't seem to clearly indicate that the proposed new approach is a consistently superior approach, which makes the paper's contribution less significant.

The paper is highly theoretical, and comprehensive empirical validation/comparison of the proposed solutions are mostly missing. It would be great to also see some more comprehensive empirical experiment analysis on broad representative tasks (rather than just some very limited extreme case examples in current manuscript).

(Though I have to admit that, although the authors are not able to successfully propose a solution to solve the issue, (for the first time) identifying this important problem/issue about  primal-dual algorithms for constrained cooperative MARL itself might be already quite valuable, and its contribution might possibly enough for publication on ICLR. Though if they are able to also provide a successful solution (in addition to identifying the problem), it would be a much stronger paper.)

**Questions:**

see weakness section comments.

---

> ### Author Response · Authors · 2023-11-18
> **Authors' reply to Reviewer Ycbe**
>
> Thank you very much for reviewing our manuscript and providing valuable feedback. Below is a response to the review comments. We have submitted a revised version with all revisions marked in **"red"**. Please let us know if further clarifications are needed.
>
> **Q1:** The contribution would be much more significant if the authors could also propose a solution to successfully solve this identified problem about the previous primal-dual algorithms for constrained cooperative MARL.
>
> The proposed decentralized primal algorithm's convergence is hindered by a gap induced by the advantage functions, which can be seen as a major limitation. The comparison of the proposed decentralized primal-dual algorithm with the existing primal-dual algorithm doesn't seem to clearly indicate that the proposed new approach is a consistently superior approach, which makes the paper's contribution less significant.
>
> **A:** Great comment. We agree that our current analysis cannot avoid the duality gap for the primal-dual algorithm and the advantage gap for the primal algorithm. This is a challenging problem that deserves future research. On the other hand, it is possible to address this issue by carefully selecting the initialization point. Specifically, in Example 1, all the advantage gaps $\zeta_0, \zeta_1, \zeta_2>0$, but the primal algorithm can still converge to the optimal solution under certain choices of initialization. Hence, the primal algorithm can be very sensitive to the initialization, indicating that the dependence of the convergence on the advantage gap $\zeta_k$ may not necessarily be tight. This indicates the possibility to address this issue by selecting the initialization, which we leave for future study.
>
> **Q2:** The paper is highly theoretical, and comprehensive empirical validation/comparison of the proposed solutions are mostly missing. It would be great to also see some more comprehensive empirical experiment analysis on broad representative tasks (rather than just some very limited extreme case examples in current manuscript).
>
> **A:** Thanks for the suggestion. We are currently conducting MARL experiments in a real constrained task and will keep the reviewer posted once the results are available.

---

> > ### Comment · Reviewer_Ycbe · 2023-11-23
> > **I've checked out all comments and keep my rating unchanged**
> >
> > I've checked out the authors' response as well as all the comments from other reviewers. I still think this is a border line paper in terms of contribution significance, and I've kept my rating unchanged.

---

### Official Review · Reviewer_WbB4 · 2023-11-07

**Soundness:** 3 good
**Presentation:** 3 good
**Contribution:** 3 good
**Rating:** 6
**Confidence:** 3

**Summary:**

This work concerns a problem of MARL known as Constrained Cooperative MARL (CC-MARL). In such a setting, multiple agents share a common reward function that depends on joint action and the transitions of the underlying MDP depend on that joint action as well; further, apart from striving to maximize the expected discounted cumulative rewards they will get, the agents attempt to minimize a set of constraints. Those constraints in fact concern the expected discounted cumulative costs.

Existing literature has offered results for correlated policies -- rather than product ones. The authors demonstrate that the problem of CC-MARL admits a formulation as a mathematical program with nonconvex (bilinear) constraints which is known to be NP-hard, in the general case. In general, in such a case, a solution of the CC-MARL is a product policy that will have an approximately zero single-agent optimality gap and constraint violation. The optimality gap is just the gain a single agent can have by unilaterally deviating from the joint output policy and the constraint violation is the amount by which a constraint is violated.

Then, the authors demonstrate how the existing primal-dual algorithmic framework only manages to provide a bound on the constraint violation that depends on the *duality gap* of the underlying lagrangian function. This means that existing art can only offer solutions that could potentially have a constraint violation as large as the maximum possible discounted cumulative reward.

Finally, the authors design an algorithm based on the single-agent RL CRPO algorithm. Convergence is proven using a potential/Lyapunov function argument. The optimality and constraint violation bounds depend on a quantity known as the *advantage gap*. The advantage gap in turn is zero if and only if the q-functions can be decomposed in a sum of functions that only depends on single-agent actions.

**Strengths:**

The authors extend previous work that existed only for the case of correlated policy optimization to a product policy setting. They even improve single-agent RL bounds for the CRPO algorithm.

The paper offers a rather rich exposition of previous theoretical results in the literature of (constrained) cooperative RL.

**Weaknesses:**

A weakness of the authors' result is the lack of a definitive answer as to the hardness of the problem of Constrained Cooperative MARL. Indeed, the fact that the optimization program corresponds to a mathematical program with bilinear constraint functions is an indication of its potential hardness, yet there is a multitude of refined computational complexity classes that it could belong to. Is the problem *total*, i.e., is the problem guaranteed to have a solution? If so, it will belong to the TFNP complexity class. Then, does it belong to some known classes such as PPAD, PPA, CLS, PLS?

I believe the paper has merit in extending previous results that considered correlated policies to quantifying the effect of being restricted to product policies. That being said, the narrative and even the title of the introductory text could benefit by stressing this fact rather than putting the focus on the hardness, since there is no definitive answer of the computational complexity.

**Questions:**

What were the main challenges you faced in proving a definitive refined hardness result?

What would be the advantage gap if the reward functions admitted a network-separable structure and the transitions were additive?

Do you think that the dependence on the advantage gap is tight? Can it be improved, or is it yet another indication of the hardness of approximation of solutions of constrained cooperative MARL problem solutions?

---

> ### Author Response · Authors · 2023-11-18
> **Authors' reply to reviewer WbB4**
>
> Thank you very much for reviewing our manuscript and providing valuable feedback. Below is a response to the review comments. We have submitted a revised version with all revisions marked in **"red"**. Please let us know if further clarifications are needed.
>
> **Q1:** Is the problem guaranteed to have a solution?  Does it belong to PPAD, PPA, CLS, PLS? Change the emphasis of the paper to the main contribution rather than the hardness, since there is no definitive answer of the computational complexity..
>
> **A:** Good question. The optimization problem in Theorem 1 has a solution since the feasible set is compact and the objective function is continuous. However, we do agree that it is challenging to figure out the complexity class of such a problem, and we think this is an interesting and fundamental problem deserving future research.
>
> We really appreciate the reviewer's great suggestion on changing the emphasis of our claim. We do agree that our statement of hardness does not reflect the actual contribution of the paper. In the revised paper, we have
> changed "hardness" to "duality gap" in the title, and changed it to "challenging" or "challenges" throughout the paper. We also changed the emphasis of our main contribution, focusing on the nonzero duality gap and development of algorithms and their convergence results.
>
> **Q2:** What were the main challenges you faced in proving a definitive refined hardness result?
>
> **A:** The main challenge is to transform a known hard problem, e.g., NP-hard, into the optimization problem presented in Theorem 1, which has specific and complicated forms. This is necessary to prove that our problem is NP-hard (or belongs to some other complexity classes).
>
> **Q3:** What would be the advantage gap if the reward functions admitted a network-separable structure and the transitions were additive?
>
> **A:** Great question. For example, in the case where the reward functions and transition kernel are separable, i.e.,
>
> $\overline{r}_k(s,a)=\frac{1}{M}\sum _ {m=1}^M r_k^{(m)}(s,a^{(m)})$
>
> and
>
> $\mathcal{P}(s'|s,a)=\sum _ {m=1}^M \theta_m \mathcal{P} _ m(s'|s,a^{(m)}),$
>
> where $\theta_m\ge 0$ and $\sum _ {m=1}^M\theta_m=1$, the Q function is decomposable as follows.
> $$
> Q_k(\pi;s,a)=\overline{r}_k(s,a)+\gamma \sum _ {s'}\mathcal{P}(s'|s,a) V_k(\pi;s')=\sum _ {m=1}^M \widetilde{Q}_k^{(m)}(\pi;s,a^{(m)})
> $$
> where $\widetilde{Q}_k^{(m)}(\pi;s,a^{(m)}):=\frac{1}{M}r_k^{(m)}(s,a^{(m)})+\gamma\sum _ {s'}\theta_m \mathcal{P} _ m(s'|s,a^{(m)})V_k(\pi;s')$. Therefore, the advantage gap vanishes.
>
> **Q4:** Do you think that the dependence on the advantage gap is tight? Can it be improved, or is it yet another indication of the hardness of approximation of solutions of constrained cooperative MARL problem solutions?
>
> **A:** Great question. We think the dependence on the advantage gap is not tight and may be improved by carefully tuning the initialization. To explain, in Example 1, all the advantage gaps $\zeta_0, \zeta_1, \zeta_2>0$, but the primal algorithm can still converge to the optimal solution under certain choices of initialization. This indicates that the primal algorithm can be very sensitive to initialization and that the constrained cooperative MARL problem is challenging. We believe this is an interesting problem for future research.

---

### Official Review · Reviewer_ditR · 2023-11-09

**Soundness:** 3 good
**Presentation:** 4 excellent
**Contribution:** 2 fair
**Rating:** 6
**Confidence:** 2

**Summary:**

The paper first examines whether strong duality holds for the constrained cooperative MARL setting (a problem which has been studied in previous works). In contrast to the constrained single agent case where strong duality holds, the authors reformulate the constrained cooperative MARL problem as a constrained optimization problem on the occupation measure associated with the agents’ product policy, and prove that strong duality does not hold for constrained MARL case, because of the existence of non convex constraints (related to the product joint policy). They establish the first convergence rate result that characterizes the impact of duality gap on the constraint violation and optimality of the output policy of the Primal-Dual algorithm. In particular, both the optimality gap and the constraint violation converge at a similar sub-linear rate, but the latter up to a convergence error that depends on the duality gap of the problem. Furthermore, the paper proposes a primal-based algorithm for constrained cooperative MARL with the convergence not involving the duality gap, based on decentralized NPG policy updates. In particular, the authors show that both the optimality gap and the constraint violation converge at the sublinear rate, up to certain convergence errors that depend on defined advantage gaps. The authors also show that these advantage gaps vanish if and only if the Q-function has a certain factorization scheme. Last but not least, the paper explicitly compares the two algorithms and proves that each of the two algorithms can be better than the other in certain scenarios.

**Strengths:**

- The paper is well-motivated, since strong duality has not been validated in previous constrained cooperative MARL works and the duality gap is crucial for the convergence of the Primal-Dual Algorithm.
- The paper is very well-written and easy-to-follow with concrete examples and good intuition of the examined algorithms.
- The paper introduces some novel technical elements, such as the upper bounds in inequalities (17) and (19) on page 8.

**Weaknesses:**

- The assumption of the decomposition schema of the Q-function, that the advantage gaps depend on, seems limiting. Is the assumption necessary to ensure the computational tractability of the problem? Moreover, in practice, similar assumptions (on the linear decomposition of the Q-function) can harm performance, even in the unconstrained setting (e.g. see the comparison between VDN and QMIX (Rashid et al. 2020)).
- It is not clear by the authors if the paper can be compared with other state-of-the-art algorithms (if exist) in terms of the optimality gap (in the constrained cooperative MARL setting).

**Questions:**

I have the following questions regarding some technical details of the paper:
- In the proof of Theorem 3, the authors have assigned $\lambda_k$ with a value larger than $\lambda_{k,{max}}$ (page 19). Can the authors be more explicit about why they are able to assign $\lambda_k$ with such a value?
- In the proof of Theorem 7, can the authors be more explicit about the proof steps and explain what $\pi_{\omega}$ and $\omega$ are?
- In the proof of Lemma 3, can the authors be more explicit about the last inequality of page 33?

---

> ### Author Response · Authors · 2023-11-18
> **Authors' reply to Reviewer ditR**
>
> Thank you very much for reviewing our manuscript and providing valuable feedback. Below is a response to the review comments. We have submitted a revised version with all revisions marked in **"red"**. Please let us know if further clarifications are needed.
>
> **Q1:** The assumption of the decomposition scheme of the Q-function, that the advantage gaps depend on, seems limiting. Is the assumption necessary to ensure the computational tractability of the problem? Moreover, in practice, similar assumptions (on the linear decomposition of the Q-function) can harm performance, even in the unconstrained setting (e.g. see the comparison between VDN and QMIX (Rashid et al. 2020)).
>
> **A:** Great question. Note that the convergence result in Theorem 4 does not rely on this assumption. Moreover, we showed in Appendix H that this assumption is necessary and sufficient to ensure a vanishing advantage gap $\zeta_k$. We agree with the reviewer that this assumption is restricted and may fail to hold in practical scenarios, and this echos our key point that constrained cooperative MARL problems can be challenging to solve.
>
> **Q2:** It is not clear by the authors if the paper can be compared with other state-of-the-art algorithms (if exist) in terms of the optimality gap (in the constrained cooperative MARL setting).
>
> **A:** Good question. To the best of our knowledge, no existing approaches for constrained cooperative MARL has convergence guarantee on the optimality gap. Specifically, as we discussed after Q1 in the introduction, the existing works only obtained  either convergence of gradient norm (weaker than the optimality gap), or convergence of only part of the policy instead of the full algorithm. In the revision, we have added Table 1 to summarize this comparison.
>
> **Q3:** In the proof of Theorem 3, why can we assign $\widetilde{\lambda} _ k$ to a larger value $2\lambda _ {k,\max}$ in the primal-dual algorithm.
>
> **A:** Thank you very much for pointing out this and we have fixed it in the revised paper. Specifically, we changed the value of $\widetilde{\lambda} _ k$ from $2\lambda _ {k,\max}I\{V_k(\pi_t)\le \xi_k\}$ to $\lambda_{k,\max}I\{V_k(\pi_t)\le \xi_k\}$. Accordingly, we have doubled the value of $\lambda _ {k,\max}$ in Lemma 1 and showed that the optimal dual variable $\lambda_k^*\le \frac{1}{2}\lambda_{k,\max}$ (not $\lambda _ {k,\max}$).
>
> **Q4:** In the proof of Theorem 7, can the authors be more explicit about the proof steps and explain what $\pi_{\omega}$ and $\omega$ are?
>
> **A:** Thanks for checking our proof so carefully. In our revised paper, we have added bold subtitles to highlight the proofs of "if" and "only if" parts. We also added more intermediate steps and explanations to the proof of ``if'' part (i.e. the Q-function decomposition implies $\zeta_k=0$). We have also changed $\pi_{\omega}$ and $\omega$ to $\pi$. Thanks for pointing out this inconsistent notation.
>
> **Q5:** In the proof of Lemma 3, can the authors be more explicit about the last inequality of page 33?
>
> **A:** Thanks for checking our proof so carefully. We think the reviewer was referring to the following inequality.
> $$\sum_a \Big|\prod_{m=1}^M\pi_{t+1}^{(m)}(a^{(m)}|s)-\prod_{m=1}^M \pi_t^{(m)}(a^{(m)}|s)\Big|$$
> $$\le\sum_a \sum_{m'=1}^M\Big|\prod_{m=1}^{m'}\pi_{t+1}^{(m)}(a^{(m)}|s)\prod_{m=m'+1}^{M}\pi_t^{(m)}(a^{(m)}|s)-\prod_{m=1}^{m'-1}\pi_{t+1}^{(m)}(a^{(m)}|s)\prod_{m=m'}^{M}\pi_t^{(m)}(a^{(m)}|s)\Big|$$
> To exlain, for any fixed action $a$, denote $C_{m'}(a):=\prod_{m=1}^{m'}\pi_{t+1}^{(m)}(a^{(m)}|s)\prod_{m=m'+1}^{M}\pi_t^{(m)}(a^{(m)}|s)$ and then the above inequality can be obtained by the following triangle inequality
> $$|C_M(a)-C_0(a)|=\Big|\sum_{m'=1}^M [C_{m'}(a)-C_{m'-1}(a)]\Big|\le \sum_{m'=1}^M |C_{m'}(a)-C_{m'-1}(a)|.$$
> We have added this clarification to the revised paper.

---

### Author Response · Authors · 2023-11-22
**Added experiment on constrained grid-world task**

Dear reviewers,

We have added an experiment on a constrained grid-world task [1] to Appendix K of the revised paper. In this task, we compared our primal-dual algorithm, our primal algorithm, and the centralized nested actor-critic (CNAC) algorithm [1]. The CNAC algorithm is very similar to our primal-dual algorithm, with the only difference that the policy is updated with one projected stochastic policy gradient ascent step.

All these algorithms converge fast to the feasible region $V_1(\pi_t)\ge -1.1$ within 10 iterations. Regarding the optimality, our primal-dual algorithm and primal algorithm converge to the optimal value $V_0(\pi_t)=-2.6695$ within 3 iterations and 70 iterations respectively. The CNAC algorithm converges to a sub-optimal value $V_0(\pi_t)\approx -4.5$ within 10 iterations, since it uses policy gradient ascent update which may get stuck at a stationary point.

We hope these updates can address the reviewers' concern about lack of numerical results. Thank you for your valuable suggestion.

[1] Diddigi, R. B., Danda, S. K. R., \& Bhatnagar, S. (2019). Actor-critic algorithms for constrained multi-agent reinforcement learning. ArXiv:1905.02907.

---

> ### Comment · Reviewer_MmRm · 2023-11-22
>
> Thank you for adding an illustrating experiment. It looks the nonzero duality gap of optimal product policy doesn't make difference between two algorithms. Is this also characterized in their convergence rates?

---

> > ### Author Response · Authors · 2023-11-22
> > **Authors' 2nd reply to Reviewer MmRm on experiments**
> >
> > Thank you for your question. About the experiment, we found that the constraint $V_1(\pi_t)\ge -1$ we used actually leads to zero duality gap, this explains the results that both our primal-dual algorithm and primal algorithm converge well.
> > Just now, we added a new experiment (see the revision) with a more relaxed constraint $V_1(\pi_t)\ge -1.1$ so that the duality gap is nonzero. From the new Figure 4, it can be seen that the learning curves of the constraint value $V_1(\pi_t)$ of all the algorithms suffer from more oscillation due to the nonzero duality gap. Regarding the objective value function $V_0(\pi_t)$, our primal dual algorithm and CNAC approach the near-optimal solution faster than the primal algorithm, and CNAC is the most stable one. We are tuning the hyperparameters and will update more results as soon as possible.

---

### Meta-Review · Area_Chair_FJkp · 2023-12-07

**Metareview:**

The paper focuses on two player zero-sum constrained stochastic games. The first main result of the paper is that the duality gap is not zero (i.e., the minmax theorem of Shapley fails) and the second main result is giving a decentralized algorithm (primal-dual type) that finds a non-markovian non-stationary approximate Nash equilibrium given that the slater condition is satisfied. The write-up of the paper and the title was a bit confusing/misleading and needed improvement as mentioned by the reviews. The authors successfully addressed these issues in the rebuttal. The results are not very surprising but quite interesting and important. The paper is above the threshold bar for acceptance.

**Justification For Why Not Higher Score:**

The paper is does not have super strong results or high score to get a spotlight distinction.

**Justification For Why Not Lower Score:**

The paper has solid results and the issues were adequately addressed in the rebuttal.

---

### Decision · Program_Chairs · 2024-01-16

Accept (poster)